# Targeted cross-linker delivery for the in situ mapping of protein conformations and interactions in mitochondria

Yuwan Chen [1,2], Wen Zhou[1,2], Yufei Xia [3], Weijie Zhang [1,2], Qun Zhao[1], Xinwei Li[1,4], Hang Gao[1,2], Zhen Liang[1], Guanghui Ma [3], Kaiguang Yang [1] ✉, Lihua Zhang [1] ✉ & Yukui Zhang[1]

Current methods for intracellular protein analysis mostly require the separation of specific organelles or changes to the intracellular environment. However, the functions of proteins are determined by their native microenvironment as they usually form complexes with ions, nucleic acids, and other proteins. Here, we show a method for in situ cross-linking and analysis of mitochondrial proteins in living cells. By using the poly(lactic-co-glycolic acid) (PLGA) nanoparticles functionalized with dimethyldioctade-cylammonium bromide (DDAB) to deliver protein cross-linkers into mitochondria, we subsequently analyze the cross-linked proteins using mass spectrometry. With this method, we identify a total of 74 pairs of protein-protein interactions that do not exist in the STRING database. Interestingly, our data on mitochondrial respiratory chain proteins (~94%) are also consistent with the experimental or predicted structural analysis of these proteins. Thus, we provide a promising technology platform for in situ defining protein analysis in cellular organelles under their native microenvironment.

Many proteins in cellular organelles work in concert with other proteins or biomolecules[1], and their compositions and functions change in different subcellular microenvironments due to changes in pH, ions, and membrane presence. For example, the oxidative phosphorylation (OXPHOS) system of the mitochondrial inner membrane is composed of five enzymes that form a higher-order structure with the aid of ancillary proteins[2]. Without the formation of this molecular structure, the catalysis of electron transfer will be interfered during the procedure of ATP synthesis[3]. Therefore, it is critical to gain insight into not only the structures of individual proteins but also their interactions with cognate partners to reveal the fundamental biological process[4]. However, the development of methods for the in situ analysis of these proteins in cellular organelles remains a challenge.

Considerable efforts have been regarding intracellular protein analysis[5–7], with the analysis of organelles isolated from cells recently receiving significant attention[8–11]. Fasci et al. applied cross-linking coupled to mass spectrometry (XL-MS) to the isolated nuclei of human osteosarcoma cells cross-linked using disuccinimidyl sulfoxide (DSSO) to study the nuclear interactome and protein conformation at the proteome level[11]. Multiple in situ XL-MS studies have explored various *N*-hydroxysuccinimide (NHS)-ester cross-linkers to reveal details regarding the mitochondrial proteins and interactions in different complex systems, such as BDP-NHP[8,12], DSSO[9], cyanurbiotindi-propionylsuccinimide (CBDPS)[13], disuccinimidyl suberate (DSS)[10,14], and *bis*(sulfosuccinimidyl) suberate (BS3)[14]. Schweppe et al. [8] and Liu et al. [9] applied XL-MS to mitochondria isolated from mouse heart tissue and provided the mitochondrial protein structure (e.g., five

[1]CAS Key Laboratory of Separation Science for Analytical Chemistry, Dalian Institute of Chemical Physics, Chinese Academy of Sciences, Dalian 116023, China. [2]University of Chinese Academy of Sciences, Beijing 100049, China. [3]State Key Laboratory of Biochemical Engineering, Institute of Process Engineering, Chinese Academy of Sciences, Beijing 100190, China. [4]Zhang Dayu School of Chemistry, Dalian University of Technology, Dalian 116024, China. ✉e-mail: yangkaiguang@dicp.ac.cn; lihuazhang@dicp.ac.cn

OXPHOS complexes) and interaction information, respectively. Ryl et al. [10] applied DSS to human mitochondria isolated from the cultured human cell line K-562 and revealed the protein flexibility of mitochondrial heat shock proteins. Makepeace et al. [13] and Linden et al. [14] used the cleavable cross-linker CBDPS and non-cleavable cross-linkers BS3 and DSS to mitochondria purified from yeast for in situ analysis of mitochondrial protein information, respectively. However, in all previous reports, the direct detection of interactions between the mitochondrial proteins under the culture condition has not been reported at the proteomic level. Although this method could elucidate the protein interactomes and structures, the procedure for organelle isolation inevitably changed the native working environment of the proteins and caused a loss of organelle-bound proteins. Methods have also been developed to directly treat the entire cell with protein cross-linkers [15–17]. For the cross-linking of cells under native conditions, formaldehyde (FA) can serve as a suitable cross-linker for analyzing the proteins within cells because the small FA molecules can rapidly permeate the cell membrane to efficiently cross-link and form covalent bonds with the proteins [18]. To apply FA cross-linking to complex systems, Tayri-Wilk et al. [15] pioneered a straightforward FA mechanism, whereby FA adducts lead to distinct reaction products with a mass of 24 Da, which was applied to the detection of cross-linked peptides in human cells fixed in situ with FA. In addition, NHS-esters are the most frequently used reactive group for cross-linkers because they can react with primary amines in proteins under physiological conditions (pH 7.0–7.5) [19]. However, most water-soluble cross-linkers have a very short residence time in cells [20]. In addition, organelles have transport barriers, which further limit the cross-linker concentration in organelles. Thus, protein cross-linking in cellular organelles is insufficient. Moreover, their high reactivity may lead to non-specific, undesired protein cross-linking in the cytosol, potentially causing high-background noise for downstream precise analysis. Water-insoluble cross-linkers can be used to treat cells after dissolution in organic solvents. However, the use of organic solvents will change the native subcellular environment or the structures of the proteins. Therefore, we studied a method for the in situ analysis of proteins in their native subcellular environments.

Nanoparticles (NPs) for targeted drug delivery have been widely studied, offering significant potential for changing the paradigm of treating human diseases with high efficacy but minimal side effects. However, the application of NPs for in situ protein analysis in subcellular environments has received little attention.

In this work, we develop a method for in situ mitochondrial protein mapping with targeted cross-linker delivery coupled with mass spectrometry (CD-MS). DSS is encapsulated in the PLGA NPs, which are functionalized with DDAB for targeting mitochondria. After intracellular delivery of the cross-linker, the cross-linked proteins are extracted through ionic liquid filtration-assisted sample preparation (i-FASP). The peptides are then fractionated using reversed-phase liquid chromatography (RPLC) for measurement in a high-throughput mass spectrometer, followed by the analysis of the protein-protein interactions with Cα-Cα distances of less than 30 Å, serving as an upper bound distance for DSS cross-linking supported by molecular dynamics simulations. With the aid of the DDAB-functionalized PLGA NPs, we achieve the release of DSS in situ in the mitochondria of HepG2 cells in the culture condition, and the protein complex is frozen at a transient condition in physical and native environments at the same time. Subsequently, the MS-based peptide sequencing technique and conventional database searching tools are used to detect the cross-linked peptides. CD-MS make it possible for the in situ/in vivo analysis of protein conformations and interactions using mass spectrometry (MS) at the organelle level in the living cells. Furthermore, we anticipate that CD-MS could be developed into a variable tool to achieve controllable cross-linking in targeted organelle by changing the targeting moiety of the nanocarrier, along with in situ release of the cross-linker into the subcellular organs of living cells.

## Results

### Establishment of a method for mitochondrial protein and interactions analysis

To define the subcellular-level protein information in living cells, the CD-MS approach employed a NP delivery strategy for sample cross-linking, combined with detection by high-resolution MS. The CD-MS method for large-scale analysis of the mitochondrial proteome in living cells consisted of five steps: 1) targeted delivery of the cross-linker to the mitochondria by the NPs; 2) extraction and digestion of the proteins by ionic liquid filtration-assisted sample preparation (i-FASP); 3) separation of the peptides using RPLC to reduce sample complexity; 4) high-throughput and sensitive detection of peptides using a high-performance Q-Exactive instrument; and 5) identification of the cross-linked peptides for matching protein structures and mapping protein interactions (Fig. 1).

We were inspired by the targeted NP delivery concept to develop an innovative protein analysis method, where the cross-linker-based NPs were carefully designed to meet the requirements of the CD-MS method for analyzing mitochondrial protein conformations and interactions in living cells. Therefore, there were three NPs designs to meet these needs. First, the PLGA consisted of the skeleton polymer of the NP, due to its excellent hydrophobic drug delivery ability, drug activity retention, and low biological toxicity according to previous reports [21–23]. The non-ionic surfactant Kolliphor EL was employed as the core of the nanocarrier to help the hydrophobic cross-linker (e.g., DSS) solubilize in the PLGA matrix, for high loading efficiency by alleviating the interfacial tension during the formation of the NP [24]. Second, the mitochondrial inner membrane was negatively charged; thus, the positively charged dimethyldioctadecylammonium bromide (DDAB) ligands were innovatively designed doped with PLGA as the outer shell of the NP, providing a positive charge in the nanocarrier to target the mitochondria. Finally, the commercial NHS-ester cross-linker DSS was chosen as the model cross-linker, as it could efficiently react with primary amines in the proteins under physiological conditions [19]. In this manner, the oil-in-particle structure enhanced the stability of the hydroxy-sensitive DSS. During the formation of the PLGA/Kolliphor EL-hybrid particles, phase separation of the oil core and PLGA occurred inside the o/w droplet. Since DSS was more inclined to dissolve within the Kolliphor EL, internal phase separation prevented the encapsulated DSS from the pre-exposure of the fluidic phase.

Hence, by using the emulsion solvent-evaporation method, we fabricated spherical DSS-DDAB@PLGA/Kolliphor EL NPs containing the above design elements. The DSS-DDAB@PLGA/Kolliphor EL NPs consisted of a uniform sphere with a core-shell structure according to the TEM results (Fig. 2a). By exposing the hydrophilic ammonium cation on the PLGA/DDAB shell of the nanocarrier, the DSS-DDAB@PLGA/Kolliphor EL NP surfaces had a positive charge of -37.9 mV (Fig. 2b and Supplementary Fig. 7a), and the surface charge of the DSS@PLGA/Kolliphor EL NPs without DDAB was approximately −25.1 mV (due to the carboxylic groups) (Supplementary Fig. 7a). The loading efficiency of DSS in the DSS-DDAB@PLGA/Kolliphor EL NPs was optimized by adjusting the amount of DSS, which was increased to 23.21% (w/w) (Fig. 2c) without causing changes in the size or zeta potential (Fig. 2b and Supplementary Table 1). Therefore, the NPs with the highest loading efficiency (70DSS-DDAB@PLGA/Kolliphor EL NPs) and a positive charge of -37.9 mV were chosen for the following applications. Furthermore, the reactivity of DSS in Kolliphor EL was confirmed using SDS-PAGE; specifically, obvious dimer cross-linked bands (Bovine Serum Albumin (BSA) dimer) were observed in both the experimental (DSS in Kolliphor EL for dilution) and positive-control group (DSS in DMSO for dilution) compared with the negative control experiment (pure BSA) (Supplementary Fig. 1). After 6 h of NP delivery, 4.86% of the proteins in the treated HepG2 cells showed a 2-fold significant difference (Supplementary Fig. 2). Thus, the cross-linker-based nanocarriers did not affect most of the detected 3002 cellular proteins

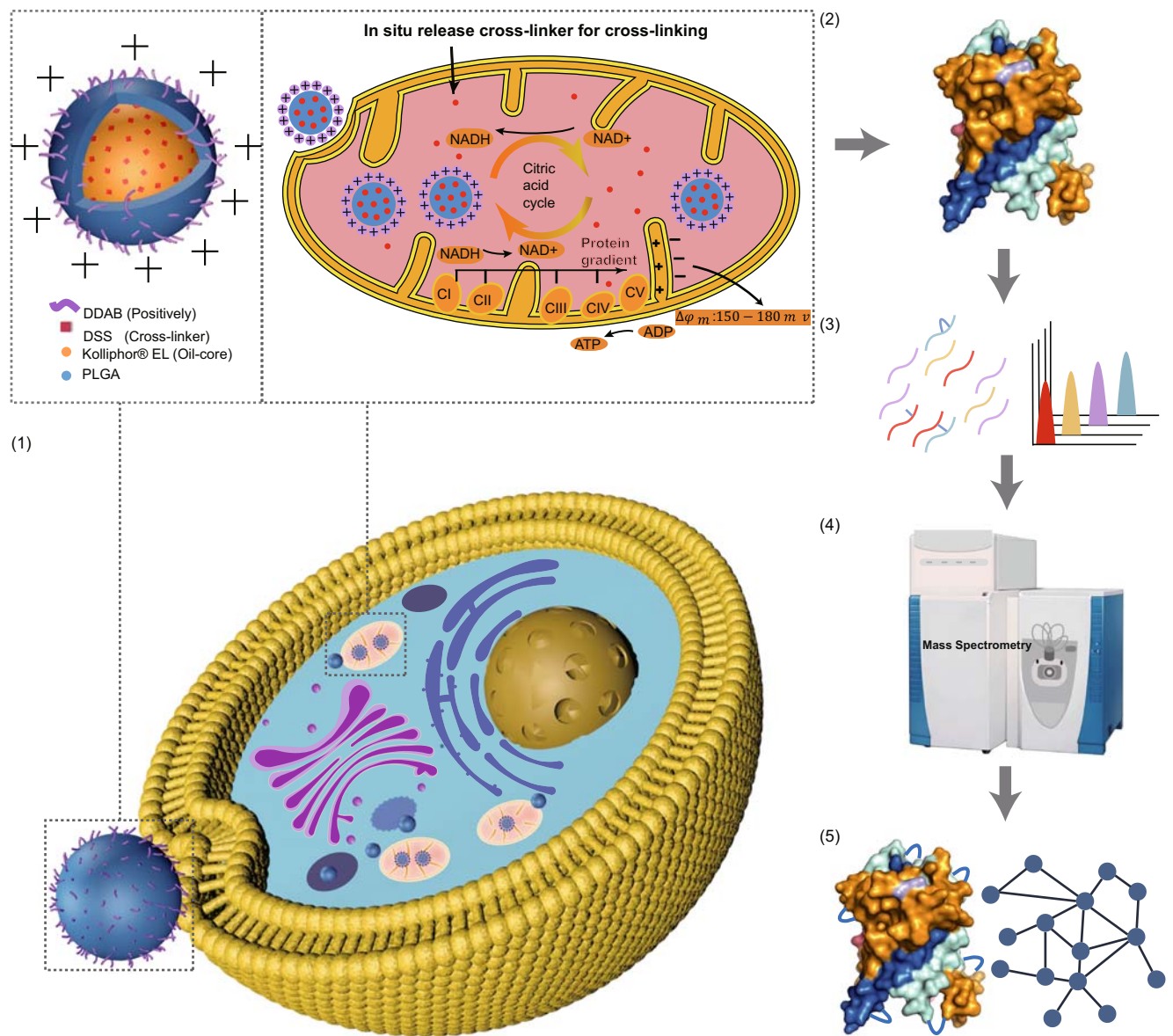

**Fig. 1 | Schematic illustration of the targeted cross-linker delivery coupled with mass spectrometry (CD-MS method).** (1) The nanoparticles (NPs) delivered the cross-linker to the mitochondria in the HepG2 cells for in situ cross-linking. Dimethyldioctadecylammonium bromide (DDAB), disuccinimidylsuberate (DSS), poly (lactic-co-glycolic acid) (PLGA), complex I–V: CI, NADH:ubiquinone oxidoreductase; CII, succinate:ubiquinone oxidoreductase; CIII, cytochrome $bc_1$ complex; CIV, cytochrome *c* oxidase and CV, ATP synthase. (2) The protein samples were extracted from the HepG2 cells, which were then hydrolyzed into peptides by trypsin. (3) The peptides were then fractionated using reversed-phase liquid chromatography (RPLC). (4) The peptide samples were eluted and ionized in LC-MS/MS for identification. (5) The cross-linked peptides were identification for matching the protein structures and mapping the protein interactions.

(2856 cellular proteins), which were not significantly different from the control at the proteomic level. In addition, when the DDAB content in the 70DSS-DDAB@PLGA/Kolliphor EL NPs was greatly reduced, the percentage of disrupted proteins in the proteome level decreased in the treated HepG2 cells (Supplementary Fig. 2). A puzzling fact was the lack of reports on the percentage of protein change in the results of XL-MS; thus, no standard was available for reference. This result also suggested that opportunities may still be available to tune NPs to be more biocompatible in the future.

To inspect intracellular locations of the positively charged DDAB@PLGA/Kolliphor EL NPs in the HepG2 cells, four subcellular organelle dyes were used as a probe to inspect the intracellular co-localization of the PLGA/Kolliphor EL NPs and organelles. In addition, both Cy5 and DSS-FITC were used to fully label the DDAB@PLGA/Kolliphor EL NPs to track the distribution of the NPs and DSS in the cells, respectively, as confirmed by flow cytometry experiments (Supplementary Fig. 6). The red fluorescence of the DDAB@PLGA/Kolliphor EL NPs and the green fluorescence of the Mito Tracker matched well with the Pearson correlation coefficient (PCC) of 0.73 (Fig. 2d) at 6 h, suggesting the mitochondria targeting ability of the DDAB@PLGA/Kolliphor EL NPs. By contrast, the negatively charged NPs (without DDAB NPs) showed significantly reduced endocytosis and weaker co-localization with mitochondria (PCC was 0.28) (Supplementary Fig. 7d). Meanwhile, the fluorescence of DSS-FITC from the DSS-FITC-DDAB@PLGA/Kolliphor EL NPs showed obvious overlaps with the mitochondria with a PCC of 0.81 (Fig. 2e). Furthermore, the Cy5-labeled experimental NPs were loaded with DSS-FITC, which was a DSS-modified fluorescent molecule. The overlap of fluorescent images between the embedded DSS-FITC and cy5-labeled experimental NPs (PCC was 0.83) also demonstrated that DSS was transported to the mitochondria by the positively charged DDAB-based NPs, instead of by diffusion (Supplementary Fig. 7e). The confocal laser scanning

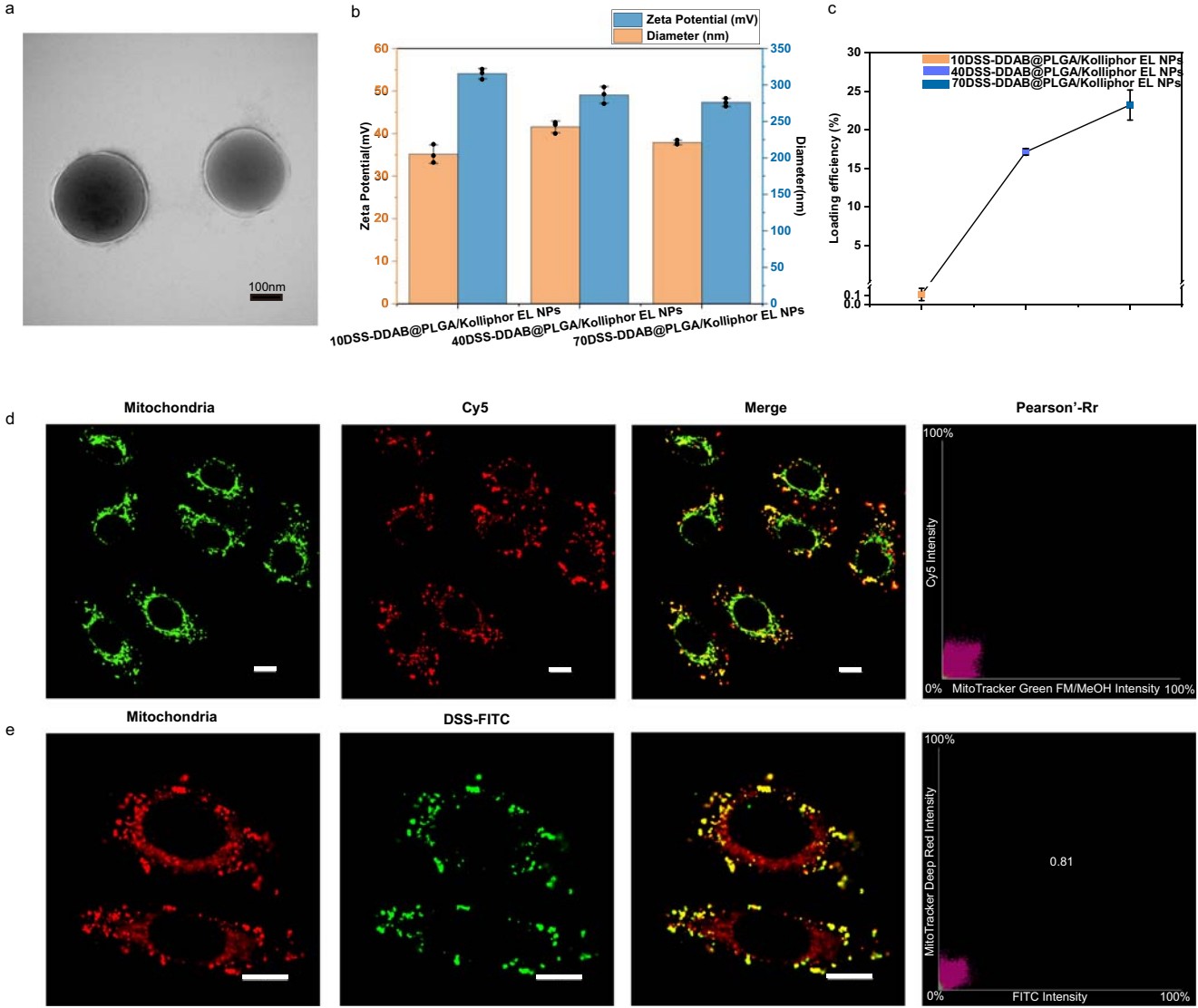

**Fig. 2 | Characterization of the nanoparticles (NPs) and experimental evidence for their co-localization with the mitochondria. a** TEM images of the 70DSS-DDAB@PLGA/Kolliphor EL NPs (scale bar = 100 nm). **b** Size (blue) and zeta potential (orange) of 10DSS-DDAB@PLGA/Kolliphor EL NPs, 40DSS-DDAB/PLGA/Kolliphor EL NPs and 70DSS-DDAB@PLGA/Kolliphor EL NPs in pure water measured using DLS. Data in (**b**) was presented as mean values ± SD, *n* = 3 independent experiments. **c** The loading efficiency of DSS in the 10DSS-DDAB@PLGA/Kolliphor EL NPs, 40DSS-DDAB@PLGA/Kolliphor EL NPs and 70DSS-DDAB@PLGA/Kolliphor EL NPs, measured using HPLC. Data in (**c**) was presented as mean values ± SD, *n* = 3 independent experiments. **d** Co-localizations of the PLGA/Kolliphor EL NPs/mitochondria, as measured using CLSM, with the PLGA/Kolliphor EL NPs and

mitochondria labeled with Cy5 (red) and MitoTracker Green (green), respectively. Scale bars = 10 μm. **e** Co-localizations of the DSS-FITC-DDAB@PLGA/Kolliphor EL NP/mitochondria, as measured using CLSM, where DSS was functionalized with FITC (green). The mitochondria were labeled with MitoTracker Deep Red (red). Scale bars = 10 μm. Dimethyldioctadecylammonium bromide (DDAB), disuccinimidyl suberate (DSS), poly (lactic-co-glycolic acid) (PLGA). Confocal images in d and e were both processed by the deconvolution method in NIS-Elements AR 5.20.00 (Nikon). Confocal image d and e from co-localization experiments are representative of biological duplicates. Source data for (**a–c**) are provided as a Source Data file, and source data for (**b–e**) also are available in the figshare repository (https://doi.org/10.6084/m9.figshare.23261618).

microscopy (CLSM) results confirmed that the encapsulated DSS was successfully delivered to the mitochondria of the HepG2 cells via the NPs. For the remaining three subcellular organelles, the cy5-labeled DDAB@PLGA/Kolliphor EL NPs and lysosome showed poor co-localization at 6 h with a PCC of 0.37 (Supplementary Fig. 8a), indicating that the DDAB@PLGA/Kolliphor EL NPs could escape from the lysosome without retention. Meanwhile, as shown in Supplementary Fig. 9, we observed decreased co-localization of the cy5-labeled DDAB@PLGA/Kolliphor EL NPs and lysosomes within 8 h of co-culturing of the NPs with the cells. We deduced that the cationic surface triggered a proton sponge under the low pH of the lysosome, resulting in escape for further delivery[25]. In addition, the red signals of the DDAB@PLGA/Kolliphor EL NPs showed weak overlap with both the

endoplasmic reticulum (PCC was 0.44 (Supplementary Fig. 8b)) and Golgi apparatus (PCC was 0.18 (Supplementary Fig. 8c)). These results further confirmed that the NPs specifically targeted the mitochondria after endocytosis. Furthermore, the release of the cross-linker was quantitatively assessed using in vitro DSS release profiles, and approximately 28.7% of the cumulative release was observed at 6 h (Supplementary Table 2).

To elucidate the effect of proteins adsorbed on the NPs on in situ mitochondrial cross-linking, the interactions of the 70DSS-DDAB@PLGA/Kolliphor EL NPs with proteins in different biological media were identified before targeting the mitochondria. From the CLSM experiment, we observed that the NPs were endocytosed by HepG2 cells in the DMEM medium, and then escaped from the

lysosomes to the cytoplasm for mitochondrion targeting. Thus, to mimic the internalization pathway, the NPs were incubated in DMEM medium to form CDNP[D] (70DSS-DDAB@PLGA/Kolliphor EL NP DMEM), which were then sequentially transferred into freshly isolated lysosomes to form CDNP[DL] (70DSS-DDAB@PLGA/Kolliphor EL NP DMEM-lysosome) and finally incubated with cytosol extract to form CDNP[DLC] (70DSS-DDAB@PLGA/Kolliphor EL NP DMEM-lysosome-cytoplasm) (Supplementary Fig. 10a, b). Subsequently, the protein corona information in the three biological media (DMEM medium, isolated lysosomal, and cytosol extracts) was separately analyzed. The amount of protein on the NPs was quantified as ~0.115 fg per particle on the CDNP[D], ~0.274 fg per nanoparticle on the CDNP[DL], and ~2.692 fg per particle on the CDNP[DLC] (Supplementary Fig. 11a). The CDNP[DL] adsorbed 0.07% of the protein in the isolated lysosomal media, while CDNP[DLC] adsorbed 0.10% of the protein in the cytosolic extraction media, compared with the total amount of protein in each biological media of the cell. Furthermore, given single HepG2 cells contained ~0.17 ×10$^6$ fg of total protein, and the mitochondria in a single HepG2 cell contained ~2.0 × 10$^4$ fg of total protein[26], the amount of protein adsorbed on the surface of the NPs was very small. To explore the hard proteins consistently adsorbed on the NPs, the overlap between the proteins on the three NPs was analyzed using a Venn diagram (Supplementary Fig. 11b and Supplementary Table 3). At each stage, approximately half of the proteins adsorbed on the NPs were in the soft corona layer, where they were weakly adsorbed on the NP surface and dynamically exchanged in the biological fluids. Considering that approximately 8.2% of all adsorbed proteins (21 overlapping proteins) were consistently present during the entire sequential incubation process, we reasoned that the DSS-DDAB@PLGA/Kolliphor EL NPs adsorbed proteins during delivery, but the hard corona, the proteins strongly bound to the NPs, only accounted for a small fraction of the total. In addition, we found that the mitochondrial proteins detected in our study were independent of the proteins absorbed on the NPs before targeting mitochondria, so the proteins that were absorbed on the NPs did not affect the in situ release of the cross-linker in the mitochondria for cross-linking (Supplementary Fig. 11c).

The effect of the CD-MS method for labeling the mitochondrial proteins was also compared with those of the in vivo XL-MS method and in vitro mitochondrial XL-MS method[10]. First, the ratio of labeling mitochondrial peptides in all detected peptides (α) was individually calculated among the three methods[27]. We found that the CD-MS method had better specificity among the three methods, and the α of the CD-MS method was 4.11 times higher than the in vivo XL-MS method (Supplementary Fig. 12a). Second, the number of mitochondrial proteins identified using the CD-MS method was higher in the three replicates than in the control group (Supplementary Fig. 12b), with an average of 149 mitochondrial proteins in the Homo sapiens database and 80 mitochondrial proteins in the Mitocarta 3.0 database[28]. Third, both the CD-MS method and in vitro mitochondrial XL-MS method detected unrevealed protein-protein interactions (PPIs) in the mitochondria, indicating the potential to independently identify unreported interactions (Supplementary Fig. 12c). In addition, the Tom 20 proteins[29,30], the main gate for molecules entering the mitochondria, were only cross-linked in the CD-MS method, which indicated that the DSS-DDAB@ PLGA/Kolliphor EL NPs in the cytoplasm were likely transported into the mitochondrial matrix by the TOM (Supplementary Fig. 12d).

### Protein structure and interaction analysis of mitochondria in living cells

Our present data demonstrated that the engineered NPs could deliver the cross-linker to the mitochondria for cross-linking in living cells. Therefore, the CD-MS method was used for the delivery of the cross-linker to the mitochondria in HepG2 cells in a natural cellular environment. Subsequently, the cross-linked proteins were extracted,

digested, and underwent RPLC fractionation for high-throughput MS. We identified 1203 cross-linked peptides in 503 proteins, which corresponded to 1245 cross-linking sites (cross-links) and 44% of the human mitochondrial proteome (503 of 1136 proteins; MitoCarta 3.0) (Supplementary Fig. 13a). In addition, the copy numbers of the detected proteins spanned a dynamic range of five orders of magnitude that related to protein expression abundance (Supplementary Fig. 13b)[26]. These data demonstrated that our approach could label mitochondrial proteins with deep coverage in HepG2 cells.

Among the 1203 cross-linked peptides, 158 intermolecular cross-linked peptides were included, which revealed 152 pairs of PPIs. The mitochondrial proteins maintained their functionality along with the other proteins and were enclosed in a relatively crowded environment in the workflow of our method. In human cells, we observed 13% inter-protein cross-links, which was higher than the results obtained by Ryl et al. (2.8%)[10], though such a percentage was still lower than that detected in mouse (Schweppe et al. (29%)[8], Liu et al. (61%)[9]) or *S. cerevisiae* (Linden et al. (17%)[14] by other studies (Supplementary Table 4). These differences were possibly due to the different properties of the applied cross-linkers, including the spacer arm length and physico-chemical properties[31], fractionation method[7,32,33], and distinct data acquisition/analysis pipelines[10]. The importance of our work was that we examined mitochondria in living conditions and analyzed mitochondrial cross-linking under culture conditions via NP delivery. Thus, the higher percentage of intermolecular cross-linked peptides was likely attributed to the higher mitochondrial protein concentrations in the living cells. Our results may provide additional physiological context when capturing information, and many interactions captured with our method were dynamic or transient for our measurement strategy. There may still be room for improving our method to achieve a higher percentage of inter-protein cross-links in the future. Moreover, 137 pairs of PPIs were annotated and distributed in all of the subcompartments in the mitochondria (Fig. 3a). Clearly, this method could detect PPIs across a wide spatial extent of mitochondria, which was attributed to the fact that the DSS-DDAB@PLGA/Kolliphor EL NPs could deliver the cross-linker into the mitochondria.

The STRING score of the PPIs could be used to determine the likelihood of any interactions being true[16,34]. In this work, 51.32% (78/152) of the PPIs detected using the CD-MS method were reported in the STRING database, as shown in Fig. 3a, where the line connecting the two proteins is shown in black, 48.71% of which (38/78) were high confidence (score > 0.7) (Supplementary Fig. 13c). The high percentage of high-scoring interactions indicated that the detected PPIs were credible. Additionally, 48.68% (74/152) of the PPIs were not annotated by the STRING, as shown in Fig. 3a, where the line connecting the two proteins is shown in red. Gene ontology (GO) analysis results indicated that the cross-linked proteins involved in non-annotated PPIs were mainly enriched in a cristae formation (Fig. 3b). Mitochondrial cristae are a major site of oxidative phosphorylation, which will undergo highly dynamic morphological changes (e.g., elongation, shortening, and fusion) in living cells[35]. In addition, most of the proteins will be involved in the bioprocesses of the ATP biosynthetic process, tricarboxylic acid cycle, and fatty acid beta-oxidation. These results confirmed that the unrevealed interactions observed in the living cell states were involved in the dynamic process of energy metabolism; thus, the interactions were transient and dynamic.

### Protein structure and interactions information in the TCA cycle from native mitochondria

The structural information conformation of the proteins involved in aerobic respiration, including the tricarboxylic acid cycle (TCA cycle), OXPHOS, and solute carriers, played an important role in the supply capacity of ATP synthesis and cellular metabolism. First, the detected cross-linked peptides belonging to the TCA cycle proteins were analyzed, and the 70 cross-linked peptides that were identified belonged

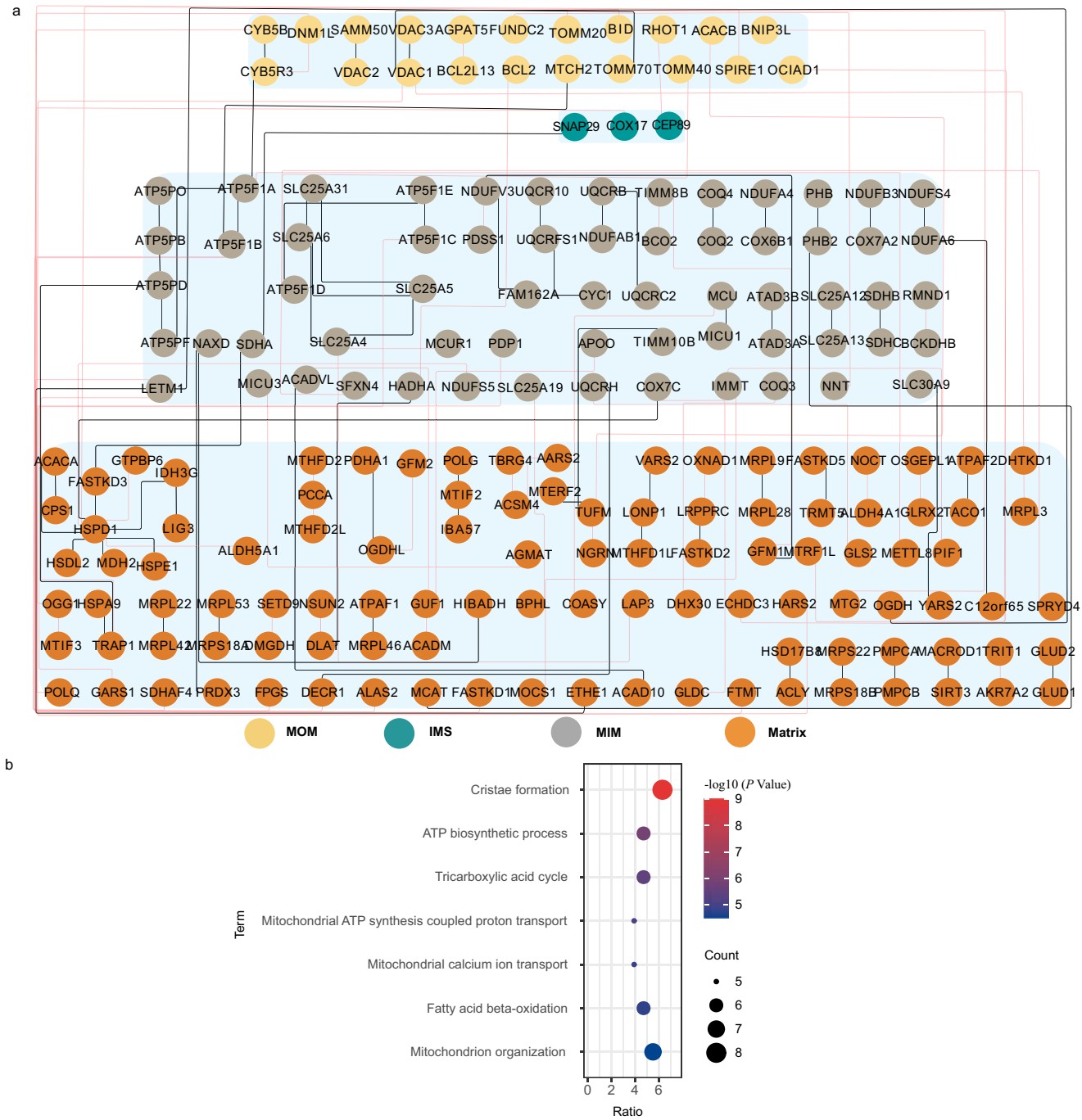

**Fig. 3 | The spatial distribution of the protein-protein interactions involved in the identified cross-linked peptides and Gene ontology biological process (GOBP) analysis of the new discovered protein-protein interactions. a** According to the SubMitoLocalization of the proteins in MitoCarta3, the mitochondrial inter-cross-linked peptides detected in all fractions were sorted and displayed according to the SubMitoLocalization of connexins. MitoCarta3_SubMitoLocalization: mitochondrial outer membrane (MOM), intermembrane space (IMS), mitochondrial inner membrane (MIM), matrix, where the nodes represent the individual proteins, and the lines represent all cross-links identified between the two proteins. The lines linking unreported interactions were shown in red, and the lines linking the reported interactions were shown in black. The proteins of MOM were colored in yellow, the proteins of IMS were dark blue, the proteins of MIM were grey, and the proteins of the matrix were orange. **b** GO analysis results of the identified unannotated protein-protein interactions categorized by the biological process. Node size relative to count points, shown in black. Each node was colored according to the -log₁₀ (P-value) of GOBP analysis by DAVID. The *P*-value number of each GOBP is available in Source Data file. Source data for (**a**, **b**) are provided as a Source Data file, and source data for (**a,b**) also are available in the PRIDE Archive with accession number: PXD035433.

to 24 proteins (Fig. 4a), covering all TCA enzymes, and their biological function pathways are shown in Supplementary Fig. 14[36]. Among these, 68 intramolecular cross-linked peptides provided the structural information for 24 proteins. The vast majority (96%(45/47)) of the intramolecular cross-linked peptides were consistent with the available human structural model in the PDB, and the majority (90%(19/21)) of the intramolecular cross-linked peptides were consistent with the predicted human structural model in AlphaFold (Fig. 4b and Supplementary Fig. 15), as Cα-Cα was observed between the cross-linked lysine, which was less than the maximum linkable distance of DSS (30 Å). This result confirmed the validity of the cross-linked peptides identified in the CD-MS method. Furthermore, two cross-linked

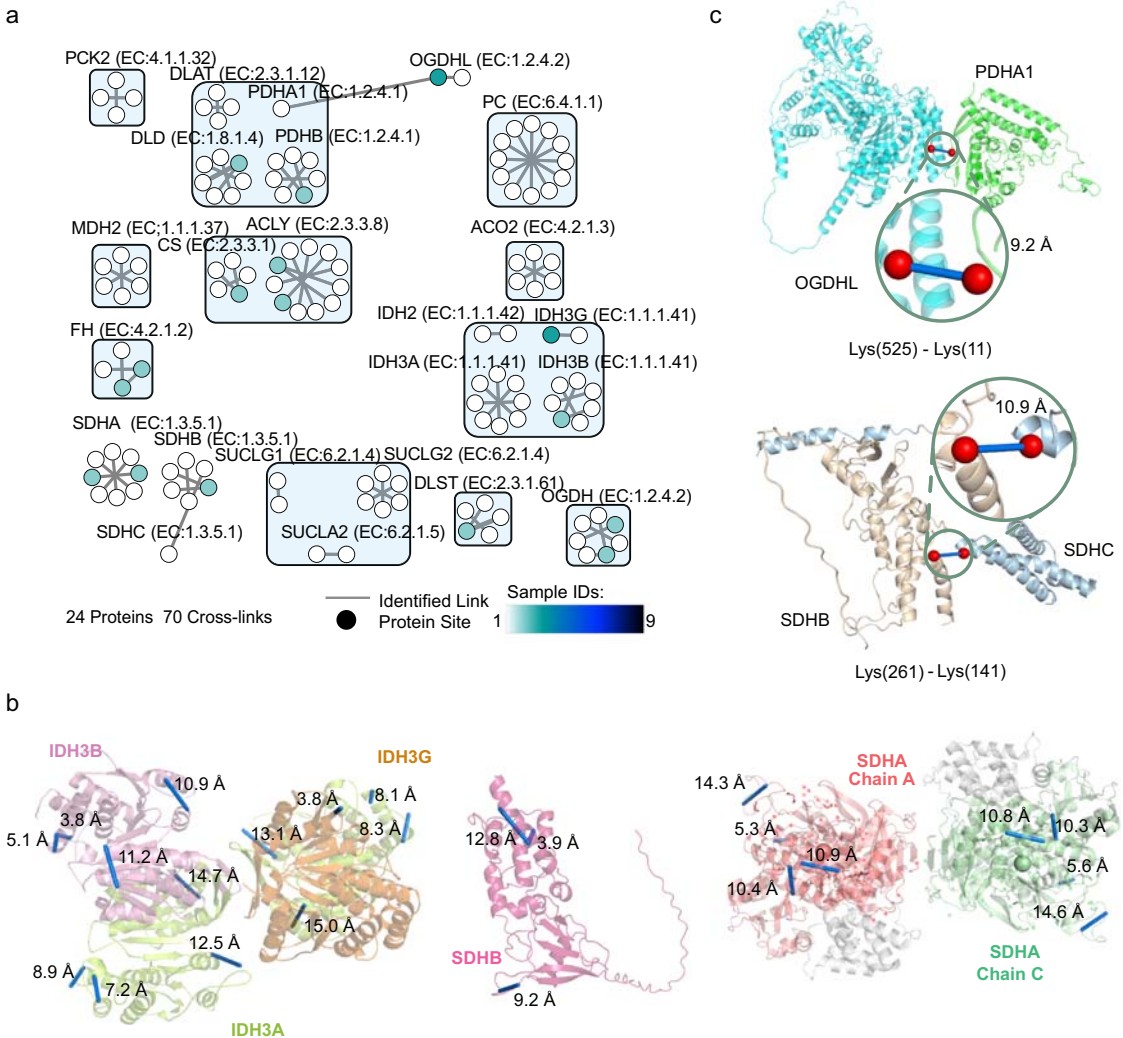

**Fig. 4 | Analysis of the cross-linked peptides of the enzymes of the tricarboxylic acid cycle (TCA cycle) identified in all fractionations. a** The cross-linked peptides of the enzymes of the TCA cycle were selected from Supplementary Fig. 13a and displayed individually, where the nodes represent individual proteins, and the lines represent all cross-links identified between two proteins. Each node was colored based on the number of samples in which each protein interaction was observed, and an interactive network depicting site-to-site interactions is available in Source Data file. **b** The detected cross-linked peptides of the proteins of the TCA cycle enzyme were mapped on the resolved structure in the PDB databases, or the unresolved structure predicted by the AlphaFold model, with each protein chain colored in a different color (including IDH3B (pink); IDH3A (limon); IDH3G (orange); Chain A of SDHA (red) and Chain C of SDHA (pale green). The structural matching information of six proteins was selected for display, and the remainder was shown in Supplementary Fig. 15. No structural information was available for SDHB; thus, the detected cross-linked peptides from the SDHB protein were mapped to the predicted AlphaFold model. The Cα-Cα distances in the links mapped on the structure were within 30 Å, showned in blue. **c** The HADDOCK was used to simulate the structural interaction model between SDHB (light brown) and SDHC (blue), as well as the structural model of the interactions between OGDHL (cyan) and PDHA (green). Among the structural models of the interactions between the two proteins, the interaction model with the shortest Cα distance between the amino acid residues connected by the cross-linked peptide was shown in blue. Source data for (**a**) are provided as a Source Data file, and source data for (**a–c**) also are available in the PRIDE Archive with accession number: PXD035433.

peptides were incompatible with the predicted AlphaFold structure. The residues, which were connected by the two cross-linked peptides, fell within a low model confidence region with the predicted local distance difference test (PLDDT), with a score below 70. Therefore, the discrepancy between the predicted and experimental structures was possibly the reason for the incompatibility of the detected cross-linked peptides with the AlphaFold model (Supplementary Fig. 16).

In addition, two unreported intermolecular cross-linked peptides were detected in TCAase. One was lysine-525 of 2-oxoglutarate dehydrogenase-like (OGDHL) linked to lysine-11 of pyruvate dehydrogenase E1 component subunit α (PDHA1), with a Cα-Cα distance of 9.2 Å, which filled the gap of lack of relevant experimental evidence in the STRING database[37] (Fig. 4c). The other cross-linked peptide connected lysine-141 of the succinate dehydrogenase cytochrome b560 subunit (SDHC) with lysine-261 of the succinate

dehydrogenase [ubiquinone] iron-sulfur subunit (SDHB), with a Cα-Cα distance of 10.9 Å (Fig. 4c). These results suggested that cross-linked sites occurred more frequently within the proximity of interaction interfaces.

## Protein structure and interactions information in the OXPHOS from native mitochondria

OXPHOS is composed of five enzymes (complex I–V: CI, NADH:ubiquinone oxidoreductase; CII, succinate:ubiquinone oxidoreductase; CIII, cytochrome $bc_1$ complex; CIV, cytochrome c oxidase, and CV, ATP synthase)[2]. CII, as the only enzyme dynamically involved in the OXPHOS system and TCA cycle, was analyzed in the TCA cycle. In addition to CII, a total of 124 cross-linked peptides were identified, belonging to 42 protein subunits of the remaining 4 enzymes (Fig. 5a and Supplementary Fig. 17).

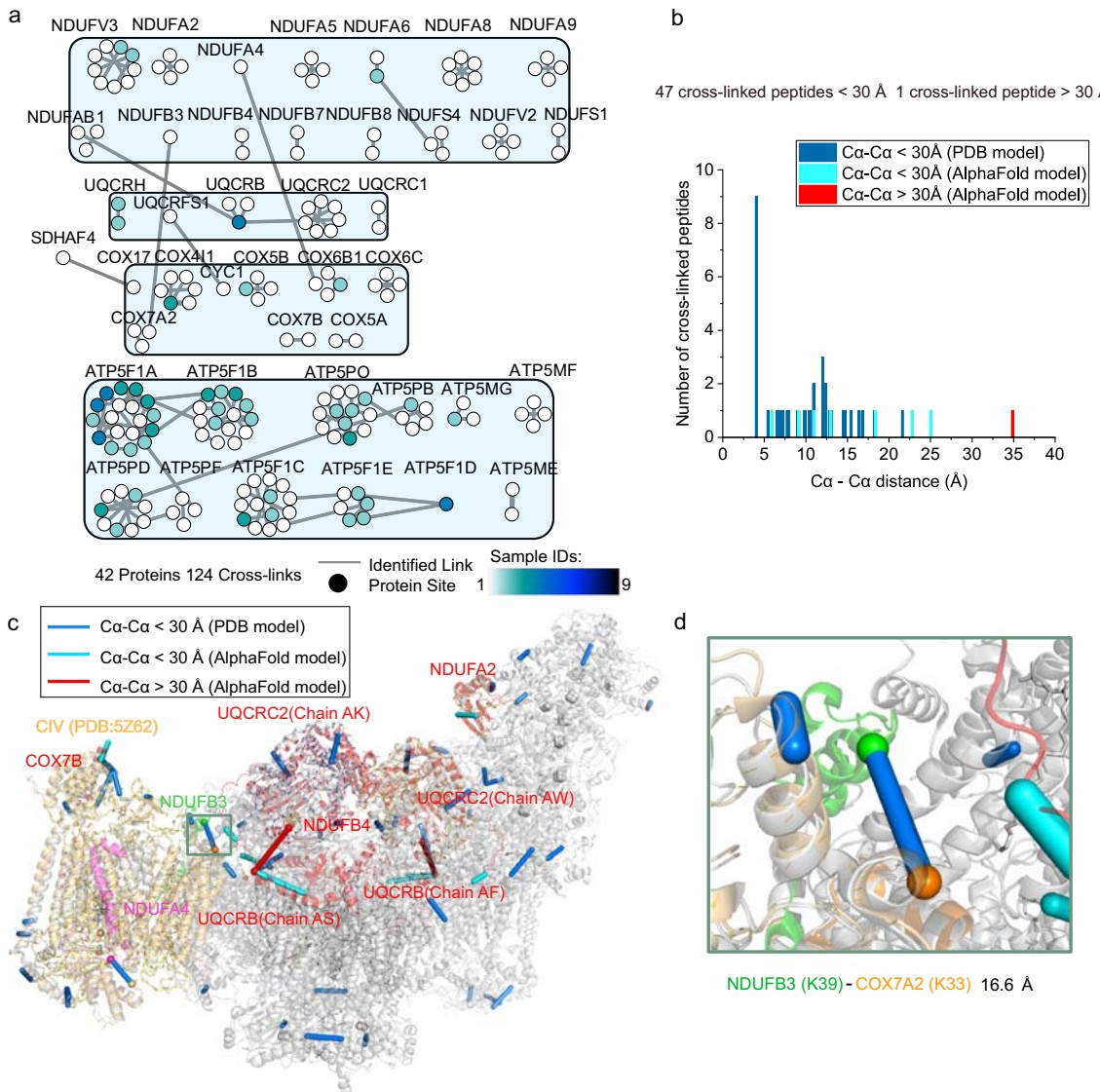

**Fig. 5 | Analysis of the cross-linked peptides of OXPHOS proteins identified in all fractionations. a** The cross-linked peptides involved in the OXPHOS proteins were selected from Supplementary Fig. 13a and displayed individually, where the nodes represent individual proteins, and the lines represent all cross-links identified between the two proteins. Each node was colored based on the number of samples in which each protein interaction was observed. **b** Histograms showing the length distribution of all distance restraints on the $SCI_1III_2IV_1$ structures in Fig. 5c after docking. **c** The detected cross-linked sites were mapped on the $SCI_1III_2IV_1$ structure, and cross-linked peptides were positioned on Cα-Cα links of subunit residues in the complex structure (PDB code: 5Z62 & 5XTH). The subunits of 5XTH were shown in grey, and the subunits of 5Z62 were shown in orange. The AlphaFold model aligned with the supercomplex ($SCI_1III_2IV_1$) was shown in red (details are available in Methods section) (including COX7B, NDUFB4, NDUFA2, UQCRC2, and UQCRB), where cross-linked peptides were positioned on Cα-Cα links of subunit residues of the AlphaFold model. **d** The structure of complex IV (PDB code: 5Z62) was aligned based on the $SCI_1III_2IV_1$ structure (PDB code: 5XTH) (RMSD = 0.775, 1292 to 1292 atoms), and the cross-linked Cα-Cα distance between the Lys-33 of COX7A2 and the Lys-39 of NDUFB3 was measured. The NDUFB3 of $SCI_1III_2IV_1$ was colored in green, and the COX7A2 of complex IV was colored in orange. In **b–d**, cross-linked peptides mapped on the structure within 30 Å were shown in blue (PDB model) or cyan (AlphaFold model), and cross-linked peptides mapped on the structure exceeded 30 Å were shown in red. Source data for (**b–d**) are provided as a Source Data file, and source data for (**a–d**) also are available in the PRIDE Archive with accession number: PXD035433.

In total, 53 cross-linked peptides were assigned to CI, CIII, and CIV. As previously described, CI, CIII, and CIV were dynamically assembled as stable supercomplex $I_1III_2IV_1$ ($SCI_1III_2IV_1$), playing important roles in energy conversion[2,38,39]. The organization and precise assignment of human $SCI_1III_2IV_1$ were well resolved in the cryo-EM structure, with the majority of residue from all the subunits of CI and CIII[2]. The crystal structure of the 13-subunit bovine homodimer CIV (PDB code: 5XTH) of human $SCI_1III_2IV_1$ was subsequently replaced by the 14-subunit human monomer CIV (PDB code: 5Z62), which added the NDUFA4 subunit and replaced the 3 subunits (e.g., Cox7A2 replaced Cox7A1)[40]. In our research, 39 of 53 cross-linked peptides were well mapped to the available homodimeric cryo-EM structure of $SCI_1III_2IV_1$

(PDB code: 5XTH) or monomeric CIV (PDB code: 5Z62), and their Cα-Cα distances were within 30 Å (Fig. 5b, c). Although the position of NDUFA4 in human $SCI_1III_2IV_1$ was easily disturbed by the environment[9], we still identified that the inter-cross-linked distance between the Lys-116 of COX6B1 and the Lys-73 of NDUFA4 in the CIV structure (PDB code: 5Z62) was 14.5 Å, which was consistent with previous studies (Supplementary Fig. 18).

For the remaining 14 of the 53 cross-linked peptides, the accuracy of five of these could not be determined because no experimental or predicted structural models were available, including five cross-linked peptides in NDUFV3. Therefore, the remaining 9 of the 14 cross-linked peptides were further explored. Among these, six cross-linked

peptides illustrated the unstructured N-terminal regions out of four subunits in human SCI₁III₂IV₁ (PDB code: 5XTH), which excellently matched with the predicted structure of the AlphaFold model (Fig. 5c). Two cross-linked peptides revealed protein interactions in the unresolved region of CIII (Lys-4 of UQCRB) in the structure of human SCI₁III₂IV₁. By aligning the AlphaFold model of UQCRB with the full-length residues for the position of UQCRB in SCI₁III₂IV₁, the cross-linked peptide revealed the interactions of Lys-88 of NDUFAB1 in CI and Lys-4 of UQCRB in CIII with a Cα-Cα distance of 25 Å. Another cross-linked peptide connected Lys-4 in UQCRB to Lys-92 in UQCRC2, and the Cα-Cα distance between the two cross-linked lysines was 35.1 Å. This extended distance potentially indicated that the cross-link was in the flexibility region of CIII (Fig. 5c).

Lastly, the interaction of the residue Lys-33 in chain 7A2 (Cox7A2) in CIV and the residue Lys-39 of NDUFB3 in CI was identified in the live HepG2 cells. It has been reported that Cox7A1 in the crystal structures SCI₁III₂IV₁ (PDB code: 5XTH) was replaced by human Cox7A2 in the human monomeric CIV structure (PDB code: 5Z62). The cryo-EM structure of human monomeric CIV (PDB code: 5Z62) was aligned based on the SCI₁III₂IV₁ structure (PDB code: 5XTH), and the measured cross-linked Cα-Cα distance was 16.6 Å (Fig. 5d). The cross-linked peptides provided distance constraint, demonstrating the molecular interactions between the subunit of complex I and the subunit of complex IV in SCI₁III₂IV₁. Thus, accurate protein subunit information could be obtained from the cross-linked proteins in the living cells.

In summary, 39 of the 53 cross-linked peptides all matched with the current structure of respirasome (SCI₁III₂IV₁), and 9 of 53 provided supplementary information for the cryo-EM structure of SCI₁III₂IV₁ (Fig. 5b, c).

Regarding the architecture of ATP synthase, approximately 99% of cross-linked peptides (70 of 71) were well mapped to the homologous bovine structure and the predicted structure of the AlphaFold model with a Cα-Cα distance of less than 30 Å. Among these, 3 of the 71 cross-linked peptides illustrated the unresolved regions of the bovine structure of ATP synthase, which also were well matched to the predicted structure of the AlphaFold model (Supplementary Fig. 19).

In addition, 23 inter-cross-linked peptides revealed the interactions of complex I–V with the other proteins, 12 of which were previously unreported (Supplementary Fig. 20). HSPD1 and HSPE1 were the chaperone proteins that repaired the protein disorders[41]. Two cross-links detected the interactions of HSPD1 with NDUFS5 and COX7C, and these cross-links indicated that HSPD1 could participate in the folding of SCI₁III₂IV₁ and play the role of an assembly factor in the assembly of SCI₁III₂IV₁.

### Structural information involving dynamic proteins from native mitochondria

The PPI sub-networks with dynamin-associated nuclear-encoded proteins were also revealed. The 27 cross-linked peptides identified belonged to 20 ancillary proteins, which were involved in the assembly of a series of intermediate assembly modules into complexes I–V (Supplementary Fig. 20)[3,39]. The structure of the ancillary proteins was not revealed by current technology, as they were dynamic and transient during the assembly of the complex and were not presented in the mature complex. The majority of the ~93% (25/27) cross-linked peptides were mapped on the predicted AlphaFold model with a Cα-Cα distance of less than 30 Å, and there were 2 important proteins among the above-mentioned proteins. One was COX7A2L, which played an important role in the assembly of SCI₁III₂IV₁, and the other was 6.8-kDa mitochondrial proteolipid protein (ATP5MPL), which was loosely linked to ATP synthase and regulated its function[42].

Furthermore, the detected cross-linked peptide pairs provided site connection information for the other members of the SLC25 family, such as aspartate/glutamate carriers aralar (SLC25A12) and citrin (SLC25A13), as well as glutamate (SLC25A18, SLC25A22) and

mitochondrial citrate carriers (SLC25A1) (Supplementary Fig. 21a)[43,44]. For the identified SLC25 family proteins, 53 cross-linked peptides were identified, belonging to 16 proteins. Among these, most of the protein sites involved in the cross-linked peptides were in the unresolved structural regions of the protein; thus, the cross-linked peptides were mapped to the predicted AlphaFold model. Approximately 96% of the cross-linked peptides (51 of 53) were well mapped to the AlphaFold model and the structure in the PDB with a Cα-Cα distance of less than 30 Å (Supplementary Fig. 21b, c). Four different isoforms of the human mitochondrial ADP/ATP carriers (SLC25A4, SLC25A5, SLC25A6, and SLC25A31) were also analyzed, which performed the function of transporting ADP from the cytoplasm to the mitochondrial matrix for ATP synthesis and provided fuel for the cells through ATP transport (Supplementary Fig. 21b)[45]. The human structural information of the ADP/ATP carrier was not resolved in the existing experimental results. Therefore, the detected cross-linked peptides were mapped on the crystal structure of malonamidase E2 from *Bradyrhizobium japonicum*, and the measured distance between Cα matched well with the reported structure (within 30 Å). In addition, an intermolecular cross-linked peptide revealed Lys-235 self-linked in the citrin N-terminal domain, further providing a site for the N-terminal domain of citrin to mediate dimerization of the human full-length aspartate/glutamate carrier[46]. Only one intramolecular crosslink (K74, K243) in SLC25A24 was incompatible with the predicted AlphaFold model. Although the position of the connecting residues was predicted with high confidence, two fragments in the AlphaFold model of SLC25A24 (residues 167–190 and 282–289) had low confidence with a prediction score of less than 70 in the PLDDT. We inferred that the incompatible cross-links were possibly caused by the flexible region of the protein, and the cross-links potentially provided a distance-constrained reference between these two residues.

Members of the mitochondrial transporter family SLC25 are typically located in the mitochondrial inner membrane and perform the vital role of shuttling amino acids, carboxylic acids, inorganic ions, and cofactors between the mitochondrial matrix and the cytosol[44,47]. Although affected by protein expression and purification, the structures of several SLC25 family members have yet to be characterized, but the detected protein cross-linked site information can provide complementary information for the structures of SLC25 family members. These results also demonstrated the advantage of this method in capturing the dynamic carrier protein structures, which provided cross-linked site information for the structures of the SLC25 family proteins in cellular metabolism.

### Application of the CD-MS method to MS-cleavable cross-linkers

To explore whether the CD-MS method could deliver MS-cleavable cross-linkers, DDAB@PLGA/Kolliphor EL NPs were used to re-load three MS-cleavable cross-linkers, including DSSO and DSBU, as well as custom-made DSBSO. According to the predicted n-octanol-water partition coefficient (log Po/w) of the cross-linker[48] (Supplementary Fig. 22a) and the elution time of the cross-linker on the C18 column (Supplementary Fig. 22b), we found three MS-cleavable cross-linkers that were more hydrophilic than DSS, with a hydrophobicity order of DSSO < DSBU < DSBSO < DSS. During the preparation process of the NPs, we found that both DSSO and DSBU were only soluble in a mixture of Kolliphor EL/dichloromethane and methanol, but not completely soluble in Kolliphor EL, which differed from the oil phase (Kolliphor EL/ dichloromethane) of the initial parameter. Finally, we did not detect the embedded MS-cleavable cross-linkers in the NPs using high-performance liquid chromatography (HPLC).

Given that the above MS-cleavable cross-linkers were not very soluble in the oily core phase, we deduced that the hydrophilic MS-cleavable cross-linker precipitated in the outer interphase (oil/water), rather than in the inner interphase (Kolliphor EL/shell). In this manner, as the solvent evaporated, coarse emulsion droplet phase separation

with an oily core phase (Kolliphor EL) formed the oil core, and a polymer-rich phase (PLGA/DDAB) formed the shell. The MS-cleavable cross-linkers incompatible with the oil core tended to form a polymer-rich phase, which precipitated on the oily core, forming a shell in the outer interphase (oil/water). Thus, the NPs did not protect the MS-cleavable cross-linkers from hydrolysis. Overall, the DDAB@PLGA/Kolliphor EL NPs could not efficiently deliver these three MS-cleavable cross-linkers, which were more hydrophilic than DSS. In addition, we found that the key factor for the NPs to deliver the cross-linker was the oil core property, which dissolved the cross-linker and encapsulated the cross-linker into the NPs to avoid hydrolysis of the cross-linker.

## Discussion

A CD-MS method was developed for mapping protein conformations and the interactions of mitochondria in living cells. The DDAB-functionalized PLGA NPs exhibited a high DSS-loading efficiency (23.21%, w/w) (Fig. 2c). The NPs with the highest loading efficiency (70DSS-DDAB@PLGA/Kolliphor EL NPs) showed good cellular biocompatibility during the 6 h delivery without causing significant changes at the proteomic level (Supplementary Fig. 2). These results confirmed that the NPs could load DSS with a high loading efficiency, offering a noninvasive and biocompatible method for capturing the mitochondrial interactome in living cells. Subsequently, the CLSM/Revolution XD results further confirmed that NPs specifically targeted the mitochondria after endocytosis (Fig. 2d), instead of the endoplasmic reticulum (PCC was 0.44 (Supplementary Fig. 8b)) or Golgi apparatus (PCC was 0.18 (Supplementary Fig. 8c)). In addition, co-localization of the DSS-FITC-DDAB@PLGA/Kolliphor EL NPs and mitochondria, as measured using CLSM, confirmed the successful delivery of the encapsulated DSS via the NPs to the mitochondria of the HepG2 cells (Fig. 2e). Furthermore, the Tom 20 protein detected using the CD-MS method provided evidence that the DSS-DDAB@PLGA/Kolliphor EL NPs in the cytoplasm were transported by TOM into the mitochondrial matrix (Supplementary Fig. 12d).

Without the need for modifying cells or isolating organelles, this method was an attempt at the delivery of a cross-linker to the mitochondria in a natural cellular environment. In 30 fractionations, 1245 cross-links were identified from the mitochondrial proteins in the HepG2 cells using the CD-MS method (Supplementary Fig. 13a). These data provided an in vivo mitochondrial interaction map composed of 503 proteins corresponding to 44% of the human mitochondrial proteome, providing broader information on the PPIs and protein structures. Among these, 51.32% of the PPIs were annotated in the STRING database, and half of these PPIs had high confidence ( > 0.7) (Supplementary Fig. 13c). Among these, PPI linked OGDHL and PDHA1 filled the gap due to the lack of relevant experimental evidence in the STRING database (Fig. 4c). For the non-annotated PPIs, GOBP analysis showed that these proteins were involved in the dynamics and transients of biological processes (Fig. 3b).

Our data also provided structural evidence for TCA, OXPHOS complexes, and dynamic proteins in living cells. Under the condition of intracellular mitochondrial cross-linking, the NPs released the cross-linker in situ, with less likelihood of significantly altering the protein structure than isolating the mitochondria from the cells. This anticipation was supported by the fact that most cross-linked peptides in the TCA enzymes, OXPHOS complexes, and dynamic proteins in our research ( ~ 94%, 257 of the 272 cross-linked peptides) were highly consistent with the analysis of structural models in the PDB database, or the unresolved structure in the predicted AlphaFold model. Therefore, the CD-MS method could provide insight into the interactions of proteins with unknown structures (e.g., dynamic proteins). In addition, our data provided additional evidence for the structure of human $SCI_1III_2IV_1$. For example, the cross-linked peptides revealed that the interactions between COX7A2 and NDUFB3 provided complementary information for the cryo-EM structure of $SCI_1III_2IV_1$ (Fig. 5d).

To explore the range of the NP delivery cross-linker, we investigated the delivery of MS-cleavable cross-linkers using the DDAB@PLGA/Kolliphor EL NPs. After fabricating the NPs, we found that the DDAB@PLGA/Kolliphor EL NPs were incompatible with cross-linkers that were more hydrophilic than DSS. This clearly showed that the key factor for the NPs to deliver the cross-linker was the oil core property, to dissolve the cross-linker and encapsulate the cross-linker in the NPs to avoid hydrolysis of the cross-linker. In addition, inspired by a variety of successful cases of functional PLGA-based NPs for subcellular targeted therapy (e.g., ER-targeting peptides (KKXX) modified PLGA NPs[49] and nuclear localization sequence (NLS) modified PLGA NPs[50]), we determined that targeting specific subcellular proteins by integrating functional groups onto the specific NP surfaces had the potential for the in situ cross-linking of other important subcellular proteins. The CD-MS method for further improvement was foreseeable in both expanding the types of cross-linkers and targeting the other subcellular orange, such as the delivery of MS-cleavable cross-linkers and functional NPs for additional subcellar targeting (for example, nuclear and endoplasmic reticulum).

In summary, this CD-MS method offers a technology platform for studies toward understanding protein conformations and interactions in the mitochondria of HepG2 cells. Certainly, this method can be further improved for controlling cross-linker release kinetics, increasing cross-linked coverage, and developing specific ligands to target different organelles, which will be investigated in our future studies.

## Methods

### Fabrication and characterization of the DSS-DDAB@PLGA/Kolliphor EL NPs

Inspired by the targeted NP delivery concept[51], the NPs (DDAB@PLGA/Kolliphor EL NPs loaded with DSS (Sigma-Aldrich, USA): DSS-DDAB@PLGA/Kolliphor EL NPs) were synthesized using the solvent-evaporation method. We dissolved PLGA (112 mg) (9 kDa (50:50), Jinan Daigang Biomaterial, China) and DDAB (10 mg) (Sigma-Aldrich, USA) in 4.5 mL of dichloromethane, and 50 μL of Kolliphor® EL (Sigma-Aldrich, USA), DDAB (2.5 mg), and DSS (10, 40, and 70 mg) were dissolved in 0.5 ml of dichloromethane. Then, the two were mixed into an organic phase. We subsequently poured the organic phase into the 50 mL water phase (1% PVA(w/v)) (Sigma-Aldrich, USA)) for ultrasonic treatment (time = 2 min, power = 120 W, and interval time = 4 s) to form a coarse emulsion. Then we transferred the solution to 150 mL of water phase (1% PVA(w/v)), which was magnetically stirred for 15 h to volatilize, and the organic solvent solidified to form the DSS-DDAB@PLGA/Kolliphor EL NPs. According to the above-mentioned solvent-evaporation method, different amounts of DSS (10, 40, and 70 mg) were mixed with PLGA, DDAB, and Kolliphor EL to fabricate NPs with an adjustable loading efficiency, and they were denoted according to the DSS amount (10DSS-DDAB@PLGA/Kolliphor EL NPs, 40DSS-DDAB@PLGA/Kolliphor EL NPs and 70DSS-DDAB@PLGA/Kolliphor EL NPs).

Different amounts of DDAB were used in NP fabrication to examine how the proteins in live HepG2 cells were influenced by the DDAB parameters, where the DDAB content in the 70DSS-DDAB@PLGA/Kolliphor EL NPs was changed to 80% (Treated-2), 60% (Treated-3), 40% (Treated-4) and 20% (Treated-5) of the initial content (Treated-1). The 70DSS-DDAB@PLGA/Kolliphor EL NPs were prepared according to the solvent-evaporation method described above. We dissolved PLGA (112 mg, 9 kDa) and DDAB (10, 8, 6, 4, and 2 mg) in 4.5 mL of dichloromethane and 50 μL of Kolliphor® EL, while DDAB (2.5, 2, 1.5, 1, and 0.5 mg) and DSS (70 mg) were dissolved in 0.5 mL of dichloromethane, and then the two were mixed into an organic phase. The preparation process of different 70DSS-DDAB@PLGA/Kolliphor EL NPs was the same as above.

The size, PDI, and zeta potential of the NPs were determined using a Zetasizer Nano-ZS instrument (Malvern Instruments, UK) with three

independent experiments, and each measurement was run three times using automated, optimal, measurement times and laser attenuation settings. The loading efficiency was measured using a HITACHI Chromaster high-performance liquid chromatography (HPLC, Japan) instrument with three independent experiments, and each measurement was run three times. The prepared NPs were dissolved in ACN and then centrifuged to obtain the supernatant for examination. The supernatant was loaded onto an XBP C18 column (5 μm, 100 Å, 4.6 × 250 mm i.d.) operated by the HPLC system at a flow rate of 1 mL/min and recorded at a UV wavelength of 200 nm. The morphologies of the prepared NPs (the highest loading efficiency of NPs: 70DSS-DDAB@PLGA/Kolliphor EL NPs) were characterized using transmission electron microscopy (TEM, JEM-2000EX, Japan). The TEM samples were stained with 1% phosphotungstic acid before they were dropped onto a copper net. The cumulative release of DSS was measured using a HITACHI Chromaster HPLC (Japan) instrument. The prepared NPs were placed in a 12, 000 Da dialysis bag. The PBS (0.01 M, pH = 7.2) solution (12 mL) was prepared with 1% DMSO (v/v) as a sustained-release solution environment (1% DMSO in PBS was employed to dissolve DSS) and magnetically stirred. The supernatant was used to detect the released DSS content using HPLC at intervals of 30 min, and 10 mL of the corresponding PBS solution in the centrifuge tube was replaced. The supernatant was loaded onto an XBP C18 column (5 μm, 120 Å, 4.6 × 250 mm i.d.) operated by the HPLC system at a flow rate of 1 mL/min and recorded at a UV wavelength of 200 nm. The loading efficiency was calculated as Eq. 1:

$$\text{Loading efficiency}(\%) = \frac{W\,DSS}{W\,total} \times 100\%, \qquad (1)$$

where $W_{DSS}$ is the weight of DSS in the NPs and $W_{total}$ is the total weight of DSS, PLGA, DDAB, and Kolliphor EL.

The cumulative release of DSS was calculated according to Eq. 2:

$$\text{Cumulative release of DSS}(\%) = \frac{W\,DSS}{W\,total\,DSS} \times 100\%, \qquad (2)$$

where $W_{DSS}$ is the weight of DSS, and $W_{total\,DSS}$ is the total weight of DSS in the NPs.

## SDS-PAGE

To assess whether the dissolved DSS in Kolliphoer EL still exhibited reactivity, a positive-control sample was generated by reacting 100 μL of BSA at 3 mg/mL or 0.3 mg/mL in PBS (0.01 M, pH = 7.2) with 0.6 mM DSS cross-linker (12 mM DSS was dissolved initially in DMSO) for 30 min at 37 °C. Then the test of the experimental group was conducted under the same condition with 0.6 mM DSS cross-linker (12 mM DSS was dissolved initially in Kolliphoer EL). The reactions were terminated by adding ammonium bicarbonate to a final concentration of 50 mM (1 h at 37 °C). The cross-linked BSA was analyzed using SDS-PAGE in 12% polyacrylamide gradient gels, and the proteins were visualized by staining with Coomassie Blue dye and then photographed by BIO-RAD (USA).

## Synthesis of DSS functionalized with FITC (DSS-FITC)

Compound 1 (Tris-succinimide ester) was obtained from ref. 17. Compound 1 (65.74 mg, 0.1257 mmol) and propargyl amine (25.39 mg, 0.2514 mmol) (Sigma-Aldrich, USA) were dissolved in 10 mL of DMSO, and Ltd.[4'-(aminomethy) fluorescein] (24.98 mg, 0.0628 mmol) (AAT Bioquest, USA) dissolved in 1 mL of DMSO was added dropwise to the solution (Supplementary Fig. 3). After stirring at room temperature for 1 min, the mixture was purified using semi-preparative RP-HPLC (A: H₂O and B: ACN; the gradient method was set as follows: B from 30% to 70% over 30 min at a flow rate of 50 mL/min monitored at a UV wavelength of 200 nm) with a C18 column (10 μm, 100 Å, 50 × 250 mm

i.d.). The product-containing fraction (retention time, 24.0–28.0 min) was lyophilized at 30 °C for 24 h to afford the targeted DSS functionalized with FITC ($C_{39}H_{35}N_3O_{14}$) as a light-yellow powder (8.00 mg, 0.01039 mmol, yield of 16.7%). The $^1$H-NMR spectra were recorded on a Bruker AVANCE II 400 MHz spectrometer using DMSO-d6 as the solvent and TMS as the internal standard (Supplementary Fig. 4): $^1$H-NMR (400 MHz, DMSO-d6, ppm), δ 10.4 (s, 1H), 10.1 (s, 1H), 8.4 (t, 1H, J = 5.0 Hz), 8.0 (d, 1H, J = 7.8 Hz), 7.8–7.7 (m, 2H), 7.2 (d, 1H, J = 7.8 Hz), 6.8 (s, 1H), 6.7–6.4 (m, 4H), 4.5 (d, 2H, J = 5.0 Hz), 2.8 (s, 8H), 2.2 (t, 2H, J = 7.5 Hz), 1.7–1.2 (m, 6H), and 1.2 (m, 1H). The mass spectra were recorded on an LTQ Orbitrap Velos (Thermo Fisher Scientific, USA). The exact mass 769.72 for $C_{39}H_{35}N_3O_{14}$, was 770.2436 $[M + H]^+$ (Supplementary Fig. 5).

## Preparation of the fluorescently labeled PLGA/Kolliphor EL NPs

For the experimental group, the cy5-labeled DDAB@PLGA/Kolliphor EL NPs were synthesized using the solvent-evaporation method. PLGA (9 kDa: 28 mg), DDAB (3.12 mg), Kolliphor® EL (12.5 μL) and Cy5 (50 μg) (Fanbo Biochemicals, China) were dissolved in 1.25 mL of dichloromethane. Then we poured the organic phase into 12.5 mL of the water phase (1% PVA(w/v)) for ultrasonic treatment (time = 2 min, power = 120 W, and interval time = 4 s) to form a coarse emulsion, which was then transferred to 37.5 mL of water phase (1% PVA(w/v)) and magnetically stirred for 15 h to volatilize. Then the organic solvent was solidified to form the cy5-labeled DDAB@PLGA/Kolliphor EL NPs.

The Cy5-labeled DSS-FITC-DDAB@PLGA/Kolliphor EL NPs were synthesized using the solvent-evaporation method. PLGA (9 kDa: 28 mg), DDAB (3.12 mg), Kolliphor® EL (12.5 μL), and DSS-FITC (0.5 mg) were dissolved in 1.25 mL of dichloromethane. The preparation process of the DSS-FITC-DDAB@PLGA/Kolliphor EL NPs was the same as above. The labeling efficiency of the NPs with fluorescent reagents was determined using an SH800S cell sorter (Sony Biotechnology).

## Cell culture

HepG2 human liver carcinoma cells line Pbabe.puro (American Type Culture Collection (ATCC), catalog No. HB-8065) were cultured in Dulbecco's modified Eagle's medium (DMEM, Gibco, USA), which was supplemented with 10% fetal bovine serum (FBS, Gibco, USA) and 1% penicillin-streptomycin (Gibco, USA) in 5% CO₂ at 37 °C in a humidified incubator. The HepG2 cells were seeded in 1 cm microscope coverslips at a density of $1 \times 10^4$ cells in the CLSM experiment. HepG2 cells were seeded in a 15 cm Petri dish and grown to 90% density for cross-linking experiments.

## Evaluation of the internalization and intracellular localization of the DDAB@PLGA/Kolliphor EL NPs and DSS

a. Observation of the intracellular NP distribution using CLSM
For the experimental group, the HepG2 cells were incubated with cy5-labeled DDAB@PLGA/Kolliphor EL NPs (0.05 mg/mL) for 6 h at 37 °C in 5% CO₂. For the control group, the HepG2 cells were incubated with cy5-labeled PLGA/Kolliphor EL NPs (0.05 mg/mL) for 6 h at 37 °C in 5% CO₂. The cells were stained with MitoTracker Green (200 nM), LysoTracker Red (500 nM), ER-Tracker Green (1 μM), Golgi-GFP (35 μL/mL) at 37 °C for a suitable time, respectively. The aforementioned dye reagents were purchased from Thermo Fisher Scientific. Then, the cells were washed five times with PBS (0.01 M, pH = 7.2). Subsequently, for the experimental group, two groups of the cell images were collected using laser scanning confocal microscopy (Nikon), including Mitotracker Green (486.4 nm) and Cy5 (638.2 nm) channels, and Lysotracker Red (561.4 nm) and Cy5 (638.2 nm) channels. Image analysis (PCC) was performed using NIS-Elements AR. Four groups of cell images were collected using Andor live cell confocal imaging platform (Revolution WD), including ER-Tracker Green

(488 nm) and Cy5 (640 nm) channels, Golgi-GFP (488 nm) and Cy5 (640 nm) channels, and Mitotracker Green (488 nm) and Cy5 (640 nm) channels in experimental and control groups. Image analysis (PCC) was performed using ImageJ software.

b. Observation of the intracellular distribution of DSS delivered by the NPs

The HepG2 cells were incubated with DSS-FITC-DDAB@PLGA/ Kolliphor EL NPs (0.05 mg/mL) for 6 h at 37 °C. Mito Tracker Deep Red (200 nM) was added and incubated for 30 min at 37 °C. The medium was removed, and the cells were fixed with 4% formaldehyde (v/v) for 10 min. The cell images were collected in the FITC (486.4 nm) and Cy5 (640 nm) channels using laser scanning confocal microscopy (Nikon). Image analysis (PCC) was performed using NIS-Elements AR.

c. Observation of the intracellular co-localization of the NPs/ lysosomes at different times

The HepG2 cells were incubated with cy5-labeled DDAB@PLGA/ Kolliphor EL NPs (0.05 mg/mL) in different intervals at 37 °C in 5% CO2. After treatment with DDAB@PLGA/Kolliphor EL NPs for 2 h, 4 h, 6 h, and 8 h, the cells were stained with Lyso Tracker (500 nM) at 37 °C for 30 min. The images were collected using laser scanning confocal microscopy (Nikon) at the channel for FITC (486.4 nm) and LysoTracker Red (561.4 nm). Image analysis (PCC) was performed using NIS-Elements AR.

## Extraction of the cytosol and lysosomes from the HepG2 cells

For lysosome isolation, a Lysosome Isolation Kit (Sigma-Aldrich, USA) was used according to the manufacturer's instructions and a study by Cai et al.[52] After they were cultured until 90% confluency, the HepG2 cells were digested with trypsin, and the cells were collected through centrifugation at 500 x g for 5 min. The HepG2 cells ($1 \times 10^8$) were then washed three times with precooled PBS (0.01 M, pH = 7.2), and the cells were collected through centrifugation at 500 x g for 5 min at 4 °C. Subsequently, the cells were resuspended in precooled 1 x extraction buffer and further broken with a Dounce homogenizer. The cells achieved 80% breakage, which was confirmed by staining with 0.4% trypan blue and observation under a microscope. The cell lysates were centrifuged at 1000 x g for 10 min at 4 °C to remove the cell debris and nuclei. Subsequently, the supernatant was collected and further centrifuged at 10,000 x g for 10 min to remove the small cell debris. Then, the supernatant was collected for centrifugation at 20,000 x g for 20 min at 4 °C to separate the cytoplasm and subcellular organelles. The final supernatant was defined as the cytosol extract, and the remaining pellets were collected in a minimal volume of 1 x extraction buffer, which was the crude lysosomal fraction (CLF) containing mainly lysosomes.

To further enrich the lysosome in the CLF, the CLF was diluted to a solution containing 19% OptiPrep™ Density Gradient Medium solution with a protein concentration of 0.5 mg-protein/mL and defined as the Diluted OptiPrep™ Fraction (DOF). The lysosomes were then further purified by adding calcium chloride to the DOF to a final concentration of 8 mM. The mixture was incubated on ice for 15 min followed by centrifugation at 5000 x g for 15 min to precipitate the rough endoplasmic reticulum and any mitochondria. The purified lysosomes in the supernatant were collected and stored at −80 °C for further use. The full procedure is summarized in Supplementary Fig. 10a. The BCA assay (BCA Protein Assay Reagent, Beyotime, China) was performed according to the manufacturer's instructions.

## Preparation of the CDNP^D, CDNP^DL, and CDNP^DLC: CDNP incubation in DMEM medium and lysosomal and cytosolic fluids

This protocol referred to the study reported by Cai et al.[52] The particle concentration was maintained at 1.8 mg/mL. 70DSS-DDAB@PLGA/ Kolliphor EL NP suspension was added dropwise into the Dulbecco's Modified Medium (DMEM, Gibco, USA) and supplemented with 10% fetal bovine serum (FBS, Gibco, USA), 1% penicillin-streptomycin

(Gibco, USA), and incubated for 1 h at 37 °C with 5% CO2. The samples were centrifuged at 16,000 x g for 10 min at 4 °C to form a pellet of the CDNP@DMEM (CDNP^D) complexes. The pellet was resuspended in buffer A (103.5 mM NaCl, 5.3 mM KCl, 5.6 mM Na2HPO4, and 1.4 mM KH2PO4, and 23.8 mM NaHCO3, pH = 7.4), transferred to a vial, and then centrifuged at 16,000 x g for 10 min at 4 °C again to wash off the unbound proteins. This procedure was repeated three times to obtain the CDNP^D. To mimic the pathways of NP internalization into the cells, the lysosomes and cytosol were incubated sequentially with the CDNP^D complexes. Specifically, to mimic the endocytosis process, the CDNP^D complexes were resuspended in lysosomal solution (pH = 4.7) at 37 °C for 1 h to form the CDNP@DMEM@Lysosomal (CDNP^DL) complexes. Pure CDNP^DL were obtained after centrifugation (16,000 x g, 4 °C, 10 min) and washed three times with PBS (0.01 M, pH = 7.2). The obtained CDNP^DL complexes were further incubated with cytosol at 37 °C for 1 h to form the CDNP@DMEM@Lysosomal@Cytosol (CDNP^DLC) complexes. Again, the CDNP^DLC were purified through centrifugation at 16,000 x g for 10 min at 4 °C and washed three times as described above. The CDNP^D, CDNP^DL, and CDNP^DLC complexes were resuspended in cold PBS (0.01 M, pH = 7.2) at 4 °C until further use. These procedures are illustrated in Supplementary Fig. 10b.

## Sample preparation of the adsorbed proteins on the NPs

The proteins were eluted from the CDNP by pooling into 7-fold excess acetone followed by precipitation at −20 °C overnight and washed once more with cold acetone to completely remove the PLGA/DDAB/ Kolliphor EL. After resuspension in 1 mL of 8 M urea, the sample was reduced with 10 mM tris(2-carboxyethyl)phosphine (Sigma-Aldrich, USA) for 1 h at 37 °C, followed by alkylation with 20 mM iodoacetamide (Sigma-Aldrich, USA) for 30 min at room temperature in the dark. The sample was diluted to 1 M urea with 50 mM ammonium bicarbonate and digested with trypsin at an enzyme-to-protein ratio of 1:50 (w/w) at 37 °C for 12 h. The peptides were desalted using the SepPak C18 column (Waters, USA).

## CD-MS methods for cross-linking

The 20 mL fresh medium containing 1.8 mg/mL of the highest loading efficiency NPs (70DSS-DDAB@PLGA/Kolliphor EL NPs) was added to a 15 cm Petri dish with HepG2 cells and incubated for 6 h at 37 °C in 5% CO2. The dish with HepG2 cells was washed three times with precooled PBS (0.01 M, pH = 7.2), and the cells were collected through centrifugation at 500 x g for 5 min. The cells were then lysed, and the proteins were extracted in lysis buffer (10% C12Im-Cl and 1% protease inhibitor cocktail), followed by trypsinization on a 10 kDa filter. The digested peptides were then fractionated using RPLC.

## Cross-linking experiment

(1) In vivo methods for cross-linking (Control 1 for CD-MS)

The dish with HepG2 cells (approximately 2 mg of protein) was washed three times with precooled PBS (0.01 M, pH = 7.2), and the cells were collected through centrifugation at 500 x g for 5 min. The cells were cross-linked with 0.225 mM DSS[10], with a protein-to-cross-linker ratio of 12:1 in 2 mL of cross-linking reaction, and then quenched with 50 mM ammonium bicarbonate for 15 min.

(2) In vitro mitochondrial methods for cross-linking (Control 2 for CD-MS)

A dish with HepG2 cells was washed three times with precooled PBS (0.01 M, pH = 7.2), and the cells were collected through centrifugation at 500 x g for 5 min. Cell lysis and mitochondria preparation were performed according to a protocol adapted from the study of Ryl, Clayton and Shadel[10,53]. Cell lysis was conducted in 5.5 mL of ice-cold RSB hypotonic buffer [10 mM N-(2-hydroxyethyl) piperazine-N′-ethanesulfonic acid (HEPES) pH =

7.5, 10 mM NaCl, 1.5 mM MgCl$_2$] using Dounce homogenization. Subsequently, 4 mL of ice-cold 2.5 x MS homogenization buffer [12.5 mM HEPES pH 7.5, 525 mM mannitol, 175 mM sucrose, and 2.5 mM ethylenediaminetetraacetic acid (EDTA)] was added to obtain an isotonic solution. The cell lysates were centrifuged three times at 1300 x g for 5 min at 4 °C to remove the cell debris and nuclei. The mitochondria were pelleted through centrifugation at 12,360 x g (15 min, 4 °C) and washed once with 5 mL of ice-cold 1 x MS homogenization buffer. The isolated mitochondria were washed twice in ice-cold PBS (0.01 M, pH = 7.2) and pelleted at 16,000 x g (15 min at 4 °C). The isolated mitochondria (0.22 mg) were cross-linked with 0.225 mM DSS[10], with a protein to cross-linker ratio of 12:1 at a protein concentration of 1 mg/mL in 220 μL of cross-linking reaction (37 °C, 30 min). The above chemical reaction was quenched with 50 mM ammonium bicarbonate for 15 min.

### Sample preparation of the cross-linked proteins using i-FASP

The i-FASP protocol[54] for sample preparation was chosen. The HepG2 cells were lysed in lysis buffer (10% C12Im-Cl and 1% protease inhibitor cocktail) through ultrasonication for 60 s at 60 W (5 s on, 15 s off), followed by centrifugation at 16,000 x g (20 min, 4 °C). The supernatant protein concentration was measured using the bicinchoninic acid assay. The proteins were reacted with 100 mM DL-Dithiothreitol (Sigma-Aldrich, USA) at 95 °C for 5 min for denaturation and reduction. Each 150 μg of protein extract was transferred to a 10 kDa filter (Sartorius, Germany). After centrifugation at 14,000 x g for 10 min at 25 °C, the concentrate was diluted in the device with 200 μL of 50 mM ammonium bicarbonate and centrifuged at 14,000 x g at 20 °C for 25 min. Subsequently, 20 mM iodoacetamide was added for alkylation at room temperature for 30 min in the dark. The proteins in the filter were washed three times with 50 mM ammonium bicarbonate followed by centrifugation at 15,000 x g to remove the C12Im-Cl, DL-Dithiothreitol, and unreacted iodoacetamide. Afterward, the proteins were digested with trypsin at an enzyme-to-protein ratio of 1:25 (w/w) at 37 °C for 15 h. Then, the generated peptides were collected through centrifugation at 15,000 x g for 25 min at 25 °C, followed by washing the filter with 10 mM ammonium bicarbonate twice. The collected peptides were stored at −80 °C for further MS analysis.

### Evaluation of the effect of CD-MS labeling mitochondrial protein

The effect of the CD-MS method for labeling mitochondrial proteins was also compared with those of the in vivo XL-MS method and in vitro mitochondrial XL-MS method in the three repeated single MS experiments. First, the ratio of labeling mitochondrial peptides in all detected peptides (α) was individually calculated among the three methods. Because there was no additional enrichment or targeting of the subcellular organelles, the α of the in vivo XL-MS method was set as the baseline for the other two groups. The α of the CD-MS method was improved by 4.11-fold, while that of the in vitro mitochondrial XL-MS method was improved by 1.84-fold (Supplementary Fig. 12a).

Second, all cross-linked peptides of the mitochondrial proteins (including cross-linked, loop-linked, and mono-linked peptides) among the three methods were counted and displayed in the protein attribution database (Supplementary Fig. 12b).

Third, according to the sub-mitochondrial localization of the proteins[28], the protein structural information and protein-protein interactions (PPIs) provided by the intra- and inter-molecule cross-linked peptides were mapped (Supplementary Fig. 12d)[55]. CD-MS could capture more mitochondrial spatial interactions in all compartments of the mitochondria. The CD-MS method provided cross-linked site information for 48 proteins including nine pairs of inter-protein interactions, 66.7% of which (6/9) were annotated in the PPI databases (STRING databases, Supplementary Fig. 12c). Similarly, the in vitro mitochondrial XL-MS provided cross-linked information for 66

proteins including 17 pairs of inter-protein interactions (64.7% (11/17) were annotated in STRING databases). In vivo XL-MS provided cross-linking information for 44 proteins including eight pairs of inter-protein interactions (100% (8/8) were annotated in STRING databases).

### Fractionation of cross-linked peptides using high-pH reversed-phase chromatography

The tryptic peptides (400 μg) were fractionated using high-pH RPLC. The workflow was as follows. The peptides were resuspended in mobile phase A [98% H$_2$O + 2% ACN, pH 10]. The peptide samples were loaded onto a Durshell C18 column (5 μm, 100 Å, 4.6 × 200 mm i.d.) and fractionated by an Agilent Technologies HPLC system at 1 mL/min using a 70-min gradient of increasing mobile phase B (2% H$_2$O + 98% ACN, pH = 10). The separation gradient was achieved by applying 0% B for 10 min, 0–2% B for 1 s, 2–30% B for 35 min, 30–45% B for 15 min, and 45–90% B for 10 min. The fractionations were collected every 1 min starting at 10 min and merged into 30 fractions (fraction 1 + 31, fraction 2 + 32 …). Each fraction was measured in technical triplicate using LC-MS on the QE mass spectrometer.

### LC-MS/MS analysis

An Easy-nLC 1000 system coupled to a Q-Exactive mass spectrometer (Thermo Fisher Scientific, USA) was used. Q-Exactive Tune Application (2.8 SP1 Build 2806) and Thermo Scientific, Xcalibur (v3.1.66.10) were used for the control of mass spectrometer and data collection. The samples were automatically loaded onto a C18 RP trap column (150 μm i.d. × 3 cm) and separated by a C18 capillary column (150 μm i.d. × 15 cm), then packed in-house with ReproSil-Pur C18-AQ particles (1.9 μm, 120 Å) with low-pH mobile phases (buffer A: 98% H$_2$O + 2% ACN + 0.1% FA; buffer B: 2% H$_2$O + 98% ACN + 0.1% FA). For unfractionated cross-linked peptides, the separation gradient was achieved by applying 3–12% B for 65 min, 12–20% B for 65 min, 20–40% B for 5 min, 40–80% B for 1 min, and 80% B for 15 min. The method parameters of the run were as follows: data-dependent acquisition; full MS resolution 70,000 at m/z 200; scan range of 300–1800; MS1 AGC target 3e6; MS1 Maximum OT 80 ms; MS/MS resolution 17,500 at m/z 200; fixed first mass 110 m/z; MS/MS AGC target 1e5; MS/MS maximum OT of 60 ms, loop count of 20; isolation window of 2.0 m/z; higher-energy collision dissociation (HCD) with the normalized collision energy (NCE) of 28; charge exclusion: unassigned 1, 2, >8; intensity threshold: 1000; and dynamic exclusion: 45 s. Each sample was analyzed three times. For the fractionated cross-linked peptides, the separation gradient was achieved by applying 2–7% B for 10 s, 7–23% B for 50 min, 23–40% B for 20 min, 40–80% B for 2 min, and 80% B for 13 min. The AGC target for MS2 was 5e4 and the dynamic exclusion was set to 18 s. The other parameter settings were the same as above. Each fractionation was analyzed three times. The sample concentration was measured using NANODROP ONE (Thermo Fisher Scientific, USA) to control the approximately same loading amount of each sample.

### Database analysis and processing

Changes in the protein abundance between the native and 70DSS-DDAB@PLGA/Kolliphor EL NPs-treated HepG2 cells were calculated through label-free quantification in MaxQuant software (v 1.6.3.3). The human protein sequences were downloaded from UniProt (downloaded in May 2018, with 70,956 entries) with a common contaminant database (https://www.uniprot.org/). The enzyme was set to trypsin with a maximum of two missed cleavages. The search tolerance for precursor ions was 10 ppm, and the search tolerance for fragment ions was 20 ppm. The fixed modification was set to carbamidomethyl cysteine. Methionine oxidation, protein N-terminal acetylation, DSS modification of protein N-terminus (C$_8$H$_{12}$O$_3$, mass shift 156.0786442), and DSS modification of lysine were set as variable modifications. Six raw files from the two samples (treated cells versus native cells) were analyzed by the LFQ algorithm in the MaxQuant environment

(v1.6.3.3), and 'match between runs' was used with a retention time window of 1 min. To obtain the ratio of DSS-labeled mitochondrial peptides among the peptides detected in the three methods, three raw files for each sample were analyzed separately in the MaxQuant environment (v1.6.3.3). The search parameters were the same as above. To analyze the ptotein corona on the NPs, three raw files from three samples (CDNP^D complexes, CDNP^DL complexes and CDNP^DLC complexes) were analyzed by the i-BAQ algorithm in the MaxQuant environment (v1.6.3.3). The search parameters were the same as above. Furthermore, the protein ID was the first majority protein ID, and its identification was accepted when the protein proportion was >99.95% and the identification by site only was excluded. The software pLink 2.0 (v2.3.5) was used to identify the cross-links with a separate false discovery rate (FDR) of 1% at the spectrum level. Inter-protein cross-linked peptide pairs were used to reveal the protein interactions, and both the intra-protein cross-linked peptide pairs and loop-linked peptides were used to map the protein structures. To reveal the mitochondrial cross-linking information, 90 raw files from 30 fractionations were retrieved together using pLink 2.0. The obtained peptide spectra were matched to the MitoCarta3.0 database of annotated human mitochondrial proteins (https://www.broadinstitute.org/mitocarta). The search parameters were as follows: precursor mass tolerance of 20 ppm, fragment mass tolerance of 20 ppm, precursor filter tolerance of 10 ppm, cross-linker DSS (cross-linking sites K and protein N-terminus, cross-link mass shift = 138.0680796, and monolink mass shift = 156.0786442). The fixed modification was set to carbamidomethyl cysteine. Methionine oxidation and protein N-terminal acetylation were set as variable modifications. To assess the effect of CD-MS labeling on the mitochondrial proteins, peptide identification experiments were performed without further pre-fractionation using pLink 2.0. The obtained peptide spectra were matched against the above-mentioned human protein sequence database. The other parameter settings were consistent with the above descriptions.

### Alignment process of the cryo-EM structure of the supercomplex I₁III₂IV₁ (SCI₁III₂IV₁) structure with the AlphaFold model structure of some subunits

The structure of complex IV (PDB code: 5Z62) was aligned based on the SCI₁III₂IV₁ structure (PDB code: 5XTH) (RMSD = 0.775, 1292 to 1292 atoms). The AlphaFold model of COX7B (AF-P24311-F1-model_v1) was aligned based on complex IV (PDB code: 5Z62) (RMSD = 0.811, 48 to 48 atoms). The AlphaFold model of NDUFB4 (AF-O95168-F1-model_v1) was aligned based on the SCI₁III₂IV₁ structure (PDB code: 5XTH) (RMSD = 1.016, 112 to 112 atoms). The AlphaFold model of NDUFA2-AF-O43678-F1-model_v1 was aligned based on the SCI₁III₂IV₁ structure (PDB code: 5XTH) (RMSD = 0.756, 76 to 76 atoms). The AlphaFold model of UQCRC2 (AF-P22695-F1-model_v1) was aligned based on the AW chain of the SCI₁III₂IV₁ structure (PDB code: 5XTH) (RMSD = 0.769, 392 to 392 atoms), and on the AK chain of the SCI₁III₂IV₁ structure (PDB code: 5XTH) (RMSD = 0.758, 382 to 382 atoms). The AlphaFold model of UQCRB (AF-P14927-F1-model_v1) was aligned based on the AF chain of the SCI₁III₂IV₁ structure (PDB code: 5XTH) (RMSD = 0.402, 97 to 97 atoms), and on the AS chain of the SCI₁III₂IV₁ structure (PDB code: 5XTH) (RMSD = 0.268, 97 to 97 atoms). Some of the cross-linked peptide residues detected were in the unresolved region of the reported protein structure, including NDUFB4 (Lys-1 to Lys-7), NDUFA2 (Lys-13 to Lys-98), UQCRC2 (Lys-21 to Lys-23), UQCRB (Lys-4 to Lys-19, Lys-4 to Lys-12), and COX7B (Lys-1 to Lys-6). Thus, the detected cross-linked peptides from the above proteins were mapped to the AlphaFold predicted model.

### Statistics & reproducibility

Results were presented as the mean ± standard derivation (S.D.). And all attempts at replication were successful. The images were analyzed by the pearson correlation coefficient (PCC) using NIS-Elements AR (version 5.20.00) and Image J software (version 1.53e). The pLink 2 software (version 2.3.5) was used to identify the cross-linked information. MaxQuant software (version 1.6.3.3) was used to identify proteins and peptides. For the label-free quantification result, Perseus software (version 1.5.8.5) was used for the Student's T-test. The p-value was adjusted by multiple tests using a false discovery rate (FDR) (Permutation-based). The significance was measured by protein fold changes > 2 and $p < 0.01$. For structure visualization and protein docking, Pymol software was used to map the detected cross-linked peptides to the structure. The HADDOCK web service was used for protein docking with default parameters. Cytoscope software (version 3.8.0) was used to map the interaction network. For analysis of the pathway, the GO terms of the proteins were enriched using the DAVID (https://david.ncifcrf.gov/). The pathway of the cross-linked proteins was sorted using the KEGG mapper website (https://www.kegg.jp/kegg/mapper/search.html). We used the origin (version b9.5.1.195) for data statistics and display.

### Reporting summary

Further information on research design is available in the Nature Portfolio Reporting Summary linked to this article.

## Data availability

The proteomics raw data could be downloaded from the PRIDE Archive with accession number: PXD035433 and PXD038658. The experimental structures of the proteins were obtained from the PDB database (https://www.rcsb.org/) by PDB ID(s) (Including: 5Z62, 5XTH, 6CTO, 1Y8N, 6I4R, 6CFO, 3BG3, 4WLE, 5UZQ, 7LLA, 5UPP, 6VFZ, 6WCV, 6G4Q, 6ZPO and 1OCK). The predicated structure of proteins were obtained from the AlphaFold structure predictions (https://alphafold.ebi.ac.uk/entry/)[56] (Including: AF-P24311-F1-model_v1, AF-O95168-F1-model_v1, AF-O43678-F1-model_v1, AF-P22695-F1-model_v1, AF-P14927-F1-model_v1, AF-Q16822-F1-model_v1, AF-P09622-F1-model_v1, AF-P11498-F1-model_v1, AF-Q99798-F1-model_v1, AF-P36957-F1-model_v1, AF-Q02218-F1-model_v1, AF-P56385-F1-model_v1, AF-P53007-F1-model_v4, AF-Q00325-F1-model_v4, AF-Q02978-F1-model_v4, AF-O75746-F1-model_v4, AF-Q9UJS0-F1-model_v4, AF-Q9H1K4-F1-model_v4, AF-O43772-F1-model_v4, AF-Q9H936-F1-model_v4, AF-Q6NUK1-F1-model_v4, AF-Q70HW3-F1-model_v4, AF-Q86VD7-F1-model_v4, AF-Q9ULD0-F1-model_v4, AF-P56381-F1-model_v4, and AF-P21912-F1-model_v1. The human protein databases were obtained from UniProt (downloaded in May 2018, with 70,956 entries) with a common contaminant database (https://www.uniprot.org/). The copy number of proteins was obtained from Ref. 26 The Human.MitoCarta3.0 database was obtained from the MitoCarta3.0 website (https://www.broadinstitute.org/mitocarta). The STRING database was obtained from the STRING website (https://cn.string-db.org/). The source data for Fig. 2b–e and Supplementary Figs. 2a, 4–9, 11a, are available in the figshare repository (https://doi.org/10.6084/m9.figshare.23261618). Source data are provided with this paper.

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

## Acknowledgements

The authors also thank Prof. Baofeng Zhao for helpful suggestions and discussions on the PPI results, and Prof. Wei Wei for helpful suggestions and discussions on the fluorescence results. This work was supported by the National Key R&D Program of China (2018YFA0507703 to Q.Z), National Natural Science Foundation of China (22274152, 21874131, and 21991083 to K.Y, 32088101 to Y.Z.), CAS Youth Innovation Promotion Association (Y2021058 to K.Y.).

## Author contributions

Y.C., Y.X., Q.Z., K.Y., L.Z., and Y.Z. designed research; Y.C. performed the preparation and characterization of the nanocarrier, the fluorescence experiment, the flow cytometry experiment and the mass spectrometry experiment; W.Z. assisted in the mass spectrometry experiment research; Wj.Z. assisted on MS analyses; X.L. and H.G. contributed DSS-FITC reagents; Y.C. analyzed data; Y.C., Y.X., K.Y., and L.Z. wrote the paper; G.M., K.Y., Z.L., L.Z., and Y.Z. conceived and directed the research.

## Competing interests

The authors declare no competing interests.
