## [Peer Review File · Nature Communications]

REVIEWER COMMENTS

Reviewer #1 (Remarks to the Author):

The authors have explored a method for in situ crosslinking and mapping of mitochondrial proteins in living cells using DSS@PLGA/Kolliphor EL nanoparticles. The authors presented intensive data of mitochondrial protein analysis by their native microenvironment. However, some major issues remains and this reviewer would not recommend its publication in Nature Communications.

1. The major problem is that there is no control group compared to DDAB-DSS@PLGA/Kolliphor EL NPs group. Thus, neither mitochondria targeting nor data of protein analysis were convincing.
2. The zeta potential of nanoparticles was almost 40 mV. The effect of nanoparticles themselves on the type and quantity of proteins should be carefully investigated? Plenty of proteins might be adsorbed onto surface of nanoparticles. How can them be isolated from proteins in native microenvironment?
3. In Fig.2B, the loading efficiencies of 10DSS@PLGA/Kolliphor EL and 40DSS@PLGA/Kolliphor EL were lost.
4. The PDI values of the prepared NPs were missing although the authors mentioned it in line 374.
5. As mentioned in line 111-112, “only 5.2% of the proteins in the treated cells showed a 2-fold significant difference”; however, 5.2% of the proteins changes is not negligible. Could the percentage be reduced through formulation optimization like adjusting the amount of DDAB?
6. In general, the writing of this manuscript is unnormalized, with an example of the order of the figures in Fig. 2a-2c.

Reviewer #2 (Remarks to the Author):

Chen et al. present a novel method for protein cross-linking of mitochondrial proteins by utilizing functionalized nanoparticles for the delivery of the DSS cross-linker into mitochondria. The authors successfully demonstrate that the cationic surface of the nanoparticles directs the particles to the negatively charged membrane of the mitochondria and that the DSS cross-linker is delivered into the mitochondria. The authors are able to cross-link mitochondrial proteins in living cells and identify interactions of mitochondrial proteins that do not exist in the STRING database. Finally, identified protein interactions were mapped onto available high-resolution structures or protein structures

predicted with AlphaFold. While the manuscript is mostly well written, there are some major points which should be addressed before considering the manuscript for publication:

Major concerns:

- The authors discuss in-cell cross-linking using soluble cross-linkers and their disadvantages (note that there are no references for these statements). However, cross-linking of intact cells with formaldehyde, which recently appeared to be a major advance, is not discussed.
- The authors discuss cross-linking of intact cells and organelles, however, they only cite studies on mitochondria. Note that some key papers (cross-linking of mitochondria) are missing. In addition, the findings of previous studies are not sufficiently discussed. The discussion section reads like a summary.
- It is not entirely clear whether these nanoparticles have been developed in this study. The authors consider this development “the key step” (p.2, l.82), however, they cite several studies for this development.
- The authors state that there is no significant change in protein expression, however, they report a significant change in expression for several proteins (Fig. S2)
- In general, it is unclear when technical or biological replicates are used and how many
- The authors report on protein interactions that are not included in the STRING database. Note that the string database include direct and indirect connections. These connections are not necessarily physical protein interactions but in some cases are just based on mentioning of proteins in literature
- In general, I think the authors should provide more information on the experiments – what has been done and why and how.

Minor comments:

- Grammar and spelling in figure legends and methods should be checked carefully (e.g. Supplementary Materials: l.163 ‘prptides’). The method section should be generally improved (spelling and grammar)
- Which identified cross-links were used? Were database search results filtered?
- The authors should elaborate on cross-links located in unstructured regions in the predicted protein structures (for example Fig. S11, DLD, DLST). These protein structures were most likely predicted with low confidence score and, therefore, flexibility of these regions might result in the identified cross-links corresponding to different structures than those predicted with AlphaFold.
- P. 2, l. 68: rephrase. The current statement suggests that high-throughput analysis of protein interactions is performed. Protein interactions are identified during database search while the LC-MS analysis only delivers masses and fragment masses of peptides.
- Include identification and data analysis of cross-links in the workflow described on page 2 (l. 76-81), compare Fig. 1

- Supp. Materials, l. 242: 'cross-linked peptides were separated and displayed in SDS-PAGE, using native BSA as negative...' -> peptides are actually proteins. In addition, 'displayed' is a very unusual expression. Rephrase.
- Supp. Materials, l. 148: '70 000@200 resolving power' please specify and express scientifically.
- Figs. S6 and S7: labels of axes of pearson correlation are not readable
- Figure S12: proteins shown in red and green are specified. It is unclear why some proteins are shown in white.
- Data availability statement and information on PRIDE upload (or similar) are missing

Reviewer #3 (Remarks to the Author):

The manuscript from Chen et al has developed a new method for in -situ crosslinking and mapping of proteins. The experiments were well designed and the presentation was sound. The manuscript could be considered for publication in Nature Communications if the following concerns could be addressed:

- Line 140: "... and the α of the CD-MS method was 4.13 times higher than the in vivo XL-MS method (Fig. S8a)." But the figure shows it was 4.14?
- Line 143: "... proteins in the Homo sapiens database and ..." but in the figure Uniprot database was used. It is better to be consistent.
- Line 155: "Subsequently, the cross-linked proteins were extracted and digested, the cross-linked peptides were subjected to reversed-phase liquid chromatography (RPLC) fractionation, and finally, the cross-linked peptides were analyzed by high-throughput mass spectrometry." Can be simplified.
- Line 169: "Thus, the higher percentage of intermolecular cross-linked peptides was likely attributed to the higher mitochondrial protein concentrations in the living cells." What does this mean? Looks like it might be comparable, or we need to use the same conditions to test them with different methods.
- Line 319: " Subsequently, the CLSM/Revolution XD results further confirm that the nanoparticle specifically targeted mitochondria after endocytosis (Fig.2d), instead of the endoplasmic reticulum (PCC was 0.44 (Fig. S6b)) ...". It is great to find the overlapping of mitochondria and NP in the figure. However, the NP is larger than 100nm but Cy 5 is much smaller. Only part of the NP was marked? This also applies to Fig 2E DSS-FITC.

Language. The language is in general very good.

- Line 26: totally – it could be better to state "a total of 74 pairs of ..." rather than totally

- Line 134: "... profiles and was approximately 28% of the total release was observed at 6 h (Table. S1)."
Pls check this sentence. Probably better just use the "cumulative release"

Reviewer #4 (Remarks to the Author):

In this study Chen et al described an in situ mitochondria crosslinking approach. They use nanoparticle technology to deliver a hydrophobic crosslinker DSS directly to mitochondria and thus allow in situ crosslinking of mitochondrial proteins in their most native environment. I think this targeted crosslinking approach is a great idea and also appreciate the authors make use of specially designed mitochondrial targeting nanoparticles for the crosslinker delivery. However, I have several major points for the authors to address to improve the quality of the paper.

1. I think it is very important to know if nanoparticle delivery can be a generic method for in situ crosslinking, therefore I would like to ask the authors to provide more data on if and how other types of crosslinkers can be delivered by nanoparticles and to which other organelles. Here the authors use Kolliphor EL as the nanocarrier to solubilize the hydrophobic crosslinker DSS. Does it only work for hydrophobic crosslinkers? What about hydrophilic crosslinkers? What about several commonly used MS-cleavable crosslinks such as DSSO, DSBU and DSBSO? Regarding subcompartment targeting, are there possibilities to target other organelles other than mitochondria?

2. The authors identified 1203 crosslinked peptides in this study and this number is compared with an in vivo XL-MS method (reference 12) and an in vitro mitochondrial XL-MS method (I didn't find a reference for it). This comparison is not sufficient in my opinion because there are several other papers, such as Schweppe et al PNAS 2017, Liu et al, MCP 2018 and Linden et al, MCP 2020, also did in vitro mitochondria crosslinking. Although these three papers used either mouse or yeast mitochondria rather than human mitochondria, it is definitely feasible to perform the comparison using percentage of mitochondrial proteome coverage and protein homology. In another argument, the authors show their intermolecular crosslink counts is 13.13% which is much higher than in the isolated mitochondria of the HeLa cells (2.8%). So they claim "the higher percentage of intermolecular crosslinked peptides was likely attributed to the higher mitochondrial protein concentrations in the living cells". However, as far as I remember, the three references I mentioned above all have much higher percentages of intermolecular crosslinks (likely higher than 13.13%). Thus, I think to fairly discuss this point, the authors should provide comparison data to the other papers I mentioned above.

3. The structural analyses in this paper are mostly standard. This is okay as a technology paper, but I would suggest to significantly shorten these parts. In particular, "NDUFA4 is a subunit of complex IV instead of complex I" has been described in 2012 (Balsa E et al Cell metabolism, 2012) and well accepted now so there is no need to put emphasize on it and make a main figure for it. Furthermore, the authors also say "the detected crosslinks also provide new structural information for several ADP/ATP carriers,

such as SLC25A12 (a typo here, SLC not SCL) SLC25A13 and SLC25A22. However, in Fig. S14c, they only did standard crosslinking mapping and measured the C-alpha distances. I couldn't find any structural elucidation on the dynamics of these carrier proteins, as the authors stated on Page 7, line 308.

Taken together, I think this paper should be substantially revised by providing more data on the technology itself (my point 1 and 2) and significantly shorten and tone down the structural analysis part unless any new structural insights was indeed discovered by the crosslinks found in this study. For the latter, the authors need to provide sufficient modeling and/or structural data to support these new findings.

Response to the comments from the reviewers, manuscript NCOMMS-22-23692

Reviewer 1

***Overall Comment:** The authors have explored a method for in situ crosslinking and mapping of mitochondrial proteins in living cells using DSS@PLGA/Kolliphor EL nanoparticles. The authors presented intensive data of mitochondrial protein analysis by their native microenvironment. However, some major issues remains and this reviewer would not recommend its publication in Nature Communications.*

Author reply: We sincerely thank the reviewer for the comments, as well as the recognition of the mitochondrial protein analysis in native microenvironment and our efforts on this work. As for the concerns raised by the reviewer, please find below the point-by-point response.

***Comment 1:** The major problem is that there is no control group compared to DDAB-DSS@PLGA/Kolliphor EL NPs group. Thus, neither mitochondria targeting nor data of protein analysis were convincing.*

Author reply: We sincerely thank the reviewer for raising this important concern. Following his/her kind suggestion, the control group is designed as DSS@PLGA/Kolliphor EL NPs without dihexadecyl-dimethyl-ammonium bromide (DDAB), and control experiments have been carried out by measuring zeta potential and observing the co-localization of nanocarriers with mitochondria. Compared with the control group, the experimental group displayed positive rather than negative charges, and significant endocytosis and localization to mitochondria were observed. The main conclusion — the positively charged DDAB ligand doped with PLGA as the outer shell of DSS-DDAB@PLGA/Kolliphor EL NPs provides positive charge to the nanocarrier for targeting mitochondria — has been further verified.

As for the experimental details, the control group (70DSS@PLGA/Kolliphor EL NPs) holds negative surface charge as ~ -18 mV (Fig.R1a) compared with the experimental group (70DSS-DDAB@PLGA/Kolliphor EL NPs) with positive surface

charge as $\sim +38.5$ mV (Fig.R1a), which proved our experimental design by employing DDAB ligands doping to provide surface positive charge thus targeting the mitochondria. Subsequently, confocal microscopy experiments were performed to compare the co-localization of positively charged experiments NPs with negatively charged control NPs. Compared with the control group, a significantly greater overall uptake of the positively charged NPs were observed: the red fluorescence of positively charged NPs and the green fluorescence of Mito-Tracker matched well (PCC was 0.72) (Fig.R1c), while negatively charged control NPs without DDAB and the green fluorescence of Mito-Tracker did not significantly colocalize (PCC was 0.28) (Fig.R1d). Furthermore, Cy5-labeled experimental NPs were loaded with DSS-FITC, which is a DSS-modified fluorescent molecule. The overlap of fluorescent images between embedded DSS-FITC and Cy5-labeled experimental NPs (PCC was 0.83) (Fig.R1d) also demonstrated that DSS was transported to mitochondria by positively charged DDAB-based NPs instead of diffusion. In summary, compared with the control group, DDAB-functionalized DSS-DDAB@PLGA/Kolliphor EL NPs nanoparticles have been proven to successfully deliver encapsulated DSS to the mitochondria of HepG2 cells under culture conditions.

The control group have been inserted and highlighted on Page 4-5 in revised manuscript. Meanwhile, Figure R1 are added to supplementary information as Figure S7. The relevant content is shown as below for the reviewer's convenience.

“By exposing hydrophilic ammonium cation on the PLGA/DDAB shells of the nanocarrier, the surface of DSS-DDAB@PLGA/DDAB NPs have a positive charge of ~ 38.5 mV (Fig.S7a), and the spherical surface charge of DSS@PLGA/Kolliphor EL NPs without DDAB is ~ -18 mV (due to carboxylic groups) (Fig.S7a)” (Line 144-148, Page 4)

“On the contrary, the negatively charged NPs (without DDAB NPs) showed significantly reduced endocytosis and weaker colocalization with mitochondria (PCC was 0.28) (Fig.S7d)” (Line 173-175, Page 4)

“Furthermore, Cy5-labeled experimental NPs were loaded with DSS-FITC, which is a DSS-modified fluorescent molecule. The overlap of fluorescent images between embedded DSS-FITC and cy5-labeled experimental NPs (PCC was 0.83) also demonstrated that the DSS was transported to mitochondria by positively charged DDAB-based NPs instead of diffusion. (Fig.S7e)” (Line 178-182, Page 4)

We want to clarify here that, in our previous manuscript, we have cooperated with Professor Xia, Institute of Process Engineering, Chinese Academy of Sciences, on the fluorescence experiments on CLSM (Nikon) in his lab (Figure 2d-2e, S7e and S8a-d in text and SI). However, COVID 19 prevented us to carry out some control experiments on the same instrument in his lab. Alternatively, we used an andor live cell confocal imaging platform (Revolution WD) for the confocal microscopy experiments on the control samples (Figure S7c-d, in SI). Nevertheless, the differences in measurement instrument would not influence our conclusions for the significantly different colocalization results supporting our experimental designs in the manuscript.

Fig. R1. Control experiments for proving the mitochondria targeting by surface positive charge doping. a. Size and zeta potential of 70DSS-DDAB@PLGA/Kolliphor EL NPs and 70DSS@PLGA/Kolliphor EL NPs in water measured by DLS. b. Flow cytometry determination of the amounts of DDAB@PLGA/Kolliphor EL NPs (Gray green), cy5 labeled DDAB@PLGA/Kolliphor EL NPs (Blue) and cy5 labeled PLGA/Kolliphor EL NPs (Red). c. Co-localizations of the DDAB@PLGA/Kolliphor EL NPs/mitochondria, as measured by Revolution WD. The DDAB@PLGA /Kolliphor EL NPs and mitochondria were labeled by Cy5 (red) and Mito Tracker® Green FM (green), respectively. d. Co-localizations of the PLGA/Kolliphor EL NPs/mitochondria, as measured by Revolution WD. The PLGA/Kolliphor EL NPs and mitochondria were labeled by Cy5 (red) and Mito Tracker® Green FM (green), respectively. e. Co-

localizations of the DSS-FITC which was embed in the cy5-labeled DDAB-PLGA/Kolliphor EL NPs and cy5-labeled DDAB-PLGA/Kolliphor EL NPs, as measured by CLSM. Scale bars = 10 μ m.

Comment 2: *The zeta potential of nanoparticles was almost 40 mV. The effect of nanoparticles themselves on the type and quantity of proteins should be carefully investigated? Plenty of proteins might be adsorbed onto surface of nanoparticles. How can them be isolated from proteins in native microenvironment?*

Author reply: We thank the reviewer for his/her very thoughtful comment. Proteins adsorbed on the surface of nanoparticles, also known as protein coronas, are highly important factors for biomedical applications and safety of NPs of intracellular delivery (*Proc Natl Acad Sci U S A, 2022,119, e2200363119*). According to the suggestion by the reviewer, we systematically analyzed the protein on the surface of nanoparticles as they traversed from DMEM medium to cell lysosomes and escaped from lysosomes to cytoplasm where they are finally targeted to mitochondria in the HepG2 cells. The type and amount of proteins adsorbed on the nanoparticles in the three stages were determined by LC-MS-based experiments.

As for the first question, we observed that the amounts of protein adsorbed on the surface of nanoparticles is small at femtogram [fg]/particle level and the types of proteins were shown in the Table R1. Such a small amount of protein would lead to negligible influence to our analysis, since the detected cross-linked proteins were from mitochondria under native conditions, rather than adsorbed proteins.

Approximately 8.2% of all absorbed proteins (21 overlapping proteins) were consistently present during the whole sequential incubation process. In additional, at each stage, approximately half of the proteins adsorbed to the nanoparticles were weakly adsorbed and desorbed on the way to the next stage. Thus, we infer that the DSS-DDAB@PLGA/Kolliphor EL NPs would adsorb proteins during delivery, but small part of the proteins was strongly bound on the nanoparticles. In addition, we

found that the mitochondrial proteins detected in our manuscript were independent of the proteins absorbed on the nanoparticles prior to targeting mitochondria, so proteins absorbed on the nanoparticles did not affect in situ release of the cross-linker in the mitochondria for cross-linking.

As for the experimental details, in order to mimic the internalization pathway, CDNPs (70DDAB@PLGA/Kolliphor EL NPs) were incubated with DMEM medium (CDNPs^D: 70DSS-DDAB@PLGA/Kolliphor EL NPs DMEM) and sequentially transferred to freshly isolated lysosomal (CDNPs^{DL}: 70DSS-DDAB@PLGA/Kolliphor EL NPs DMEM-lysosome) and cytosol extracts (CDNPs^{DLC}: 70DSS-DDAB@PLGA/Kolliphor EL NPs DMEM-lysosome-cytoplasm) from Human hepatocellular carcinomas cell line (HepG2 cells) (Fig.R2a, Fig.R2b). The amount of protein on nanoparticles was quantified as ~0.115 fg per particle on CDNPs^D, ~0.274 fg per particle on CDNPs^{DL}, and ~2.692 fg per particle on CDNPs^{DLC} (Fig. R3a). CDNPs^{DL} adsorbed 0.07% of protein in isolated lysosomal media, while CDNPs^{DLC} adsorbed 0.10% of protein in cytosolic extraction media, compared to the total amount of protein in each cell's biological media. Furthermore, given that single HepG2 cells contain $\sim 0.17 \times 10^6$ fg of total protein and mitochondria in a single HepG2 cell contain $\sim 2 \times 10^4$ fg of total protein (*J Proteomics 2016, 136, 234-247*), the amount of protein adsorbed on the surface of nanoparticles is very small.

To explore hard proteins consistently adsorbed on nanoparticles, the overlap between the proteins on the three nanoparticles was analyzed by Venn (Fig. R3b). 8.2% of all adsorbed proteins (21 overlapping proteins) were consistently present during the whole sequential incubation process. 51.9% of the adsorbed proteins were weakly adsorbed to the nanoparticles, as they were only detected in the respective fractions. This result suggested that about half of proteins are dynamically changing at each stage of protein adsorption on nanoparticles. As for the cross-linking information of proteins adsorbed on nanoparticles before targeting to mitochondria, we identified a pair of cross-linked peptides that were not related to the mitochondrial proteins identified in

the real experiment (Fig.R3c). Therefore, the adsorbed protein is not related to the proteins in the real experiment.

The following text was added to the revised manuscript (line 108). Meanwhile, Figure R2-R3 are added to supplementary information as Figure S10- Figure S11: "To elucidate the effect of adsorbed proteins on nanoparticles on in situ mitochondrial crosslinking, the interactions of 70DSS-DDAB@PLGA/Kolliphor EL NPs with proteins at different stages before targeting to mitochondrion. From the CLSM experiment, we observed that the nanoparticles were endocytosed by HepG2 cells in DMEM medium and then escaped from lysosomes to cytoplasm for targeting to the mitochondrion. Thus, to mimic the internalization pathway, nanoparticles were incubated in DMEM medium to form CDNPs^D (70DSS-DDAB@PLGA/Kolliphor EL NPs DMEM), which were then sequentially transferred into freshly isolated lysosomes to form CDNPs^{DL} (70DSS-DDAB@PLGA/Kolliphor EL NPs DMEM-lysosome), and finally incubated with cytosol extract to form CDNPs^{DLC} (70DSS-DDAB@PLGA/Kolliphor EL NPs DMEM-lysosome-cytoplasm) (Fig. S10a, Fig. S10b). Subsequently, the protein corona information in three biological media (DMEM medium, isolated lysosomal, cytosol extracts) were analyzed separately. The amount of protein on nanoparticles was quantified as ~0.115 fg per particle on CDNPs^D, ~0.274 fg per particle on CDNPs^{DL}, and ~2.692 fg per particle on CDNPs^{DLC} (Fig. S11a). CDNPs^{DL} adsorbed 0.07% of protein in isolated lysosomal media, while CDNPs^{DLC} adsorbed 0.10% of protein in cytosolic extraction media, compared to the total amount of protein in each cell's biological media. Furthermore, given that single HepG2 cells contain $\sim 0.17 \times 10^6$ fg of total protein and mitochondria in a single HepG2 cell contain $\sim 2.0 \times 10^4$ fg of total protein (*J Proteomics 2016, 136, 234-247*), the amount of protein adsorbed on the surface of nanoparticles is very small. To explore hard proteins consistently adsorbed on nanoparticles, the overlap between the proteins on the three nanoparticles was analyzed by Venn (Fig. S11b). Approximately 8.2% of all absorbed proteins (21 overlapping proteins) were consistently present during the whole sequential incubation process. In addition, at each stage, approximately half of the

proteins adsorbed to the nanoparticles were weakly adsorbed and desorbed on the way to the next stage. Thus, we infer that the DSS-DDAB@PLGA/Kolliphor EL NPs would adsorb proteins during delivery, but small part of the proteins was strongly bound on the nanoparticles. In addition, we found that the mitochondrial proteins detected in our manuscript were independent of the proteins absorbed on the nanoparticles prior to targeting mitochondria, so proteins absorbed on the nanoparticles did not affect their in situ release of the cross-linker in the mitochondria for cross-linking (Fig. S11c).” (Line 197-227, Page 5)

Fig.R2. Flowchart of Nanoparticle-corona complexes preparation. a. Flowchart of (a1) cytosol and (a2) lysosome from HepG2 cells. b. Flowchart of nanoparticle incubation in the extracted bio-fluids to mimic endocytosis of CDNPs.

Fig R3. Assessment of nanoparticle-corona complexes. a. Quantification (protein femtogram [fg] per particle) of the protein amount within the three specific protein coronas. Data are shown as mean \pm SD from triplicates. Proteins identified by LC-MS/MS in the respective nanoparticle coronas under three different incubation conditions, based on the biological processes of the CDNPs before reaching the mitochondria of HepG2 cells. b. Venn diagram of the numbers of identified proteins on the CDNPs^D, CDNPs^{DL}, and CDNPs^{DLC} surfaces. Detailed values for all individual proteins are available in SI Appendix, Table S2. c. Evaluation of the number of cross-links on the CDNPs-DLC surfaces.

Table R1 are added to supplementary information as Table S3.

Table R1. The adsorbed proteins on the CDNPs^D, CDNPs^{DL}, and CDNPs^{DLC} surfaces, corresponding to the numbers in Fig.R3b. Proteins from Homo sapiens are shown in Blank (From HepG2 cell); Proteins from Bovine are shown in Red (From the with 10% fetal bovine serum).

	21	97	83	31	3	2	19	
CDNPs-DLC	CD44 H2BC4 IFIH3 HSPA5 APOA1 GAPDH COTRI1 TUBA1C PKM VIM ACTG1 AMBP KRT1 KRT18 EEF1A1 HBB H1-2 VTN KRT17 KRT7 APOE	HSPD1 ACTB YBX1 ANXA2 SUMF2 ENO1 RPL12 HYOU1 TUBB HSPB1 PRKB PRKCSH RRBP1 RPS18 RPLP1 PPIA SCAMP3 LMAN1 HNRNPH TXNDC5 HLA-C FAU MYH9 RPL19 EEF2 HNRNPM	APEX1 FDIA5 TFRC RPL6 SPTBN1 MPDU1 RPL13A SERPINH RACK1 ILF3 RPS14 RPS5 SRP14 RPN1 ATP5F1B MT-CO2 RPS24 HNRNPU PLEC HSP90AB PRDX4 YCP SLC3A2 RPS16 ALDOA RPL18	SQSTM1 SNRPD3 HSPA8 FDIA3 RPL11 TUBB4B RPL9 SPCS2 RPL8 SNRPG PLP2 H2AC6 EIF3F VDAC1 HNRNPK RHOG SSR4 BANF1 HNRNPC EIF4A1 ALPP P4HB CKAP4 TMX1 COLGALT1 PARP1 RPL15 ATP5F1A GNAO1 H4C1 HNRNPA2B1	RPL10 RPLP0 RPS25 PCBP1 VAT1 RPS17 RPS4X RPS3A RPL36 RPN2 RPL23 CRIP1 SAR1A CNPY2 ITGB1 RPL10A RPL7A RPL17 RPS7 CKB CALR EIF3F ACTC1 NONO G3BP1 RPL2 RPS12 RPL27 HSPA7 DAD1 PCBP2 CCT2 RPL38 RHEB RPS6 ARL8B RPL35 RPL32 MYL12A RAB8A FUS RPS9 RPS15A RPL4	HNRNPA1 HSG15 RPL39P5 TMED10 RRM1 S100A10 RPL27A RAB7A H3C1 M6PR RPL31	C4 TUBA1B	
CDNPs-DL				GAPDH5 NDUFS6 PDHA1 NDUFS2 RNPS1 TMEM3 PHB2 MDH2	PSMD11 NEFM SEC22B NDUFS1 UQCRRS1 SYNCRIP DDX18 LETM1	AK2 ERP29 SFPQ TOMM8 LAMTOR3 SUCLA2 H3-7 ARL8B	MFF ERP44 HAX1 PHB1 PF4 TOP1 EDF1	PGAM2 KRT5 PRSS1
CDNPs-D							CREG1 FMOD PARVG OR1L6 XP32 F13B KRT14 LUM KPRP CLU	OXNAD1 KRT9 S100A8 MRPS21 CCNC F2 KRT10 NEFH S100A9

Comment 3. In Fig.2B, the loading efficiencies of 10DSS@PLGA/Kolliphor EL and 40DSS@PLGA/Kolliphor EL were lost.

Author reply: We thank the reviewer for pointing it out. The loading efficiencies of 10DSS@PLGA/Kolliphor EL NPs was 0.102%, shown in Fig.2B as an orange symbol; the loading efficiencies of 40DSS@PLGA/Kolliphor EL NPs was 19.37%, shown in Fig.2B as a purple symbol. Since the symbol representing the loading efficiencies of 10DSS@PLGA/Kolliphor EL NPs was too small to be seen clearly in the original image, we adjusted its size of the Fig.2B in the revised manuscript.

Comment 4. The PDI values of the prepared NPs were missing although the authors mentioned it in line 374.

Author reply: We thank the reviewer for pointing it out. The results of this experiment are addressed individually as follows. Table R2 are added to supplementary information as Table S1.

Table R2. Size PDI and Zeta Potential Variation from three NPs in Fig.2b.

	Diameter(nm)	PDI	Zeta Potential(mv)
10DSS@PLGA/Kolliphor EL NPs	321.8±5.9	0.080	37.5±0.4
40DSS@PLGA/Kolliphor EL NPs	297.2±2.4	0.238	40.0±1.0
70DSS@PLGA/Kolliphor EL NPs	275.7±5.5	0.067	38.5±2.5

Comment 5. *As mentioned in line 111-112, “only 5.2% of the proteins in the treated cells showed a 2-fold significant difference”; however, 5.2% of the proteins changes is not negligible. Could the percentage be reduced through formulation optimization like adjusting the amount of DDAB?*

Author reply: We followed this constructive suggestion by adjusting the amount of DDAB to optimize the nanoparticles and reduce the proteomic perturbations. By adjusting the DDAB content in DSS-DDAB@PLGA/Kolliphor EL NPs, we achieve the percentage of proteins changes reduced from the 4.86% to 3.49%, as the reviewer supposed.

As for the experimental details, different amounts of DDAB were applied in nanoparticles fabrication to examine how the proteins in live HepG2 cells were influenced by the DDAB parameters, where DDAB content in 70DSS-DDAB@PLGA/Kolliphor EL NPs was changed to 80% (Treated-2), 60% (Treated-3), 40% (Treated-4) and 20% (Treated-5) of the initial content (Treated-1), respectively. As the amount of DDAB in the nanoparticles was reduced to 20% (Treated-5) of the initial content, the percentage of proteins with 2-fold significant changes at the proteome level in the treated HepG2 cells decreased from 4.86% to 3.49% (Fig.R4b), and the surface charge of the nanoparticles decreased from ~+36.4 mV to ~+22.2 mV (Fig.R4a). The surface charge reduction of nanoparticles induced by DDAB is not responsible for changes in the number of changed proteins, as the charge was similar between groups 3 and 5, but the percentage of protein change was lower for the latter. Therefore, we infer that DDAB itself may affect the cells, which we would like to reform in the future. A puzzling fact is the lack of reports on the percentage of changes

protein after cross-linking coupled to mass spectrometry (XL-MS) measurements, thus there are no standard to refer to. The current result is barely a start, which we would like to further improve in the future.

Meanwhile, Figure R4 was added to supplementary information to replace Figure S2. A brief discussion was also added below the figure.

Fig. R4. The effect of DDAB content in nanoparticles on the percentage of proteins with 2-fold significant changes at the proteome level in the treated HepG2 cells. a. Size and zeta potential of DSS-DDAB@PLGA/Kolliphor EL NPs with different amount of DDAB in pure water measured by DLS. b. Significant changes of protein abundances. Protein changes at the cellular protein levels in cells treated with the highest loading efficiency nanoparticle for 6 h, which were measured using the label-free protein quantification method (treated1-5). c-g. Label-free quantification was performed to calculate the changes of protein abundances between native and different 70DSS-DDAB@PLGA/Kolliphor EL NPs-treated HepG2 cell. The DDAB content in different 70DSS-DDAB@PLGA/Kolliphor EL NPs was changed to 80% (Treated-2), 60% (Treated-3), 40% (Treated-4) and 20% (Treated-5) of the initial content (Treated-1), respectively. Proteins with a significant change in LFQ intensity of more than 2-fold

(p-value < 0.01) in the treated group compared to the untreated condition are shown as red dots, while the remaining proteins are shown as black dots.

We therefore rephrase our conclusions in a more careful manner: “After 6 h of nanoparticle delivery, 4.9% of the proteins in the treated cells showed a 2-fold significant difference (Fig. S2). Thus, the cross-linker based nanocarrier did not cause most detected 3002 cellular proteins (2856 cellular proteins) to differ significantly from the control in proteome level. In addition, when the DDAB content in DSS-DDAB@PLGA/Kolliphor EL NPs was greatly reduced, the percentage of disrupted proteins in the proteome level was reduced in treated HepG2 cells. A puzzling fact is the lack of reports on the percentage of changes protein in the results of cross-linking coupled to mass spectrometry (XL-MS), thus there are no standard to refer to. This result also suggests that we still have opportunities to tune nanoparticles to be more biocompatible in the future.” (Line 156-165, Page 4)

6. In general, the writing of this manuscript is unnormalized, with an example of the order of the figures in Fig. 2a-2c.

Author reply: According to the suggestion, we change the order of the figures.

Finally, we would like to thank the reviewer again for the critical comments, and hope we have clarified them appropriately.

Reviewer 2:

Overall Comment: Chen et al. present a novel method for protein cross-linking of mitochondrial proteins by utilizing functionalized nanoparticles for the delivery of the DSS cross-linker into mitochondria. The authors successfully demonstrate that the cationic surface of the nanoparticles directs the particles to the negatively charged

membrane of the mitochondria and that the DSS cross-linker is delivered into the mitochondria. The authors are able to cross-link mitochondrial proteins in living cells and identify interactions of mitochondrial proteins that do not exist in the STRING database. Finally, identified protein interactions were mapped onto available high-resolution structures or protein structures predicted with AlphaFold. While the manuscript is mostly well written, there are some major points which should be addressed before considering the manuscript for publication:

Author reply: We thank the reviewer for the high evaluation and constructive suggestions, which will further enhance the quality of this manuscript. Please find below the point-by-point responses to all the comments.

Major concerns:

Comment 1. *The authors discuss in-cell cross-linking using soluble cross-linkers and their disadvantageous (note that there are no references for these statements). However, cross-linking of intact cells with formaldehyde, which recently appeared to be a major advance, is not discussed.*

Author reply: We completely agree with the reviewer and acknowledge this recommendation in the introduction. Following this suggestion, references on the hydrolysis of soluble NHS cross-linkers are cited in the modified manuscript (*Mol Cell Proteomics*, 2010, **9**, 1634-1649). N-hydroxysuccinimide ester cross-linker is the most frequently used reactive group because it can reactive with primary amine in proteins under physiological conditions (pH 7.0-7.5) (*Mol Biosyst*, 2010, **6**, 939-947). At the same time, NHS ester is rapid hydrolysis in aqueous solutions, and the half-life is about tens of minutes under typical reaction conditions (pH > 7, 25–37 °C). (*Mol Cell Proteomics*, 2010, **9**, 1634-1649).

For crosslinking of cells under native condition, formaldehyde is also a suitable crosslinker for analyzing the protein within cells because the small FA molecule can

rapidly permeate cell membrane to efficient crosslink and form covalent bonds with proteins. Limited by the identification of formaldehyde cross-linking, formaldehyde cannot yet to be commonly employed in the directed analysis of protein–protein interactions and cellular networks in complex system (*J Mass Spectrom*, 2008, 43, 699-715). Terry Wilke et al. (*Nat Commun*, 2020, 11, 3128) pioneered a straightforward formaldehyde mechanism whereby FA adducts lead to distinct reaction products with a mass of 24 Da, well resolved this issue. Then, they successfully applied FA crosslinking MS to human cell and explained new structural insights into different proteins in a complex or within the same protein with a relatively short spacer arm (in the range of 2.3–2.7 Å). Similar, in situ analysis of proteins in cellular organelles under their native working environments, remains challenging in the field of biology. Due to the excellent membrane penetration of FA, we think the FA crosslinking MS has the potential provide a protein information in a native subcellular, while it may be more prior to provide the structure information of protein with a relatively short spacer arm. Due to the excellent membrane penetration of FA, we believe that FA crosslinking MS has the potential to provide close-range cross-linking information in protein in native microenvironment. Here, we introduced an unprecedented method (targeted crosslinker delivery coupled with mass spectrometry (CD-MS)) for in situ protein analysis in living cells under native microenvironments by integrating targeted nanoparticle delivery and high-throughput mass spectrometry.

We completely agree with the reviewer and therefore complete our introduction by adding a discussion of this important finding: “For crosslinking of cells under native condition, formaldehyde is a suitable crosslinker for analyzing the protein within cells because the small FA molecule can rapidly permeate cell membrane to efficient crosslink and form covalent bonds with proteins (*J Mass Spectrom*, 2008, 43, 699-715). In order to apply formaldehyde crosslinking to complex systems, Terry Wilke et al. (*Nat Commun*, 2020, 11, 3128) pioneered a straightforward formaldehyde mechanism whereby FA adducts lead to distinct reaction products with a mass of 24 Da and applied it to the detection of cross-linked peptides in human cells fixed in situ with

formaldehyde. In addition, N-hydroxysuccinimide ester cross-linker is the most frequently used reactive group because it can reactive with primary amine in proteins under physiological conditions (pH 7.0-7.5) (*Mol Biosyst*, 2010, 6, 939-947).” (Line 66-75, Page 2)

Comment 2. *The authors discuss cross-linking of intact cells and organelles, however, they only cite studies on mitochondria. Note that some key papers (cross-linking of mitochondria) are missing. In addition, the findings of previous studies are not sufficiently discussed. The discussion section reads like a summary.*

Author reply: We thank the reviewer for bringing up this important point. Following this suggestion, recent key studies on intact cells (*Nat Commun*, 2020, 11, 3128; *Proc Natl Acad Sci U S A*, 2021, 118, e2023360118; *Anal Chem*, 2022, 94, 7551-7558), isolated nuclei (*Mol Cell Proteomics*, 2018, 17, 2018-2033) and mitochondria (*Mol Cell Proteomics*, 2020, 19, 1161-1178, *Mol Cell Proteomics*, 2020, 19, 624-639) are cited in the revised manuscript.

We modified our introduction by adding a discussion of the findings of previous studies: “Fassi et al. apply XL-MS to isolated nuclei of osteosarcoma cells cross-linked by DSSO to study the nuclear interactome and protein conformation at proteome level (*Mol Cell Proteomics*, 2018, 17, 2018-2033). Multiple in situ XL-MS studies have explored various NHS-ester crosslinkers to reveal details on mitochondrial proteins and interactions in different complex systems, such as BDP-NHP (*Proc Natl Acad Sci U S A*, 2017, 114, 1732-1737; *Cell Syst*, 2018, 6, 136-141 e135), DSSO (*Mol Cell Proteomics*, 2018, 17, 216-232), CBDPS (*Mol Cell Proteomics*, 2020, 19, 624-639), DSS (*J Proteome Res* 19, 2020, 327-336, *Mol Cell Proteomics*, 2020, 19, 1161-1178) and BS3 (*Mol Cell Proteomics*, 2020, 19, 1161-1178). Among them, Schweppe et al (*Proc Natl Acad Sci U S A*, 2017, 114, 1732-1737) and Lu et al (*Mol Cell Proteomics*, 2018, 17, 216-232) applied XL-MS to mitochondria isolated from mouse heart tissue and provided mitochondrial protein structure (e.g., five OXPHOS complexes) and interaction information, respectively. Ryle et al (*J Proteome Res* 19, 2020, 327-336)

applied the DSS to human mitochondria isolated from the cultured bone marrow lymphoblast cell line K-562 and reveal protein flexibility of mitochondrial heat shock proteins. Makepeace et al (*Mol Cell Proteomics*, 2020, 19, 624-639) and Linden et al (*Mol Cell Proteomics*, 2020, 19, 1161-1178) applied cleavable cross-linker CBDPS and noncleavable cross-linkers BS3 and DSS to mitochondria purified yeast for in situ analysis mitochondrial protein information, respectively. However, in all previous reports, direct detection of interactions between mitochondrial proteins under culture condition has not been reported at the proteomic level.” (Line 49-63, Page 2)

Comment 3. *It is not entirely clear whether these nanoparticles have been developed in this study. The authors consider this development “the key step” (p.2, l.82), however, they cite several studies for this development.*

Author reply: Nanoparticles for targeted drug delivery have been widely studied with great potential of changing the paradigm of treating human diseases with high efficacy and accuracy. However, targeted nanoparticle delivery for in situ protein analysis in subcellular environments has received little if any attention. We were inspired by this targeted nanoparticle delivery concept to develop this innovative protein analysis method. Considering the purpose of delivering the cross-linker to the mitochondria through the nanoparticles, the positively DSS-DDAB@PLGA/Kolliphor EL NPs are carefully designed by us with three elements: 1. The PLGA/Kolliphor EL-hybrid particle was first made and encapsulated the DSS with high loading efficiency (24.84%(w/w)). 2. The positively charged dihexadecyldimethylammonium bromide (DDAB) ligands were innovatively designed doped with PLGA as the outer shell of the nanoparticle, providing a positive charge in the nanocarrier to target the mitochondria. We firstly demonstrated that DDAB@PLGA/Kolliphor EL NPs can target mitochondria, which broadens the range of cationic groups targeted to mitochondria. 3. DSS-DDAB@PLGA/Kolliphor EL NPs can deliver crosslinkers under culture conditions to efficiently crosslink mitochondrial proteins in situ.

We therefore rephrase our designs in a more careful manner in the revised manuscript: “We were inspired by the targeted nanoparticle delivery concept to develop an innovative protein analysis method, where crosslinker-based nanoparticles

have been carefully designed to meet the needs of CD-MS methods for analyzing the mitochondrial proteome in living cells.” (Line 120-123, Page 3)

Comment 4. *The authors state that there is no significant change in protein expression, however, they report a significant change in expression for several proteins (Fig. S2).*

Author reply: We completely agree with the reviewer and acknowledge this recommendation. This comment is also proposed by reviewer 1 as “Could the percentage be reduced through formulation optimization like adjusting the amount of DDAB?”

As suggested, by adjusting the DDAB content in DSS-DDAB@PLGA/Kolliphor EL NPs, we achieve the percentage of proteins changes reduced from the 4.86% to 3.49%, as the reviewer supposed. Thus, we therefore rephrase our conclusions in a more careful manner in the revised manuscript: “After 6 h of nanoparticle delivery, 4.8% of the proteins in the treated cells showed a 2-fold significant difference (Fig. S2). Thus, the cross-linker based nanocarrier without causing most of the 3002 cellular proteins (2856 cellular proteins) detected significantly differ from control. In addition, when the DDAB content in DSS-DDAB@PLGA/Kolliphor EL NPs was greatly reduced, the percentage of disrupted proteins in the proteome level was reduced in treated HepG2 cells. A puzzling fact is the lack of reports on the percentage of changes protein in the results of cross-linking coupled to mass spectrometry (XL-MS), thus there are no standard to refer to. This result also suggests that we still have opportunities to tune nanoparticles to be more biocompatible in the future.” (Line 156-165, Page 4)

Comment 5. *In general, it is unclear when technical or biological replicates are used and how many*

Author reply: 1. For the experiments involving DSS-DDAB@PLGA/Kolliphor EL NPs, such as DSS-DDAB@PLGA/Kolliphor EL NPs preparation, Size and Zeta potential test, etc., technical triplicates were carried out were carried out.

2. For the LC-MS analysis of cross-linked HepG2 and the percentage of disturbed protein in treated HepG2 by LFQ methods, every sample subjected to LC-MS/MS were measured in triplicates on an QE Mass Spectrometry and each raw is analyzed separately. In addition, Cross-linked peptides of every peptide RPLC fraction subjected to LC-MS/MS were measured in three technical duplicates on an QE Mass Spectrometry and the ninety raw files were combined for data analysis.

3. For the detection of protein absorbed on the DSS-DDAB@PLGA/Kolliphor EL NPs before targeting to the mitochondria during the intracellular delivery, independent experiment was carried out and the single raw was used for analysis.

Comment 6. *The authors report on protein interactions that are not included in the STRING database. Note that the string database include direct and indirect connections. These connections are not necessarily physical protein interactions but in some cases are just based on mentioning of proteins in literature*

Author reply: We compared protein-protein interactions not included in the STRING database with the BioGrid database and found that they were also not included in the BioGrid database. For protein-protein interactions annotated in the string database based on proteins mentioned in the literature, we found one crosslink in the STRING database was barely from the literature. Thus, the cross-linked peptide pair linking lysine 525 of 2-oxoglutarate dehydrogenase-like (OGDHL) to lysine 11 of pyruvate dehydrogenase E1 component subunit α (PDHA1), filled a gap in the STRING database where the related experimental evidence was lacking. The intracellular fulfilled with the crowd macromolecular (most are proteins and nucleus) with several hundred milligrams per milliliter, which impacts on protein conformations and interactions (*J Mol Biol* 222,1991, 599-620; *J Am Chem Soc*, 2013, 135, 13796-13803). Thus, some of the interactions captured via our method are dynamic or transient.

Comment 7. *In general, I think the authors should provide more information on the experiments – what has been done and why and how.*

Author reply: We thank the reviewer for the comments. The manuscript (experiments) has been rigorously revised to explain the experimental details.

Minor comments:

Comment 8. *Grammar and spelling in figure legends and methods should be checked carefully (e.g. Supplementary Materials: 1.163 ‘prptides’). The method section should be generally improved (spelling and grammar)*

Author reply: Grammar and spelling in figure legends and methods have been checked carefully. The method section has been polished in the aspect of spelling and grammar.

Comment 9. *Which identified cross-links were used? Were database search results filtered?*

Author reply: The identified cross-links consisted of cross-linked peptides pairs and loop-linked peptides, which were retrieved by pLink 2.0 software. Cross-linked peptides are two peptides linked by a crosslinker, including intraprotein cross-linked peptide pairs where the linked peptides belong to the same protein, and interprotein cross-linked peptide pairs where the linked peptides belong to different proteins. Loop-linked peptides represent single peptide chains modified with the cross-linker. Interprotein cross-linked peptide pairs were used to reveal protein interactions, and both intraprotein cross-linked peptide pairs and loop-linked peptides were used to map protein structures. The plink 2.0 was used for the identification of the cross-links with separate false discovery rate (FDR) of 1% at spectrum level. This information has been added in the revised SI: “The plink 2.0 (v 2.3.5) was used for the identification of the cross-links with separate false discovery rate (FDR) of 1% at spectrum level.

Interprotein cross-linked peptide pairs were used to reveal protein interactions, and both intraprotein cross-linked peptide pairs and loop-linked peptides were used to map protein structures.”

Comment 10. *The authors should elaborate on cross-links located in unstructured regions in the predicted protein structures (for example Fig. S11, DLD, DLST). These protein structures were most likely predicted with low confidence score and, therefore, flexibility of these regions might result in the identified cross-links corresponding to different structures than those predicted with AlphaFold.*

Author reply: We completely agree with the reviewer, and acknowledge this recommendation. Although the observed crosslink violated predicted model, it may be due to the flexibility of these regions and the distance constraints of cross-linked peptides may also provide a reference for the structure of dynamic regions of the protein.

In line we added the sentence in the revised manuscript: “What’s more, there are two incompatible cross-links on predicted structure. The residues that connected by the two cross-links are fall within the confidently prediction with the predicted local distance difference test (pLDDT) score below 70, it indicated that there is a certain deviation between the predicted result and the actual result. Therefore, flexibility of these regions might result in the identified cross-links corresponding to different structures than those predicted with AlphaFold (Fig. S16).” (Line 307-313, Page 7)

Figure R5: AlphaFold prediction model. a. AlphaFold prediction model for DLD. The prediction is coloured by model confidence band. b. AlphaFold prediction model for DLST. The prediction is coloured by model confidence band.

Comment 11. P. 2, l. 68: rephrase. The current statement suggests that high-throughput analysis of protein interactions is performed. Protein interactions are identified during database search while the LC-MS analysis only delivers masses and fragment masses of peptides.

Author reply: We completely agree with the reviewer, and acknowledge this recommendation in introduction. We changed the sentence in the revised manuscript: “The peptides were fractionated by reversed-phase liquid chromatography (RPLC) for measurement in a high-throughput mass spectrometer, followed by analysis of protein-protein interactions with $C\alpha$ - $C\alpha$ distances below 30 Å, an upper bound distances for DSS crosslinking supported by molecular dynamics simulations. (*J. Proteome Res.* 2020, 19, 1, 327–336; *Protein Science*, 2014, 23, 747-759)” in the revised manuscript. (Line 93-97, Page 2)

Comment 12. Include identification and data analysis of cross-links in the workflow described on page 2 (l. 76-81), compare Fig. 1

Author reply: We note that the fifth step in Figure 1 – determining crosslinks and matching protein structures and mapping protein interactions – is missing in the workflow. Thus, we modified the workflow in the revised manuscript: “(5) identification of cross-linked peptides for matching protein structures and mapping protein interactions” in the revised manuscript. (Line 118-119, Page 3)

Comment 13. Supp. Materials, l. 242: ‘cross-linked peptides were separated and displayed in SDS-PAGE, using native BSA as negative...’ -> peptides are actually proteins. In addition, ‘displayed’ is a very unusual expression. Rephrase.

Author reply: We thank the reviewer for pointing it out. In the method we rephrased the sentence as “cross-linked BSA was separated by 12% SDS-PAGE gel, using native BSA as negative control. The gel was stained with Coomassie Brilliant Blue.”

Comment 14. Supp. Materials, 1. 148: ‘70,000@200 resolving power’ please specify and express scientifically.

Author reply: We thank the reviewer for carefully spotting this unscientific expression. In the method we changed the sentence “The method parameters of the run were as follows: data-dependent acquisitions; Full MS resolution 70,000 at m/z 200; Scan range 300-1800; MS1 AGC target 3e6; MS1 Maximum OT 80 ms; the loop count was 20, the isolation window of 2.0 m/z. MS/MS scans were detected at the resolution of 17,500 at m/z 200.”

Comment 15. Figs. S6 and S7: labels of axes of Pearson correlation are not readable

Author reply: Thanks for the kind notification. The labels of axes of Pearson correlation have been corrected in Figs. S8 and S9 for easy reading in the revised Supporting Information. This figure has been changed in the revised *SI*.

Comment 16. Figure S12: proteins shown in red and green are specified. It is unclear why some proteins are shown in white.

Author reply: The KEGG molecular networks are developed in a generic way, namely, in terms of functional orthologs, called KO (KEGG Orthology) groups, rather than individual genes or proteins, so that experimental evidence in specific organisms can be extended to other organisms (*Protein Sci*, 2020, 29, 28–35). Therefore, in Figure S12, the current work is carried on eukaryotes and the identified proteins in the human signaling pathways were shown in red or green and belong to eukaryotes (Abbreviation: E). At the same time, the KEGG mapper also shows proteins from bacteria or archaea (Abbreviation: B/A) in white.

We have added the annotations to the Figure S12 and attached to the revised SI:
“Proteins from bacteria or archaea are shown in white. (Abbreviation: B/A).”

Comment 17. Data availability statement and information on PRIDE upload (or similar) are missing.

Author reply: The mass spectrometry data have been deposited to the ProteomeXchange Consortium via the PRIDE partner repository with the dataset identifier: PXD035433 (Username: reviewer_pxd035433@ebi.ac.uk Password: B1Jo6S80); PXD038658 (Username: reviewer_pxd038658@ebi.ac.uk; Password: 6q8NKAYW).

Again, we thank the reviewer for his/her time on reviewing this manuscript, and for providing invaluable comments which significantly improves the quality of this manuscript.

Reviewer 3

Overall Comment: *The manuscript from Chen et al has developed a new method for in-situ crosslinking and mapping of proteins. The experiments were well designed and the presentation was sound. The manuscript could be considered for publication in Nature Communications if the following concerns could be addressed:*

Author reply: We sincerely appreciate the reviewer’s positive comments on our work. We have seriously considered all the comments, and have made modifications to the manuscript accordingly. Please find below our response to all the comments.

Comment 1. *Line 140: “... and the α of the CD-MS method was 4.13 times higher than the in vivo XL-MS method (Fig. S8a).” But the figure shows it was 4.14?*

Author reply: We thank the reviewer for pointing out the discrepancy. The mean value of the α of the CD-MS method is 4.13667, thus we changed the sentence "... and the α of the CD-MS method was 4.14 times higher than the in vivo XL-MS method (Fig. S8a)."

Comment 2. Line 143: "... proteins in the Homo sapiens database and ..." but in the figure Uniprot database was used. It is better to be consistent.

Author reply: We appreciate the reviewer for the careful reading. Following his/her suggestion, we used Homo sapiens for keeping our analyses consistent.

Comment 3. Line 155: "Subsequently, the cross-linked proteins were extracted and digested, the cross-linked peptides were subjected to reversed-phase liquid chromatography (RPLC) fractionation, and finally, the cross-linked peptides were analyzed by high-throughput mass spectrometry." Can be simplified.

Author reply: We thank the reviewer for this recommendation. We simplified the sentence as following "the cross-linked proteins were extracted, digested and reversed-phase liquid chromatography (RPLC) fractionation for high-throughput mass spectrometry." (Line 247-249, Page 5)

Comment 4. Line 169: "Thus, the higher percentage of intermolecular cross-linked peptides was likely attributed to the higher mitochondrial protein concentrations in the living cells." What does this mean? Looks like it might be comparable, or we need to use the same conditions to test them with different methods.

Author reply: Thanks for this question. We would like to express that crosslinkers capture a higher percentage of interprotein cross-linked peptides in their native subcellular environment, where proteins are highly crowded (Protein and nucleic acid concentrations on the order of 200-400 g/L) and maintain the crowded interior of living cell (*J Mol Biol* 222,1991, 599-620; *J Am Chem Soc*, 2013, 135, 13796-13803).

Interactions of proteins with other macromolecules are highly relied on the intracellular environment, which includes ions (eg, Ca^{2+} cell signaling), pH, and membranes. Previous study has shown that a high percentage of interprotein crosslinks tends to be identify in intact organelles and cells than cell lysate, as the average density of partial-specific in cells is 730 ml/ml, which is almost a thousand times higher than in most lysate preparations (*Histochem Cell Biol*, 2005, 123, 217-228, *Mol Cell Proteomics*, 2018, 17, 216-232). In addition, most of these peptide-peptide links connected lysines within a protein. This class of cross-links is expected to be prevalent in a sample because the protein surface generally has many solvent exposed lysines whereas the site of a protein-protein interaction might be so small that only few cross-links can form (*Nat Methods*, 2008, 5, 315–318). Therefore, the crowded and natural environment of living cells are attributed to crosslinkers to capture transient or weekly protein interactions in mitochondria.

For the comparison with other methods, various NHS-ester crosslinkers have been used in many previous XL-MS studies, including BDP-NHP(*Proc Natl Acad Sci U S A*, 2017, 114, 1732-1737), DSSO(*Mol Cell Proteomics*, 2018, 17, 216-232), DSS(*J Proteome Res*, 2020, 19, 327-336) and BS3(*Mol Cell Proteomics*, 2020, 19, 1161-1178), to reveal protein structure and interactions information of mitochondrion which is isolated from different complex systems, including mice heart tissue, cultured human cell line (K-562) and cultured *S. cerevisiae* (Table R3). In human cells, we indeed observed a higher percentage of interprotein crosslinks than Ryl et al. (2.8%) (*J Proteome Res*, 2020, 19, 327-336), though such a percentage is still lower than that detected in mice (Schweppe et al. (29%), Liu et al. (61%)) or *S. cerevisiae* (Linden et al (17%)). The above difference may be due to the different properties of the applied crosslinks, including spacer arm length and physicochemical properties. In additional, cross-linking peptides have highly charged, larger size and low frequency of occurrence relative to unmodified peptides in complex samples. Strong cation exchange chromatography (SCX), fractionation with size exclusion chromatography (SEC) and affinity with biotin affinity tag as opposed to the reversed-phase chromatographic (RP-

the standard C18), have been shown to be beneficial separation of cross-linked peptides. Furthermore, distinct data acquisition and data analysis pipelines may also contribute to the differences.

Taken together, the detected cross-linked peptide information is related to the specific approaches toward XL-MS experiments, including the cross-linker, fractionation method and data analysis pipelines. The importance of our work is that we first peer mitochondria in living condition and analysis the mitochondrial crosslinking under culture conditions via nanoparticle delivery. Since the mitochondria is still part of the cell, it maintains the crowded interior of living cell fulfilled with crowd macromolecule and our results may provide additional physiological context when capturing information. In order to better explore the cross-linking information of mitochondria in the future, we still have room to improve the method in terms of experimental details.

Table R3 are added to supplementary information as Table S4.

Table R3. Cross-linking information on mitochondrion from different studies in terms of number of detected inter-peptides, number of endo/loop peptides, percentage of inter-peptides, number of proteins, unique PPI, FDR, etc.

Author	Inter Peptide number	Intra/Loop Peptide	Total Cross-links	Inter-peptide percent	Protein number	Unique PPIs	FDR	Database	Cross linker	Length	MS	Fractionation	Search Engine	Species
Devin K. Schweppe, et al	701	1851	2427	29 %	327	236	5%	Mitocarat2.0	PIR(BDN-NHP)	43Å	Velos-FT	SCX	ReACT	Mouse
Fan Liu [Ⓢ] , et al	2040	1282	3322	61.4%	359	608	2%	Mitocarat2.0	DSSO	30-40Å	Lumos	SCX	XlinkX v2.0	Mouse
Fan Liu [Ⓢ] , et al	1419	1454	2873	49.4%	290	357	2%	Mitocarat2.0	DSSO	30-40Å	Lumos	SCX	XlinkX v2.0	Mouse
Andreas Lindén [Ⓢ] , et al	359	1741	2100 [Ⓢ]	17%	263	139	1%	A database with the 400 most abundant proteins (top400)	BS3	30Å	Lumos	SEC	Plink1	S. cerevisiae

Andreas Linden et al.	265	1522	1787 [Ⓢ]	15%	260	113	1%	A database with the 400 most abundant proteins (top400)	BS3	30Å	Lumos	SEC	Plink1	S. cerevisiae
Petra S. J. Ryl et al.	152	5367	5518 [Ⓢ]	2.8%	792	152	5%	MitoCarta 2.0 database	DSS	30Å	Lumos	SCX-SD-SEC	Xi (version 1.6.731)	Human
Yuwan Chen et al.	158	394/651	1203	13.13%	501	152	1%	Mitocarta3.0 database	DSS	30Å	QE	RPLC	Plink2	Human
①. Native mitochondrion isolated from mouse heart tissue. ②. Mitochondrion isolated from mouse heart tissue and treated with salt. ③. Mitochondrion isolated from S. cerevisiae growth on Glycerol. ④. Mitochondrion isolated from S. cerevisiae growth on Glucose. ⑤. Data sets were filtered by removing ambiguous identifications, cross-links only supported by a single cross-linked peptide spectrum match (CSM) and/or with a log10-transformed pLink 1 spectrum score below 4. ⑥. Unique cross-linked residue pairs.														

In the revised manuscript we added the sentence: “In human cells, we indeed observed a higher percentage of interprotein crosslinks than Ryl et al. (2.8%), though such a percentage is still lower than that detected in mice (Schweppe et al. (29%), Liu et al. (61%)) and *S. cerevisiae* (Linden et al (17%)) by other studies. These differences may be due to the different properties of the applied crosslinks, including spacer arm length and physicochemical properties (*Chem Rev*, 2022, 122, 7647-7689), fractionation method (*Nat Protoc*, 2019, 14, 2318-2343; *Nat Methods*, 2008, 5, 315-318; *Nat Protoc*, 2014, 9, 120-137) and distinct data acquisition/analysis pipelines (*J Proteome Res*, 2020, 19, 327-336). The importance of our work is that we first peer mitochondria in living condition and analysis the mitochondrial crosslinking under culture conditions via nanoparticle delivery. Our results may provide additional physiological context when capturing information and many interactions captured by our method are dynamic or transient for our unique measurement strategy. There are still room to improve our method to achieve higher crosslinking percentage in the future.” (Line 259-270, Page 6)

Comment 5. Line 319: “Subsequently, the CLSM/Revolution XD results further confirm that the nanoparticle specifically targeted mitochondria after endocytosis (Fig.2d), instead of the endoplasmic reticulum (PCC was 0.44 (Fig. S6b)) ...”. It is great to find the overlapping of mitochondria and NP in the figure. However, the NP is larger than 100nm but Cy 5 is much smaller. Only part of the NP was marked? This also applies to Fig 2E DSS-FITC.

Author reply: Thanks for this question. Both Cy5 and DSS-FITC can completely label DDAB@PLGA/Kolliphor EL NPs, which was further confirmed by flow cytometry experiments. It is very easy to discriminate between NPs without fluorochromes labeling and the cy5 and DSS-FITC stained NPs, since the signals of the two are completely separated (Fig.R6a and Fig.R6b). In addition, we also observed a good overlap of the DSS-FITC embedded in DDAB@PLGA/Kolliphor EL NPs and Cy5-labeled DDAB@PLGA/Kolliphor EL NPs, indicating that the nanoparticles are also well loading the Cy5 and DSS-FITC at the same time in the HepG2 cells (Fig.R1f).

Fig. R6: Flow cytometry analysis the labeled NPs and unlabeled NPs. a. Encapsulation of Cy5 into DDAB@PLGA/Kolliphor EL nanoparticles (red channel). For the negative control, DDAB@PLGA/Kolliphor EL nanoparticles (grey-green channel) were not labeled with cy5. Flow cytometry analysis was performed using SH800S cell sorter (Sony Biotechnology). b. Encapsulation of DSS-FITC into DDAB@PLGA/Kolliphor EL nanoparticles (red channel). For the negative control, DDAB@PLGA/Kolliphor EL

nanoparticles (grey-green channel) were not labeled with DSS-FITC. Flow cytometry analysis was performed using SH800S cell sorter (Sony Biotechnology).

Fig.R1f. Co-localizations of the DSS-FITC which embed in the cy5-labeled DDAB-PLGA/Kolliphor EL NPs and cy5-labeled DDAB-PLGA/Kolliphor EL NPs, as measured by CLSM. Scale bars = 10 μ m.

Meanwhile, Figure R6 and R1f are compiled and added to supplementary information as Figure S6 and Figure S7f. A brief discussion was also added below the figure: “Both Cy5 and DSS-FITC could completely label DDAB@PLGA/Kolliphor EL NPs for tracking the distribution of NPs and DSS in cells, which was further confirmed by flow cytometry experiments.” (Line 168-171, Page 4)

Comment 6. Language. The language is in general very good.

- Line 26: *totally* – it could be better to state “a total of 74 pairs of ...” rather than *totally*

- Line 134: “... profiles and was approximately 28% of the total release was observed at 6 h (Table. S1).” Pls check this sentence. Probably better just use the “cumulative release”

Author reply: Thanks for the reviewer’s comment. According to the reviewer’s suggestion, “totally identified 74 pairs of ...” have been changed into “a total of 74 pairs of...”. In Line 134: It has been corrected into “Furthermore, the release of the cross-linker was quantitatively assessed, while the cumulative release was approximately 28% at 6h.”

Finally, we would like to thank the reviewer again for the critical but important comments, and hope we have clarified them appropriately.

Reviewer 4

Overall Comment: In this study Chen et al described an in situ mitochondria crosslinking approach. They use nanoparticle technology to deliver a hydrophobic crosslinker DSS directly to mitochondria and thus allow in situ crosslinking of mitochondrial proteins in their most native environment. I think this targeted crosslinking approach is a great idea and also appreciate the authors make use of specially designed mitochondrial targeting nanoparticles for the crosslinker delivery. However, I have several major points for the authors to address to improve the quality of the paper.

Author reply: We appreciate the reviewer for the careful reading and thoughtful comments, which will surely enhance the quality of this manuscript. As for the concerns raised by the reviewer, please see below the point-by-point response.

Comment 1: *I think it is very important to know if nanoparticle delivery can be a generic method for in situ crosslinking, therefore I would like to ask the authors to provide more data on if and how other types of crosslinkers can be delivered by nanoparticles and to which other organelles. Here the authors use Kolliphor EL as the nanocarrier to solubilize the hydrophobic crosslinker DSS. Does it only work for hydrophobic crosslinkers? What about hydrophilic crosslinkers? What about several commonly used MS-cleavable crosslinks such as DSSO, DSBU and DSBSO? Regarding sub-compartment targeting, are there possibilities to target other organelles other than mitochondria?*

Author reply: We thank the reviewer for the insightful comment. As suggested by the reviewer, we indeed investigated the delivery of MS-cleavable crosslinks by

DDAB@PLGA/Kolliphor EL NPs. We found that DDAB@PLGA/Kolliphor EL NPs incompatible with crosslinkers that are more hydrophilic than DSS. We found that the key factor for nanoparticles to deliver the cross-linker is the oil core property to dissolve the cross-linker or encapsulate the cross-linker into the nanoparticles to avoid hydrolysis of the cross-linker.

Commercial crosslinkers DSSO and DSBU and homemade DSBSO were applied to nanoparticles with the same formulation as 70DSS-DDAB@PLGA/Kolliphor EL NPs. Unfortunately, we did not detect the embedded MS-cleavable crosslinks in the nanoparticles by the HPLC. Thus, we infer that DDAB@PLGA/Kolliphor EL NPs incompatible with crosslinkers that are more hydrophilic than DSS. In order to successfully prepare the carrier with more kinds of crosslinkers in the future, we investigated the preparation mechanism of the crosslinker carrier from the perspective of hydrophobicity.

Fig. R7. Synthetic route of homemade DSBSO (C₂₅H₃₄N₂O₁₂S₂). DSBSO crosslinkers were used for chemical crosslinking mass spectrometry in the form of alkyne-A-DSBSO or azide-A-DSBSO with enriched group. To simplify the synthesis steps, homemade DSBSO is to replace alkyne-A or azide-A with cyclohexanone, which did not contain the enriched group.

To predict the hydrophobic/hydrophilic properties of the crosslinker, the n-octanol-water partition coefficient (log Po/w) of the cross-linker were predicted by

Swiss ADME (*Sci Rep*, 2017, 7, 42717), and its hydrophobicity order is DSSO < DSBU < DSBSO < DSS (Fig. R8a). By comparing their retention times on the C18 column, we found that the three MS cleavable cross-links elute earlier than DSS, which shows that their polarity is stronger than DSS (Fig. R8b). In general, the more polar a substance is, the more soluble it is in water. This result is consistent with its predicted log Po/w value. (Fig. R8a).

Fig. R8. Hydrophobic/hydrophilic properties of crosslinkers. a. the n-octanol-water partition coefficient (log Po/w) of the cross-linker. b. The chromatogram of cross-linker analyzed on C18.

We found that both DSSO and DSBU are only soluble in a mixture of Kolliphor EL/dichloromethane and methanol, but not completely soluble in Kolliphor EL, which differ from the oil phase (Kolliphor EL/dichloromethane) of initial parameter. Given that the above crosslinkers are not very soluble in the oily core phase, we deduce that the hydrophilic crosslinker is precipitate in the outer interphase (oil/water), rather than in the inner interphase (Kolliphor EL/shell). In this manner, as the solvent evaporates, coarse emulsion droplet phase separation with an oily core phase (Kolliphor EL) forming the oil core and a polymer-rich phase (PLGA/DDAB) forming the shell, crosslinkers incompatible with the oil core tend to a polymer-rich phase which precipitates on the oily core, forming a shell in the outer interphase (oil/water). Thus, the nanoparticles cannot avoid exposure of the crosslinkers to external interface (oil/water) and may lead to it hydrolysis.

Although the non-cleavable cross-linkers (DSS) could also achieve the reliability for proteome-scale identification of cross-linked peptides in complex systems, before

cleavable crosslinkers could be encapsulated for in vivo protein crosslinking. In future, in order to expand the range of nanoparticle delivery cross-linker, we should further explore the oily-core property to dissolve the cross-linker or encapsulate the cross-linker into the nanoparticle by adjusting the methods to avoid crosslinker hydrolysis.

As for the subcellular targeting, diverse functional materials that take subcellular structures as therapeutic targets are well developed for precise treatment of diseases (*Adv Mater*,2019,31, e1802725). Mitochondria, nucleus and endoplasmic reticulum are prime targets for various therapies because their functions are intimately linked to many cellular processes and diseases. (*Biomaterials*, 2016, 97, 10-21). For subcellular targeting, many appropriate strategies have been developed to target to specific subcellular by integrating functional groups onto specific nanoparticles surface.

For nuclear targeting strategies, nanoparticles are tuned to ultra-small size to overcome the nuclear pore complex (NPC) barrier, or with the aid of a nuclear localization sequence (NLS) that interacts with IMP family receptors and induces receptor-mediated nuclear import (*Biomaterials*, 2016, 97, 10-21). Cheng et al. demonstrated that NLS (CGGGPKKKRKVG)-functionalized PLGA nanoparticles (~72 nm) and NLS functionalized quantum dot-conjugated PLGA nanoparticles (~168 nm) could enter into the nucleus of HeLa cells (*Biomaterials*, 2008, 29, 2104-2112).

Stepensky et al. encapsulated antigenic peptide in PLGA nanoparticles with the surface conjugation of ER-targeting peptides (KKXX signal) for targeting the ER of dendritic cells. The resulting nanoparticle induced prolonged low magnitude cross-presentation of the antigenic peptide (*Mol Pharm*, 2011, 8, 1266-1275).

Given the fact that mitochondria are recognized as one of the most important targets as they are the powerhouse of the cell and have been related to cancer, cardiovascular, and neurological diseases. Currently, the most effective way to deliver drugs specifically to mitochondria are lipophilic and cationic, which harness the mitochondrial membrane potential (-160 mv to -180 mv) to enter the organelle. At the subcellular organelle level, we first achieved in situ analysis of mitochondrial protein

structure and interactions. Inspired by the above successful cases, we believe that targeting specific subcellular proteins by integrating functional groups onto specific nanoparticle surfaces has the potential for in situ analysis of other important subcellular proteins. The related studies are on-going in our laboratory.

In the revised manuscript we added the sentences: “To explore the range of nanoparticle delivery cross-linker, we investigated the delivery of MS-cleavable crosslinks by DDAB@PLGA/Kolliphor EL NPs. After fabricating the nanoparticles, we found that DDAB@PLGA/Kolliphor EL NPs incompatible with cross-linkers that are more hydrophilic than DSS. This clearly shows that the key factor for nanoparticles to deliver the cross-linker is the oil core property to dissolve the cross-linker or encapsulate the cross-linker into the nanoparticles to avoid hydrolysis of the cross-linker. In addition, inspired by a variety of successful cases of functional PLGA-based nanoparticles for subcellular targeted therapy (e.g., ER-targeting peptides (KKXX) modified PLGA NPs (*Mol Pharm*, 2011, 8, 1266-1275) and nuclear localization sequence (NLS) modified PLGA NPs (*Biomaterials*, 2008, 29, 2104-2112)) targeted therapy, we reason that targeting specific subcellular proteins by integrating functional groups onto specific nanoparticle surfaces has the potential for in situ crosslinking of other important subcellular proteins. CD-MS methods for further improvements are foreseeable at both expand the range of the kind of delivery of crosslinker and targeting to the other subcellular organelle, such as delivery of MS-cleavable crosslinks and functional nanoparticles for another subcellular targeting (for example, nuclear and endoplasmic reticulum).” (Line 490-505, Page 8)

Comment 2: *The authors identified 1203 cross-linked peptides in this study and this number is compared with an in vivo XL-MS method (reference 12) and an in vitro mitochondrial XL-MS method (I didn't find a reference for it). This comparison is not sufficient in my opinion because there are several other papers, such as Schweppe et al PNAS 2017, Liu et al, MCP 2018 and Linden et al, MCP 2020, also did in vitro mitochondria crosslinking. Although these three papers used either mouse or yeast*

mitochondria rather than human mitochondria, it is definitely feasible to perform the comparison using percentage of mitochondrial proteome coverage and protein homology. In another argument, the authors show their intermolecular crosslink counts is 13.13% which is much higher than in the isolated mitochondria of the HeLa cells (2.8%). So they claim “the higher percentage of intermolecular cross-linked peptides was likely attributed to the higher mitochondrial protein concentrations in the living cells”. However, as far as I remember, the three references I mentioned above all have much higher percentages of intermolecular crosslinks (likely higher than 13.13%). Thus, I think to fairly discuss this point, the authors should provide comparison data to the other papers I mentioned above.

Author reply: We thank the reviewer for the comments and really appreciate his/her critical opinion. According to the reviewer’s suggestion, we compared our results with several other papers, including the reviewer’s suggested papers, such as Schweppe. et al (*Proc Natl Acad Sci U S A*, 2017, 114, 1732-1737), Liu. et al (*Mol Cell Proteomics*, 2018, 17, 216-232), and Linden et al (*Mol Cell Proteomics*, 2020, 19, 1161-1178) from the aspect of detected inter peptide number, intra/loop peptide number, inter-peptide percentage, protein number, unique PPIs, FDR et al. (as shown in the Table R3).

Many XL-MS studies have explored various NHS-ester crosslinkers, including BDP-NHP(*Proc Natl Acad Sci U S A*, 2017, 114, 1732-1737), DSSO(*Mol Cell Proteomics*, 2018, 17, 216-232), DSS(*J Proteome Res*, 2020, 19, 327-336) and BS3(*Mol Cell Proteomics*, 2020, 19, 1161-1178) to reveal protein structure and interactions information of mitochondrion which was isolated from different complex systems (mice heart tissue, cultured human cell line (K-562) and cultured *S. cerevisiae*) (Table S3). As the reviewer commented, in human cells, we indeed observed a higher percentage of interprotein crosslinks than Ryl et al. (2.8%) (*J Proteome Res*, 2020, 19, 327-336), though such a percentage is still lower than that detected in mice (Schweppe et al. (29%), Liu et al. (61%)) or *S. cerevisiae* (Linden et al (17%)). The above difference may be due to the different properties of the applied crosslinks, including spacer arm length and physicochemical properties. Furthermore, distinct data

acquisition and data analysis pipelines may also contribute to the differences (e.g., Ryle et al. thought that their identification of lower PPI links is due to their separate PPI links and self-links for FDR analysis, which differs from the FDR calculations of others). In pLink 2, peptide spectrum matches PSMs are filtered within group (intra-protein, inter-protein, loop-linked, monolinked, and regular peptide spectrum matches) according to the specified thresholds (*Nat Commun*, 2019 **10**, 3404). In our work, we used the inter-cross-linked peptides, intra-cross-linked peptide and loop-linked peptide to calculate the percentage of intermolecular crosslinks. If we exclude the loop-linked peptides, the percentage of intermolecular cross-linked peptides pairs among the cross-linked peptide pairs increased to 28%. In addition, cross-linking peptides have high charged, larger size and low frequency of occurrence relative to unmodified peptides in complex samples. It has been reported that strong cation exchange chromatography (SCX), fractionation with size exclusion chromatography (SEC) and affinity with biotin affinity tag as opposed to the reversed-phase chromatographic (RP-the standard C18) (our work), has been shown to be beneficial separation of cross-linked peptides. Taken together, the detected cross-linked peptide information is related to the detail specific approaches toward XL-MS experiments, including the cross-linker, fractionation method and data analysis pipelines.

The importance to be noted in our work is that we first peer mitochondria in living condition and analysis the mitochondrial crosslinking under culture conditions via nanoparticle delivery. Since the mitochondria is still part of the cell, it maintains the crowded interior of living cell fulfilled with crowd macromolecular environment and our results may provide additional physiological context when capturing information. Some interactions that can be captured by our method are dynamic or transient. Among these, 51.32% of the PPIs were annotated in the STRING database, and half of these PPIs had high confidence (>0.7) (Fig.S13c). Among these, PPI linked OGDHL and PDHA1 filled the gap due to the lack of relevant experimental evidence in the STRING database (Fig.4c). For the non-annotated PPIs, GOBP analysis showed that these proteins were involved in the dynamics and transients of biological processes. In order

to better explore the cross-linking information of mitochondria in the future, we still have room to improve the method in terms of experimental details.

In the revised manuscript we added the sentence: “In human cells, we indeed observed a higher percentage of interprotein crosslinks than Ryl et al (2.8%), though such a percentage is still lower than that detected in *mice* (Schweppe et al (29%), Liu et al (61%)) or *S. cerevisiae* (Linden et al (17%)) by other studies. These differences may be due to the different properties of the applied crosslinks, including spacer arm length and physicochemical properties (*Chem Rev*, 2022, 122, 7647-7689), fractionation method (*Nat Protoc*, 2019, 14, 2318-2343; *Nat Methods*, 2008, 5, 315-318; *Nat Protoc*, 2014, 9, 120-137) and distinct data acquisition/analysis pipelines (*J Proteome Res*, 2020, 19, 327-336). The importance of our work is that we first peer mitochondria in living condition and analysis the mitochondrial crosslinking under culture conditions via nanoparticle delivery. Our results may provide additional physiological context when capturing information and many interactions captured by our method are dynamic or transient for our unique measurement strategy. There are still room to improve our method to achieve higher crosslinking percentage in the future.” (Line 259-270, Page 6)

Table R3 are added to supplementary information as Table S4.

Table R3. Cross-linking information on mitochondrion from different studies in terms of number of detected inter-peptides, number of endo/loop peptides, percentage of inter-peptides, number of proteins, unique PPI, FDR, etc.

Author	Inter Peptide number	Intra/Loop Peptide	Total Cross-links	Inter-peptide percent	Protein number	Unique PPIs	FDR	Database	Cross linker	Length	MS	Fractionation	Search Engine	Species
Devin K. Schweppe et al	701	1851	2427	29 %	327	236	5%	Mitocarat2.0	PIR(BDN-NHP)	43Å	Velo-s-FT	SCX	ReACT	Mouse
Fan Liu et al	2040	1282	3322	61%	359	608	2%	Mitocarat2.0	DSSO	30-40Å	Lumos	SCX	XlinkX v2.0	Mouse

Fan Liu [Ⓢ] , et al	1419	1454	2873	49%	290	357	2%	Mitocarat2.0	DSS O	30-40Å	Lumos	SCX	XlinkX v2.0	Mouse
Andreas Lindén [Ⓢ] , et al	359	1741	2100 [Ⓢ]	17%	263	139	1%	A database with the 400 most abundant proteins (top400)	BS3	30Å	Lumos	SEC	Plink1	S. cerevisiae
Andreas Lindén [Ⓢ] , et al	265	1522	1787 [Ⓢ]	15%	260	113	1%	A database with the 400 most abundant proteins (top400)	BS3	30Å	Lumos	SEC	Plink1	S. cerevisiae
Petra S. J. Ryl, et al	152	5367	5518 [Ⓢ]	2.8%	792	152	5%	MitoCarta 2.0 database	DSS	30Å	Lumos	SCX-SD-SEC	Xi (version 1.6.731)	Human
Yuwan Chen, et al	158	394/651	1203	13%	501	152	1%	Mitocarta3.0 database	DSS	30Å	QE	RPLC	Plink2	Human
①. Native mitochondrion isolated from mouse heart tissue. ②. Mitochondrion isolated from mouse heart tissue and treated with salt. ③. Mitochondrion isolated from S. cerevisiae growth on Glycerol. ④. Mitochondrion isolated from S. cerevisiae growth on Glucose. ⑤. Data sets were filtered by removing ambiguous identifications, cross-links only supported by a single cross-linked peptide spectrum match (CSM) and/or with a log10-transformed pLink 1 spectrum score below 4. ⑥. Unique cross-linked residue pairs.														

Comment 3: *The structural analyses in this paper are mostly standard. This is okay as a technology paper, but I would suggest to significantly shorten these parts. In particular, “NDUFA4 is a subunit of complex IV instead of complex I” has been described in 2012 (Balsa E et al Cell metabolism, 2012) and well accepted now so there is no need to put emphasize on it and make a main figure for it. Furthermore, the authors also say “the detected crosslinks also provide new structural information for several ADP/ATP carriers, such as SLC25A12 (a typo here, SLC not SCL) SLC25A13 and SLC25A22. However, in Fig. S14c, they only did standard crosslinking mapping and measured the C-alpha distances. I couldn’t find any structural elucidation on the dynamics of these carrier proteins, as the authors stated on Page 7, line 308.*

Author reply: We sincerely thank the reviewer for this considerate comment. In the revised manuscript, we have readjusted Figure 5d to SI as Figure S18 to summarize the structural analysis part. In addition, “SCL25A12” has been corrected to “SLC25A12” in the revised manuscript. In the previous version of the manuscript, we only analyzed the cross-links of the human mitochondrial ADP/ATP carriers (SLC25A4, SLC25A5, SLC25A6 and SLC25A31) to the crystal structure, thus we analyzed the rest of the cross-linked SLC25 family protein model information. We found that most of the protein sites involved in cross-linked peptides are located in unresolved structural regions of the protein and one crosslinker further provides information on the site of dimerization of the N-terminal domain of citrin.

In the revised manuscript we attached Fig.R9 to the figure S21 in the revised *SI* document and added the sentence: “For the identified SLC25 family proteins, 54 unique cross-linked peptides were identified belonging to 16 proteins. Among them, most of the protein sites involved in the cross-linked peptides are located in unresolved structural regions of the protein, thus the cross-linked peptides were mapped to the predicted AF model. Approximately 98% of the cross-linked peptides (53 of 54) were well mapped to the AF Fold structure with a C α -C α distance of less than 30 Å. (Line 408-413, Page 9)

Protein-protein crosslinks reveals Lys-235 self-linked in the citrin (SLC25A13) N-terminal domain, further providing a site for the N-terminal domain of citrin to mediate dimerization of the human full-length aspartate/glutamate carrier(ref). An incompatible intramolecular cross-link (K74, K243) involved in SLC25A24 was not compatible with the predicted AF model, while the residue positions linked were predicted with very high confidence, but two segments of the AF model of SLC25A24 (Residues 167–190 and 282–289) have a low predicted scores below 70 in the predicted local distance difference test (PLDDT). We inferred that the incompatible cross-links might be caused by the flexible region of the protein, and that the cross-links might provide a distance-constrained reference between these two residues.

Members of the mitochondrial carrier family (SLC25) are typically located in mitochondrial inner membrane and perform the vital role of shuttling amino acids, carboxylic acids, inorganic ions and cofactors between the mitochondrial matrix and the cytosol (*Trends Biochem Sci*,2020, 45, 244-258, *Mol Aspects Med*, 2013, 34, 465-484). Although affected by protein expression and purification, the structures of several SLC25 family members remain to be characterized, but the detected protein cross-linking site information can provide complementary information for the structures of SLC25 family members.” (Line 422-439, Page 10)

Fig.R9. Structural match of detected cross-linked peptides of SLC25 family. Each

chain of protein was colored in a different color. There is no available structure information for SLC25A1, SLC25A3, SLC25A11, SLC25A18, SLC25A20, SLC25A22, SLC25A26 and SLC25A42, thus the detected cross-linked peptides from the above proteins were mapped to the AlphaFold predicted model. Some of the cross-linked peptide residues detected are located in the unresolved region of the reported protein structure, including SLC25A25((Lys-74 to Lys-243, Lys-190 to Lys-477, Lys-320 to Lys-243, Lys-228 to Lys-437, Lys-333 to Lys-336 and Lys-268 to Lys-276), SLC25A12 (Lys-234 to Lys-244 and Lys-403 to Lys-406), SLC25A13 (Lys-235 to Lys-5, Lys-235 to Lys-245, Lys-235 to Lys-312, Lys-405 to Lys-408 and Lys-353 to Lys-580), thus the detected cross-linked peptides from the above proteins were mapped to the AlphaFold model.

Taken together, I think this paper should be substantially revised by providing more data on the technology itself (my point 1 and 2) and significantly shorten and tone down the structural analysis part unless any new structural insights was indeed discovered by the crosslinks found in this study. For the latter, the authors need to provide sufficient modeling and/or structural data to support these new findings.

We appreciate all these insightful comments. For the point 1 and 2, extensive experiments were carried out on the technology and the corresponding discussions are included in the revised manuscript. The structural analysis part was modified in the revised manuscript, and we sincerely hope the revised discussions can provide helpful information on the structures, dynamics and functions of SLC25 family.

As the reviewer kindly pointed out, the current work is designed to explore in situ crosslinking of mitochondrial proteins in their most native environment, which has long-time been regarded very important but hard to carry out in experiments. We hope the breakthroughs achieved, though could still be primary in methodology, can inspire more efforts in this area with the broad readership afforded by the superior platform of Nature Communications.

REVIEWER COMMENTS

Reviewer #2 (Remarks to the Author):

While the authors addressed all my comments, I am disappointed about the implementation. Almost every new section of the manuscript contains many mistakes. I do not have the impression that the revised manuscript was checked carefully. In addition, the authors state in the reporting summary that only technical replicates (i.e. the same sample was measured several times) were performed. I somehow doubt the outcome of the study if not a single biological replicate was performed. I cannot recommend publication of the manuscript in the current state.

Manuscript:

Line 67: ...under native condition... -> should read under native conditions

Line 68: protein -> should read proteins

Line 69: efficient -> should read efficiently

Line 70: The reference 'Terry Wilke' doesn't exist -> should be corrected to Tayri-Wilk et al (see also reference list provided with the manuscript, which contains the correct reference name)

Line 73: In additional, ... -> should read "In addition..."

Line 158-160: This sentence is difficult do understand. What is caused? Do the authors mean "affected"?

Line 160: In additional -> should read "In addition..."

Line 307: "What's more, ..." -> This is not correct English. The rest of the sentence is also not grammatically correct.

Line 308: The residues that connected -> should read "The residues that are connected"

Line 309: cross-links are fall within -> deleted are or fall

Line 311: "actual result" -> very colloquial

Supplementary Information:

Line 217/128: The pLink -> should read "pLink" or "The software pLink"

General comments:

Supplementary Tables and Figures are mixed.

If the response of the authors is correct, only technical replicates were performed (i.e. the same sample was measured several times). There is no information on replicates given in the entire manuscript (only in the reporting summary). I somehow doubt the outcome of this study when not even a single biological replicate was performed.

Reviewer #3 (Remarks to the Author):

The concerns raised previously were addressed adequately.

Reviewer #4 (Remarks to the Author):

The authors nicely addressed my comments in the response letter. However, I am puzzled why some of these knowledge are not sufficiently incorporated in the main text. As I believe these information are also important to other readers, I'd like to ask the authors to kindly include them (in a more concise manner) in the main text. In particular,

- 1) To my comment 1, adding the hydrophobicity comparison of DSS, DSBU, DSSO and DSBSO and their HPLC elution profile in the result session, not only in the discussion. State clearly DSSO, DSBU and DSBSO are not deliverable.
- 2) "CD-MS methos for further improvements are foreseeable at both expand the range of the kind of delivery of crosslinker and targeting to the other subcellular orange" What is CD-MS? "methos" is a typo.
- 3) In the reply to my comment 2, "In human cells, we indeed observed a higher percentage of interprotein crosslinks than Ryl et al (2.8%), though such a percentage is still lower than that detected in mice (Schweppe et al (29%), Liu et al (61%)) or *S. cerevisiae* (Linden et al (17%)) by other studies." Please specify the percentage observed in this study.
- 4) In the reply to my comment 3, "Among them, most of the protein sites involved in the cross-linked peptides are located in unresolved structural regions of the protein, thus the cross-linked peptides were mapped to the predicted AF model. Approximately 98% of the cross-linked peptides (53 of 54) were well

mapped to the AF Fold structure with a C α -C α distance of less than 30 Å. (Line 408-413, Page 9)". Please use AlphaFold rather than AF to be consistency with the rest of the paper.

5) The authors did not give a proper response to my Ndufa4 comment. Please at least shorten the descriptions of Nudfa4.

Response to the comments from the reviewers, manuscript NCOMMS-22-23692-A

Reviewer 1

Comment 1: *“To elucidate the effect of adsorbed proteins on nanoparticles on in situ mitochondrial crosslinking, the interactions of 70DSS-DDAB@PLGA/Kolliphor EL NPs with proteins at different stages before targeting to mitochondrion.” This is not a complete sentence.*

Author reply: Thanks for the kind notification. Following the reviewer’s suggestion, we have therefore rephrased this sentence in a more careful manner in the revised manuscript. **“To elucidate the effect of proteins adsorbed on the NPs on in situ mitochondrial cross-linking, the interactions of the 70DSS-DDAB@PLGA/Kolliphor EL NPs with proteins in different biological media were identified before targeting the mitochondria.” (Line 198–200, Page 5)**

Comment 2: *It was explained: “In additional, at each stage, approximately half of the proteins adsorbed to the nanoparticles were weakly adsorbed and desorbed on the way to the next stage. Thus, we infer that the DSS-DDAB@PLGA/Kolliphor EL NPs would adsorb proteins during delivery, but small part of the proteins was strongly bound on the nanoparticles. In addition, we found that the mitochondrial proteins detected in our manuscript were independent of the proteins absorbed on the nanoparticles prior to targeting mitochondria, so proteins absorbed on the nanoparticles did not affect in situ release of the cross-linker in the mitochondria for cross-linking.” This confusing. If they were weakly adsorbed and desorbed, why only a small portion of proteins were strongly bound?*

Author reply: We thank the reviewer for this considerate suggestion. In our study, weakly adsorbed and desorbed proteins on nanoparticles (NPs) means low-affinity proteins on the NPs (soft corona), while strong binding proteins on the NPs means high affinity to NPs during delivery (hard corona).

The interaction of NPs with proteins in biological media is non-covalent and thermodynamically bound (*Colloids Surf B Biointerfaces*, 2020, 188, 110816). It occurs

when NPs enter biological fluids (e.g., blood, cellular component), and biomolecules spontaneously form an adsorption layer around the NPs, called “protein corona” (*ACS Nano*, 2016, 10, 10842-10850). As for the protein's adsorption layer, it involves a high-affinity layer (hard corona) that interacts stably with the NP surface and a low-affinity layer (soft corona) that enables dynamic exchange in solution. Generally, a hard corona forms on the surface of NPs in the initial biological environment when NPs pass through different biological media (*Rep Pract Oncol Radiother*, 2018, 23, 300-308). Subsequently, as the biological environment changes, soft corona proteins undergo secondary interactions with NPs in the presence of hard corona proteins, and the corona composition begins to evolve due to competition between proteins. Finally, the final corona composition of NPs largely depends on the physicochemical properties of the NPs (*Colloids Surf B Biointerfaces*, 2020, 188, 110816).

Taken together, we have revised the elaboration in the revised manuscript: “At each stage, approximately half of the proteins adsorbed on the NPs were in the soft corona layer, where they weakly adsorbed on the NP surface and dynamically exchanged in the biological fluids. Considering that approximately 8.2% of all adsorbed proteins (21 overlapping proteins) were consistently present during the entire sequential incubation process, we reasoned that the DSS-DDAB@PLGA/Kolliphor EL NPs adsorbed proteins during delivery, but the hard corona, the proteins strongly bound to the NPs, only accounted for a small fraction of the total.” (Line 219–225, Page 5)

Reviewer2

Overall Comment: *While the authors addressed all my comments, I am disappointed about the implementation. Almost every new section of the manuscript contains many mistakes. I do not have the impression that the revised manuscript was checked carefully. In addition, the authors state in the reporting summary that only technical replicates (i.e. the same sample was measured several times) were performed. I somehow doubt the outcome of the study if not a single biological replicate was performed. I cannot recommend publication of the manuscript in the current state.*

Author reply: We sincerely thank the reviewer for raising this important concern.

According to the reviewer's suggestion, in this revision, all the implementation was double-checked from the technical and linguistic perspective. Moreover, we have improved the language of the manuscript with the help of LetPub (www.letpub.com). For the biological replication, actually, it has been included in the previous revision from two experiments, the proteome perturbation experiment and fluorescent experiments for co-localization of the nanoparticles with the mitochondria, including Figure S2 in manuscript NCOMMS-22-23692 and Figure S2c in manuscript NCOMMS-22-23692-A; and Figure 2d and Figure S7c in manuscript NCOMMS-22-23692-A. The relevant content is shown as below for the reviewer's convenience.

Fig. S2. Significant changes of protein abundances. Protein changes at the cellular protein levels in cells treated with the highest loading efficiency nanoparticle for 6 h, which were measured using the label-free protein quantification method. Comparison with the untreated conditions, showing that the LFQ intensity of the protein significantly changed by more than 2 times (p value < 0.01), as shown by the red dots, while the remaining proteins are shown as black dots. (In manuscript NCOMMS-22-23692) (We want to clarify here that the highest loading efficiency nanoparticle is 70DSS-DDAB@PLGA/Kolliphor EL NPs.)

Fig. S2. The effect of DDAB content in the NPs on the percentage of proteins with 2-fold significant changes at the proteome level in the treated HepG2 cells. **a** Size and zeta potential of the DSS-DDAB@PLGA/Kolliphor EL NPs with different amounts of DDAB in pure water, as measured using DLS. Data are shown as mean \pm SD from three independent experiments. **b** Significant changes in protein abundance. The protein changes at the cellular protein levels in cells treated with the highest loading efficiency NPs for 6 h were measured using the label-free protein quantification method (Treated-1 to -5). **c–g** Label-free quantification was performed to calculate the changes in protein abundance between the native and different 70DSS-DDAB@PLGA/Kolliphor EL NP-treated HepG2 cells. The DDAB content in the different 70DSS-DDAB@PLGA/Kolliphor EL NPs was changed to 80% (Treated-2), 60% (Treated-3), 40% (Treated-4), and 20% (Treated-5) of the initial content (Treated-1), respectively. Proteins with a significant change in LFQ intensity of more than 2-fold (p -value < 0.01) in the treated group compared with the untreated condition are shown as red dots, while the remaining proteins are shown as black dots. (In manuscript NCOMMS-22-23692-A)

Fig. 2d Co-localizations of the PLGA/Kolliphor EL NPs/mitochondria, as measured by CLSM. The PLGA/Kolliphor EL NPs and mitochondria were labeled by Cy5 (red) and Mito Tracker Green FM (green), respectively. (In manuscript NCOMMS-22-23692-A)

Fig. S7c Co-localizations of the DDAB@PLGA/Kolliphor EL NPs/mitochondria, as measured by Revolution WD. The DDAB@PLGA /Kolliphor EL NPs and mitochondria were labeled by Cy5 (red) and Mito Tracker Green FM (green), respectively. (In manuscript NCOMMS-22-23692-A)

The results of both experiments showed good reproducibility. In the revised manuscript and reporting-summary, we state this clearly. Furthermore, we accordingly performed biological replicate experiments to measure the amount of protein corona on the NPs, as well as three independent experiments to prepare and characterize the NPs. All the above-mentioned replicates have been clearly stated in the revised manuscript. As for the concerns raised by the reviewer, please see below the point-by-point response.

Comment 1: Line 67: ...under native condition... -> should read under native conditions

Line 68: protein -> should read proteins

Line 69: efficient -> should read efficiently

Line 70: The reference 'Terry Wilke' doesn't exist -> should be corrected to Tayri-Wilk et al (see also reference list provided with the manuscript, which contains the correct reference name)

Line 73: In additional, ... -> should read "In addition..."

Line 158-160: This sentence is difficult do understand. What is caused? Do the authors mean "affected"?

Line 160: In additional -> should read "In addition..."

Line 307: "What's more, ..." -> This is not correct English. The rest of the sentence is also not grammatically correct.

Line 308: The residues that connected -> should read "The residues that are connected"

Line 309: cross-links are fall within -> deleted are or fall

Line 311: “actual result” -> very colloquial

Supplementary Information:

Line 217/128: The pLink -> should read “pLink” or “The software pLink”

General comments:

Supplementary Tables and Figures are mixed.

Author reply: We thank the reviewer for the careful comments and have made modifications to the manuscript accordingly.

-Line 67: “...under native condition...” has been corrected to “...under native conditions, ...” (Line 68, Page 2).

-Line 68: “...protein...” has been corrected to “...proteins...” (Line 69, Page 2).

-Line 69: “...efficient...” has been corrected to “...efficiently...” (Line 70, Page 2).

-Line 70: “...Terry Wilke...” has been corrected to “...Tayri-Wilk et al. ...” (Line 71–72, Page 2).

-Both Line 73 and Line 160: “In additional, ...” has been corrected to “In addition, ...” (Line 74, Page 2 and Line 159, Page 4).

-Line 158-160: The sentence has been rephased in a more careful manner in the revised manuscript “Thus, the cross-linker-based nanocarriers did not affect most of the detected 3002 cellular proteins (2856 cellular proteins), which were not significantly different from the control at the proteomic level.” (Line 157–159, Page 4).

- Line 307: “What’s more, ...” has been corrected to “Furthermore, ...” (Line 308, Page 7).

- Line 308-311: The rest of the sentence has been corrected to “Furthermore, two cross-links were incompatible with the predicted AlphaFold structure. The residues, which were connected by the two cross-links, fell within a low model confidence region with the predicted local distance difference test (PLDDT), with a score below 70. Therefore, the discrepancy between the predicted and experimental structures was possibly the reason for the incompatibility of the detected cross-links with the AlphaFold model.” (Line 308–313, Page 7).

-Line 217/128: “The pLink ...”> has been corrected to “The software pLink” (Line 234, Page 5 in revised supporting information)

According to the suggestion, we changed the order of the figures and tables to show them separately in the revised supporting information.

Please note that there are other modifications to the manuscript in the spirit of improving the quality, which are highlighted in the enclosed revised manuscript. Furthermore, the revised manuscript and supporting information have been polished by LetPub (www.letpub.com).

Comment 2: *If the response of the authors is correct, only technical replicates were performed (i.e. the same sample was measured several times). There is no information on replicates given in the entire manuscript (only in the reporting summary). I somehow doubt the outcome of this study when not even a single biological replicate was performed.*

Author reply: We sincerely thank the reviewer for raising this important concern. In the first review of our work, in response to reviewer #1's comment on our work, the proteome perturbation experiments and the fluorescence experiments of co-localization of the DSS-DDAB@PLGA/Kolliphor EL NPs with mitochondria have been reproduced. As shown above, in the treatment group (70DSS-DDAB@PLGA/Kolliphor EL NPs), the percentage of 2-fold significantly different proteins was 5.2% for the first detection (Fig. S2, NCOMMS-22-23692) and 4.9% for the second detection (Fig. S2c, NCOMMS-22-23692-A). The Pearson correlation coefficient (PCC) of the red fluorescence of DSS-DDAB@PLGA/Kolliphor EL NPs and the green fluorescence of Mito-Tracker was 0.73 for the first detection (Fig. 2d, NCOMMS-22-23692-A) and 0.72 for the second detection (Fig. S7c, NCOMMS-22-23692-A). Thus, the conclusions obtained are consistent with all our earlier observations, showing good reproducibility of the experiment.

Considering the conclusion of our method, we also repeated the experiment which quantified the protein amount of three specific protein corona on nanoparticles (CDNP^D, CDNP^{DL}, and CDNP^{DLC} complexes), and we obtained a similar result in the replicate.

Therefore, the conclusions for protein content within protein coronas in the revised manuscript are the average of two biological replicates. Relevant conclusions in revised manuscript are shown below for the reviewer's convenience: "The amount of protein on the NPs was quantified as ~0.164 fg per particle on the CDNP^D, ~0.380 fg per nanoparticle on the CDNP^{DL}, and ~2.543 fg per particle on the CDNP^{DLC} (Fig. S11a)." (Line 210–212, Page 5)

Relevant conclusions in the previous manuscript are shown below for comparison by reviewers: "The amount of protein on nanoparticles was quantified as ~0.115 fg per particle on CDNPs^D, ~0.274 fg per particle on CDNPs^{DL}, and ~2.692 fg per particle on CDNPs^{DLC} (Fig. S11a)."

In addition, the reproducibility of DSS-DDAB@PLGA/Kolliphor EL NPs is also important to our work. Thus, the results of DSS-DDAB@PLGA/Kolliphor EL NPs, including size and zeta potential as well as the loading efficiency, were verified by repeated experiments; each experiment was repeated three times independently and each replicate was measured triplicate. In addition, the conclusions about DSS-DDAB@PLGA/Kolliphor EL NPs in revised the manuscript are the average results of three independent experiments. Relevant conclusions in revised manuscript are shown below for the reviewer's convenience: "By exposing the hydrophilic ammonium cation on the PLGA/DDAB shell of the nanocarrier, the DSS-DDAB@PLGA/Kolliphor EL NP surfaces had a positive charge of ~37.9 mV (Figs. 2b and S7a), and the surface charge of the DSS@PLGA/Kolliphor EL NPs without DDAB was approximately -25.1 mV (due to the carboxylic groups) (Fig. S7a). The loading efficiency of DSS in the DSS-DDAB@PLGA/Kolliphor EL NPs was optimized by adjusting the amount of DSS, which was increased to 23.21%(w/w) (Fig. 2c) without causing changes in the size or zeta potential (Fig. 2b; Table S1)." (Line 143–150, Page 3-4)

Relevant conclusions in the previous manuscript are shown below for comparison by reviewers: "By exposing hydrophilic ammonium cation on the PLGA/DDAB shells of the nanocarrier, the surface of DSS-DDAB@PLGA/DDAB NPs have a positive charge of ~38.5 mV (Fig. 2b, Fig. S7a), and the spherical surface charge of

DSS@PLGA/Kolliphor EL NPs without DDAB is ~ -18 mV (due to carboxylic groups) (Fig. S7a). The DSS loading efficiency in the nanoparticle was optimized by adjusting the amount of DSS, which was increased to 24.84%(w/w) (Fig. 2c) without causing changes in the size or zeta potential.”

We sincerely appreciate the reviewer for the careful reading of the manuscript and these helpful comments for elevating the quality of the manuscript.

Reviewer 4

***Overall Comment:** The authors nicely addressed my comments in the response letter. However, I am puzzled why some of these knowledge are not sufficiently incorporated in the main text. As I believe these information are also important to other readers, I'd like to ask the authors to kindly include them (in a more concise manner) in the main text. In particular,*

Author reply: We appreciate the reviewer's positive assessment. Please find below our point-by-point response to all the comments.

***Comment 1:** To my comment 1, adding the hydrophobicity comparison of DSS, DSBU, DSSO and DSBSO and their HPLC elution profile in the result session, not only in the discussion. State clearly DSSO, DSBU and DSBSO are not deliverable.*

Author reply: We thank the reviewer for the constructive suggestions. Following this suggestion, we have added discussions on the delivery capabilities of DDAB@PLGA/Kolliphor EL NPs for DSSO, DSBU and DSBSO on Page 10, as:

“**Application of CD-MS method to MS cleavable crosslinks.** To explore whether the CD-MS method could deliver MS-cleavable cross-linkers, DDAB@PLGA/Kolliphor EL NPs were used to re-load three MS-cleavable cross-linkers, including DSSO and DSBU, as well as custom-made DSBSO. According to the predicted n-octanol-water partition coefficient ($\log P_{o/w}$) of the cross-linker (*Sci Rep, 2017, 7, 42717*) (Fig. S22a) and the elution time of the cross-linker on the C18 column, we found three MS-cleavable cross-linkers that were more hydrophilic than DSS, with a hydrophobicity order of DSSO < DSBU < DSBSO < DSS (Fig. S22b). During the preparation process of the NPs, we found that both DSSO and DSBU were

only soluble in a mixture of Kolliphor EL/dichloromethane and methanol, but not completely soluble in Kolliphor EL, which differed from the oil phase (Kolliphor EL/dichloromethane) of the initial parameter. Finally, we did not detect the embedded MS-cleavable cross-linkers in the NPs using high-performance liquid chromatography (HPLC).

Given that the above MS-cleavable cross-linkers were not very soluble in the oily core phase, we deduced that the hydrophilic MS-cleavable cross-linker precipitated in the outer interphase (oil/water), rather than in the inner interphase (Kolliphor EL/shell). In this manner, as the solvent evaporated, coarse emulsion droplet phase separation with an oily core phase (Kolliphor EL) formed the oil core, and a polymer-rich phase (PLGA/DDAB) formed the shell. The MS-cleavable cross-linkers incompatible with the oil core tended to form a polymer-rich phase, which precipitated on the oily core, forming a shell in the outer interphase (oil/water). Thus, the NPs did not protect the MS-cleavable cross-linkers from hydrolysis. Overall, the DDAB@PLGA/Kolliphor EL NPs could not efficiently deliver these three MS-cleavable cross-linkers, which were more hydrophilic than DSS. In addition, we found that the key factor for the NPs to deliver the cross-linker was the oil core property, which dissolved the cross-linker or encapsulated the cross-linker into the NPs to avoid hydrolysis of the cross-linker.” (Line 434–459, Page 10)

Comment 2: “CD-MS methos for further improvements are foreseeable at both expand the range of the kind of delivery of crosslinker and targeting to the other subcellular orange” What is CD-MS? “methos” is a typo.

Author reply: In this study, we developed a CD-MS method that enables in situ mapping of mitochondrial proteins by targeted crosslinker delivery coupled with mass spectrometry (CD-MS). In addition, the relevant description is as follows in the revised manuscript: “We developed a method for in situ mitochondrial protein mapping with targeted cross-linker delivery coupled with mass spectrometry (CD-MS). Disuccinimidyl suberate (DSS) was encapsulated in the PLGA NPs, which were functionalized with DDAB for targeting mitochondria. After intracellular delivery of

the cross-linker, the cross-linked proteins were extracted through ionic liquid filtration assisted sample preparation (i-FASP). The peptides were then fractionated using reversed-phase liquid chromatography (RPLC) for measurement in a high-throughput mass spectrometer, followed by the analysis of the protein-protein interactions with Ca-Ca distances of less than 30 Å, serving as an upper bound distance for DSS cross-linking supported by molecular dynamics simulations.” (Line 89–98, Page 2-3)

Comment 3: *In the reply to my comment 2, “In human cells, we indeed observed a higher percentage of interprotein crosslinks than Ryl et al (2.8%), though such a percentage is still lower than that detected in mice (Schweppe et al (29%), Liu et al (61%)) or *S. cerevisiae* (Linden et al (17%)) by other studies.” Please specify the percentage observed in this study.*

Author reply: Thank you for pointing it out, and we have made modifications to the manuscript accordingly. It has been corrected into “In human cells, we observed 13% inter-protein cross-links, which was higher than the results obtained by Ryl et al. (2.8%) (10), though such a percentage was still lower than that detected in mice (Schweppe et al. (29%) (8), Liu et al. (61%) (9)) or *S. cerevisiae* (Linden et al. (17%) (14) by other studies (Table S4).” (Line 260–263, Page 6)

Comment 4: *In the reply to my comment 3, “Among them, most of the protein sites involved in the cross-linked peptides are located in unresolved structural regions of the protein, thus the cross-linked peptides were mapped to the predicted AF model. Approximately 98% of the cross-linked peptides (53 of 54) were well mapped to the AF Fold structure with a Ca-Ca distance of less than 30 Å. (Line 408-413, Page 9)”. Please use AlphaFold rather than AF to be consistency with the rest of the paper.*

Author reply: Thank you for pointing it out, and all “...AF...” have been corrected to “...AlphaFold...” in the revised manuscript.

Comment 5: *The authors did not give a proper response to my Ndufa4 comment. Please at least shorten the descriptions of Nudfa4.*

Author reply: In the revised manuscript, we have rewritten the following sentences: “Furthermore, the correction of the NDUFA4 subunit assignment was interpreted by

subsequent studies, as NDUFA4 was sensitive to electrostatic perturbation, readily repositioned, and was capable of promiscuous binding when perturbed”. And “Specifically, the inter-cross-linked distance between Lys-116 of COX6B1 and Lys-73 of NDUFA4 was 14.5 Å in the CIV structure (PDB code:5Z62), which further verified the findings in a previous study, where NDUFA4 was the subunit of complex IV instead of complex I (Fig. S18).

Thus, we have therefore rephrased the above sentence in a shortened manner in the revised manuscript: “Although the position of NDUFA4 in human SC_IIII₂IV₁ was easily disturbed by the environment (9), we still identified that the inter-cross-linked distance between the Lys-116 of COX6B1 and the Lys-73 of NDUFA4 in the CIV structure (PDB code:5Z62) was 14.5 Å, which was consistent with previous studies (Fig. S18).” (Line 341–345, Page 8) In addition, the following sentence has been removed from the conclusion of the revised manuscript: “the cross-linking between COX6B1 and NDUFA4 further validates the findings in previous studies (40), which showed that NDUFA4 is a subunit of complex IV rather than complex I (Fig. S18).”

Again, we want to express our sincere appreciation to the reviewer for the careful review and kind suggestions, which are beneficial for our understanding and largely raise the quality of our manuscript.

REVIEWERS' COMMENTS

Reviewer #2 (Remarks to the Author):

My concerns raised in the previous round of review were addressed adequately.